# FEDNet: Frequency Enhanced Decomposed Network for Out-of-Distribution Time Series Classification

## Abstract

Time series classification is a crucial task with widespread applications in various fields such as medicine and energy. Due to the non-stationary property of time series, its data distribution will change over time, which makes it challenging for models to generalize to the *out-of-distribution* (OOD) environment. However, limitations persist in the current research on OOD time series classification, particularly the absence of a unified consideration addressing both domain distribution shift and temporal distribution shift. To this end, we view the time series distribution shift from the frequency perspective and propose a novel method called ***F**requency **E**nhanced **D**ecomposed **Net**work* (FEDNet) for OOD time series classification. FEDNet utilizes frequency domain information to guide the decomposition of time series and further eliminates domain shift and temporal shift, it then obtains domain-invariant features for adapting to OOD data. Finally, we provide theoretical insights of FEDNet to validate its superiority for OOD time series classification. Comprehensive results on synthetic and real-world datasets demonstrate that FEDNet achieves state-of-the-art performance in OOD time series classification tasks, surpassing previous methods by up to 7%. Our code is available at https://anonymous.4open.science/r/FEDNet-743E.

## 1 Introduction

Time series classification is a pivotal task with applications in bio-signals processing (Salehi et al., 2021), medical diagnostics (Supratak et al., 2017), and human activity recognition (Tang et al., 2020). Recent studies have introduced methods like TCN (Bai et al., 2018) and shallow RNN (Dennis et al., 2019) for this purpose, largely adhering to the *independently identically distributed* (i.i.d) assumption. However, this assumption no longer holds in reality as the testing data do not always follow the same distribution as the training data, i.e. *out-of-distribution* (OOD), thus the performance on the testing data is severely degraded.

In practical situations, it proves to be highly challenging to acquire the distribution of the testing set (Wang et al., 2022b). As shown in Figure 1, we can only get training samples from a finite number of domains, these domains usually represent different types of populations, while the testing data is invisible and inaccessible in reality. Therefore, we cannot utilize the testing data for domain adaptation (Patel et al., 2015) and the limited samples are not enough to build a powerful pre-training model for transfer learning (Pan & Yang, 2009). How to use limited domain datasets to improve the model's generalization on datasets in unseen domains becomes a realistic problem. Moreover, traditional domain generalization methods following the *invariant risk minimization* (IRM) paradigm are suboptimal to OOD time series classification since the marginal probability distribution of time series data would change when facing non-stationary situations, which may cause their distribution to deviate from the corresponding domains.

Unfortunately, research on OOD generalization for time series classification is limited. Existing methods can be divided into two main types, domain relabeling and disentangled representation learning methods. On the one hand, the domain relabeling methods (Lu et al., 2022; Du et al., 2021) try to divide the time series into groups of segments with large distance in data distribution and relabel its domain for training. On the other hand, the disentangled representation learning methods

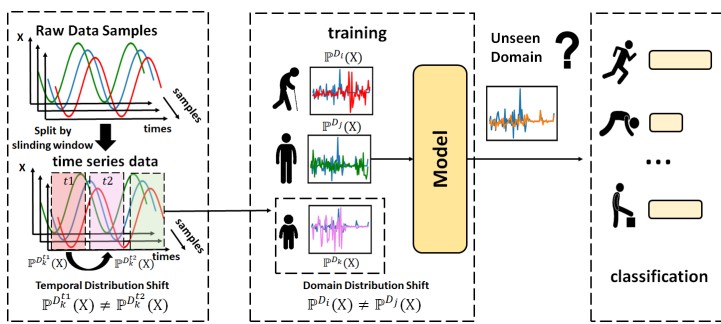

Figure 1: OOD time series classification scenario. The cross-people division (by age or other rules) causes domain distribution shift, while different periods caused temporal distribution shift.

(Qian et al., 2021) expect to separate the data into domain-invariant and domain-specific parts. The domain-invariant part represents features that remain consistent or unchanged across domains, while the domain-specific part represents features that exhibit variations or specificity in different domains. Nonetheless, domain relabeling methods directly utilize the whole segments to reset domain labels but ignore the noise and redundant information between segments, and disentangled representation learning methods ignore temporal distribution shift within the domain by directly utilizing the domain labels to aid training. In addition, these methods analyze distribution shift directly from the time domain without considering the global view of the frequency information in time series.

To fill the gap of frequency information and deal with temporal distribution shift in OOD time series classification, we propose a method called *Frequency Enhanced Decomposed Network* (FEDNet). Inspired by Wold's Theorem (Jenkins et al., 1955) and Koopa's (Liu et al., 2023) successful practice in non-stationary time series forecast, we realized that time series can be decomposed into time-deterministic component and time-stochastic component by frequency information. The time-deterministic part is less affected by the temporal changes, while the time-stochastic part mainly depends on temporal changes. It encouraged us to introduce frequency domain information and time series decomposition ideas into OOD time series classification to analyze the data in different domains. Moreover, we use both *fast Fourier transform* (FFT) and *Discrete Wavelet Transform* (DWT) to extract a certain percentage of high average amplitude frequency components as time-deterministic features by scanning all training data in different domains. Specifically, the time-deterministic features are less affected by time and can be decoupled to extract domain-invariant features, and the remaining frequency components are used to model the temporal stochastic components. Finally, we theoretically compare FEDNet with previous OOD generalization methods.

The contributions of our work can be summarized as follows:

- To the best of our knowledge, it is the first work to investigate OOD time series classification from the frequency perspective. Additionally, we postulate and formulate the concept of frequency distribution shift for modeling temporal distribution.
- We propose a novel method called FEDNet and provide its theoretical insights. FEDNet decomposes time series into time-deterministic and time-stochastic parts separating the temporal distribution by frequency information. We propose that domain feature contrastive learning simplifies the constraints on domain-invariant learning and accelerates convergence.
- Experiments on several datasets demonstrate the state-of-the-art performance of FEDNet and the effectiveness of frequency information. Moreover, we found the stability of the frequency component to time-distribution shifts and verified that the essence of frequency enhancement originate from the orthogonality of frequencies.

## 2 RELATED WORK

**Domain Generalization.** Domain generalization (Wang et al., 2022a) is a difficult challenge where the goal is to obtain robust models from multiple domains that eventually generalize to unseen domains and are inaccessible to the training process. Existing domain generalization schemes are

divided into three main levels: data augmentation, feature representation, and learning strategies. In data augmentation, Mixup(Zhang, 2017) data from different domains is often used in data processing and some methods try to augment data from a frequency domain perspective(Demirel & Holz, 2024; Xu et al., 2021) verified the robustness of frequency information. In representation learning, a common strategy is to extract domain-invariant features across multiple source domains with IRM paradigm(Arjovsky et al., 2019; Krueger et al., 2021), domain adversarial learning (Ganin et al., 2016) and disentanglement-based method(Ilse et al., 2020) serve the same purpose. For learning strategies, one of the most famous approaches is distributional robust optimization, such methods (Sagawa* et al., 2020; Kuhn et al., 2019) dynamically penalize the weighted average loss of all domain distribution sets, thus minimizing the generalization expectation of the unseen distributions theoretically. Some methods attempt to decompose features to capture more stable domain-invariant information. StableNet(Zhang et al., 2021), inspired by causal mechanisms, introduces a novel nonlinear feature decomposition correlation technique for capturing domain invariant information. Unfortunately, due to the widespread temporal distribution shift in time series, these above methods cannot be fully applicable to domain generalization for time series classification.

**OOD Time Series Classification.** The research on OOD time series classification is very limited. AdaRNN (Du et al., 2021) proposes a temporal distribution characterization and matching module that divides the time series segments into finite large-distance groups in data distribution to extract invariant features. Diversify (Lu et al., 2022) identifies the latent distribution domains for the "worst-case scenarios" through adversarial training, and then reduces the gaps between time series segments in latent domains. GILE (Qian et al., 2021) is a disentanglement method designed to extract domain-invariant and domain-specific representations through variational inference (Kingma & Welling, 2013). However, these methods do not take into account both domain and temporal shifts, resulting in suboptimal performance.

**Frequency-Based Time Series Representation.** Frequency domain information has been widely used in recent years. In time series, many methods get better results by introducing frequency domain information. For example, FEDformer (Zhou et al., 2022) improves the computational efficiency and performance of long-short forecasting with the help of frequency attention, FreTS (Yi et al., 2023) is MLP-based architecture with frequency spectrum, and TF-C (Zhang et al., 2022) proposes time-frequency consistency to do time series pre-training and transfer learning. These works show that frequency domain information can improve the generalization performance of the model.

## 3 PRELIMINARY

**Definition 1. Time Series Data.** A multi-domain time series dataset can be defined as $\mathcal{E} = \{\mathcal{X}, \mathcal{Y}, \mathcal{D}\}$, where $\mathcal{X}, \mathcal{Y}, \mathcal{D}$ represent time series segments pre-processed raw data samples by sliding window, category labels and domain labels respectively. Specifically, $\mathcal{X} = \{x_i\}_{i=1}^N$ indicates $N$ time series segments in total, where $x_i \in \mathcal{X} \subset \mathbb{R}^{L \times C}$ is the $C$-channel instance with $L$ timestamps and $y_i \in \mathcal{Y} = \{c_1, \cdots, c_{N_y}\}$ is its label, where $N_y$ is the number of the category labels. In addition, the dataset is usually divided into multiple domains environments $\mathcal{D} = \{d_1, d_2, \cdots, d_{N_d}\}$, where $N_d$ is the number of the domain labels. Each domain consists of a set of samples $D_k = \{(x_i, y_i)\}_{i=1}^{N_k}$, $N_k$ denotes the number of samples in the $k$-th domain $D_k$.

**Definition 2. Domain Distribution Shift.** Given a multi-domain time series dataset $\mathcal{E} = \{\mathcal{X}, \mathcal{Y}, \mathcal{D}\}$, these domains usually belong to different data distributions. We define the joint distribution of data and label as $\mathbb{P}(x, y)$. Domain distribution shift can be described as

$$\forall D_i \neq D_j, \mathbb{P}^{D_i}(x) \neq \mathbb{P}^{D_j}(x), \mathbb{P}^{D_i}(y|x) = \mathbb{P}^{D_j}(y|x) \rightarrow \mathbb{P}^{D_i}(x, y) \neq \mathbb{P}^{D_j}(x, y). \quad (1)$$

**Definition 3. Temporal Distribution Shift.** Given a group of time series data in $k$-th domain $D_k = \{(x_i, y_i)\}_{i=1}^{N_k}$, suppose these segments come from $\{D_k^t\}_{t=1}^T$ peroids, $T$ is unpredictable. Temporal distribution shift exists between different periods caused by non-stationary or equipment factors.

$$\exists i, j \in [1, T], \mathbb{P}^{D_k^i}(x) \neq \mathbb{P}^{D_k^j}(x) \neq \mathbb{P}^{D_k}(x) \xrightarrow{\mathbb{P}^{D_k}(y|x)} \mathbb{P}^{D_k^i}(x, y) \neq \mathbb{P}^{D_k^j}(x, y) \neq \mathbb{P}^{D_k}(x, y). \quad (2)$$

**Definition 4. Frequency Distribution Shift.** The time series can be transformed to the frequency domain by the Fourier transform, and we can use the frequency domain statistics to represent the

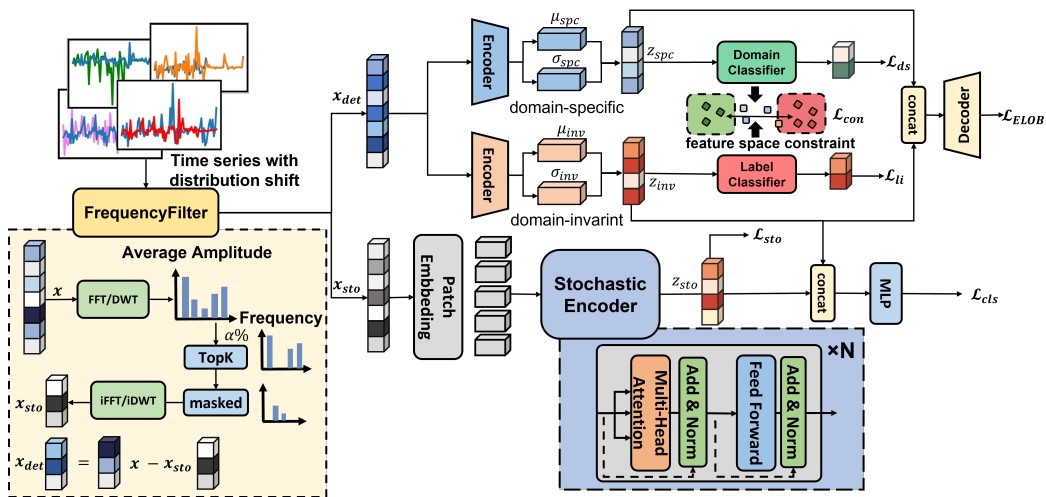

Figure 2: Architecture of the proposed FEDNet.

distribution in the same way. Here, we propose the concept of frequency distribution shift to describe the marginal probability distribution shift of segment in frequency domain.

$$\exists i, j \in [1, T], \mathbb{P}_F^{D_k^i}(x) \neq \mathbb{P}_F^{D_k^j}(x) \neq \mathbb{P}_F^{D_k}(x) \xrightarrow{\mathbb{P}_F^{D_k}(y|x)} \mathbb{P}_F^{D_k^i}(x, y) \neq \mathbb{P}_F^{D_k^j}(x, y) \neq \mathbb{P}_F^{D_k}(x, y). \quad (3)$$

## 4 PROBLEM FORMULATION

Given a multi-domain time series dataset $\mathcal{E} = \{\mathcal{X}, \mathcal{Y}, \mathcal{D}\}$, we follow the cross-domain rule to divide the dataset into training dataset $\mathcal{E}_{tr} = \{\mathcal{X}^{tr}, \mathcal{Y}^{tr}, \mathcal{D}^{tr}\}$ and unseen testing dataset $\mathcal{E}_{te} = \{\mathcal{X}^{te}, \mathcal{Y}^{te}, \mathcal{D}^{te}\}$. Specifically, in our paper, three conditions are imposed on the dataset: (1) OOD environments: $\mathbb{P}^{tr}(x, y) \neq \mathbb{P}^{te}(x, y)$. (2) Domain distribution shift: $\mathbb{P}^{D_i}(x, y) \neq \mathbb{P}^{D_j}(x, y), \forall D_i \neq D_j$. (3) Temporal distribution shift: $\mathbb{P}^{D_k^i}(x, y) \neq \mathbb{P}^{D_k^j}(x, y) \neq \mathbb{P}^{D_k}(x, y), \exists i \neq j \in [1, T]$. Our goal is to learn an optimal model $f_\theta^* : \mathcal{X} \to \mathcal{Y}$ under the above conditions, the model aims to minimize both domain distribution shift and temporal distribution shift to generalize well to OOD testing data:

$$f_\theta^* = \arg\min_{f_\theta} \mathbb{E}_{(x,y) \sim \mathcal{E}_{te}}[\ell(f_\theta(x), y)], \quad (4)$$

where $\mathbb{E}$ denotes expectation and $\ell(\cdot, \cdot)$ denotes loss function.

## 5 METHOD

In this section, we introduce the detailed pipeline of FEDNet, which is shown in Figure 2. FEDNet decomposes time series into a time-deterministic block and a time-stochastic block to extract invariant features for OOD time series classification. The time-deterministic block is less affected by time, its feature is mainly affected by the domain distribution shift, which can be prompted with domain labels. The time-stochastic block is more susceptible to the temporal distribution shift. We convert data into finite patches to simulate the moving average weighting process of the stochastic component. FEDNet decomposes the time series components theoretically to model the effects of domain distribution shift and temporal distribution shift for OOD time-series classification. Ultimately, we theoretically analyze previous IRM-based methods and FEDNet to demonstrate that FEDNet is better suited for OOD time series classification.

## 5.1 FREQUENCY FILTER

**Lemma 1 (Wold's Theorem).** *Given weak-sense stationarity time series $x_t$ can be formally decomposed as the sum of two time series, one deterministic and one stochastic.*

$$x_t = \eta_t + \sum_{j=0}^{\infty} b_j \varepsilon_{t-j}, \tag{5}$$

where $\eta_t$ denotes the deterministic component and $\varepsilon_t$ is the stochastic component that is input to an infinite vector of moving average weights $\{b_j\}$. Suppose Time series data in the same period can be regard as local weak-sense stationarity, The main benefit of weak-sense stationarity is that any time series can be put into the context Hilbert Space. It means any time series can be decomposed by a set of orthogonal increments in the space. Bochner's theorem (Loomis, 2013) ensures that there exists a group of Fourier-type complex exponential function $\{e^{-2\pi i \xi t}\}$ componets to generate $x_t$.

$$x_t = \int e^{-2\pi i \xi t} d\omega_\xi \tag{6}$$

where $d\omega_\xi$ is the measure weight with the $\xi$-th frequency wave. Due to the orthogonality of frequencies, it inspired us to explore whether the deterministic and stochastic components are dominated by specific frequencies, even in different periods and domains (detail study in Figure 3).

We first precompute FFT or DWT of each data in training set, calculate the averaged amplitude of each position in spectrum $\mathcal{S} = \{0, \cdots, [L/2]\}$ and sort them by corresponding amplitude. Then we use $\alpha$ to obtain top frequency positions $\mathcal{S}_a$ for decompostion in train and inference. $\mathcal{S}_\alpha$ can represent active stable frequency positions among different periods in multi-domain dataset, which has time-deterministic property (less affected by time) intuitively. Here we use $\mathcal{F}$ to denote frequency tramsform operation, $\mathcal{F}^{-1}$ denotes the inverse operation.

$$x_{sto} = \mathcal{F}^{-1} \left( \text{Mask} \left[ \mathcal{S}_\alpha \right] \cdot \mathcal{F}(x) \right), x_{det} = x - x_{sto} \tag{7}$$

where $x_{det}$ denotes time-deterministic features and $x_{sto}$ denotes time-stochastic features. Mask operation sets the amplitude of specific position components into zero.

## 5.2 TIME-DETERMINISTIC BLOCK

We use disentangled representation learning methods based on the paradigm of variational inference to deal with time-deterministic features $x_{det}$.

We construct the domain-invariant and domain-specific probabilistic encoders to decompose $x_{det}$ feature space and a decoder merges the two parts of the features to reconstruct origin features. In parallel, it is supervised by constraint loss from both domain labels and category labels.

**Domain Disentangled Probabilistic Encoder.** Here we design domain-invariant and domain-specific probabilistic encoders denoted as $q_\phi(z_{inv}|x_{det})$ and $q_{\phi_d}(z_{spc}|x_{det})$ to represent the corresponding feature space seperately. The two encoders are used to learn the two feature space statistics $(\mu_{inv}, \sigma_{inv})$ and $(\mu_{spc}, \sigma_{spc})$. We use the reparametrization trick to resample $z_{spc}$ and $z_{inv}$ from standard normal distribution statistics.

$$q_\phi(z_{inv}|x_{det}) = \mathcal{N}\left(z_{inv} \mid \mu_{inv}, \sigma_{inv}; \phi\right), \tag{8}$$

$$q_{\phi_d}(z_{spc}|x_{det}) = \mathcal{N}\left(z_{spc} \mid \mu_{spc}, \sigma_{spc}; \phi_d\right), \tag{9}$$

where $\phi$ and $\phi_d$ denote the parameters of the domain-invariant encoder and domain-specific encoder respectively.

In order to adapt to prior distribution shift in different domains, we also set the prior hypothesis space $p(z_{inv})$ and $p(z_{spc})$ to data-driven Gaussian distributions by domain label and category label as well.

Finally, a probabilistic decoder $p_\theta(x_{inv}|z_{inv}z_{spc})$ is used to feed $z_{spc}$ and $z_{inv}$ to reconstruct original information, where $\theta$ are learnable parameters of the decoder. The probabilistic encoders and

decoder are learned with the following objective $\mathcal{L}_{ELBO}$:

$$\begin{aligned}
\mathcal{L}_{ELBO} = \ &\mathbb{E}_{q_\phi(z_{inv}|x_{det}),q_{\phi_d}(z_{spc}|x_{det})} \left[\log p_\theta(x_{inv}|z_{inv},z_{spc})\right] \\
&- D_{\mathrm{KL}}\left(q_\phi(z_{inv} \mid x_{det})\|p(z_{inv})\right) \\
&- D_{\mathrm{KL}}\left(q_{\phi_d}(z_{spc} \mid x_{det})\|p(z_{spc})\right),
\end{aligned} \tag{10}$$

where the first term represents the reconstruction loss to $x_{det}$, and the next two terms are the prior matching terms from $z_{inv}$ and $z_{spc}$ by KL-divergence regularization.

**Constraint Loss.** The generation process under unsupervised signals proved to be unreliable (Locatello et al., 2019), so we use both domain labels and category labels for constraint loss. It encourages the domain-invariant space close to the label space, and the domain-specific space close to the domain space. We provide two schemes for constraint loss, one is the common feature cross-prediction loss (Qian et al., 2021), and the other is our proposed simplification of the constraint process using supervised contrastive loss (Khosla et al., 2020), which achieves the same goal by reducing the similarity of two features (Shi et al., 2024).

(1) feature cross-prediction loss.

$$\mathcal{L}_{con1} = \mathcal{L}_{li} + \mathcal{L}_{ds} - \mathcal{L}_{di} - \mathcal{L}_{ls}. \tag{11}$$

where $\mathcal{L}_{li}$, $\mathcal{L}_{ls}$ denotes $z_{inv}$, $z_{spc}$ with label loss,$\mathcal{L}_{di}$,$\mathcal{L}_{ds}$ denotes $z_{inv}$,$z_{spc}$ with domain class loss.

(2) domain-invirant contrastive loss. We use $z_{inv}$ and $z_{spc}$ of each batch of data itself as negative sample pairs, which not only increases the gap between the domain-invariant and domain-specific parts, but also reduces the individual differences between the positive samples.

$$\mathcal{L}_{con2} = -\sum_{i \in I} \log \frac{\exp(z_i, z^+/\tau)}{\sum_{k=0}^{K} \exp(z_i, z_k/\tau)}. \tag{12}$$

where $I$ is the origin index set of the $z_{inv}$ and $z_{spc}$ and we concat them to $\{z_k\}_{k=0}^{K}$ features, and we made $\left(z_i, \{z^+\}_{k=0}^{K^+}, \{z^-\}_{k=0}^{K^-}\right)$, while $z^+$ represents the same feature space and the $z^-$ represents the opposite space, $\tau$ denotes to scalar temperature.

## 5.3 TIME-STOCHASTIC BLOCK

The time-stochastic block is constructed for time-stochastic features $x_{sto}$. It can be considered as stochastic components with moving average weighting, so we try to capture the local time-variant dynamic features with the help of patches (Nie et al., 2023), recent studies have shown that domain-invariant features in collaboration with other classification-related features help to improve the robustness of the model (Yu et al., 2024).

**Patch Embedding.** We divide the input time series into patches and set the patch length $P$ and patch stride $S$, to divide the $L$-length sequence $x_{sto}$ into $M$ patches $x_{patch} \in \mathbb{R}^{P \times M}$, where $M = \lfloor \frac{L-P}{S} \rfloor + 2$. We map the patches through a linear layer $W_p \in \mathbb{R}^{D \times P}$ to the transformer space with a learnable position encoding $W_{pos} \in \mathbb{R}^{D \times M}$ to get the final patch embedding $x_h \in \mathbb{R}^{D \times M}$.

$$x_h = W_p x_{patch} + W_{pos}. \tag{13}$$

**Stochastic Encoder.** We use self-attention (Vaswani et al., 2017) in the transformer encoder to model the dynamically varying weighting of the time dimension and to avoid mixing effects between different variables, we follow a channel-independent design, taking each patch as input and final splicing the outputs, and then an MLP classifier that constrains the classification loss in that part. We keep stacking multi-head attention and feed-forward network with residual connections (He et al., 2016) and layer normal (Ba et al., 2016).

$$\begin{aligned}
Q_h &= x_h^T W_h^Q, K_h = x_h^T W_h^K, V_h = x_h^T W_h^V, \\
O_h &= Attention(Q_h, K_h, V_h) = softmax(\frac{Q_h K_h^T}{\sqrt{d_k}})V_h,
\end{aligned} \tag{14}$$

where $O_h$ is the output from the multi-head attention layer and input to the feed-forward layer.

Finally, an MLP is utilized for label prediction, transferring the output of the encoder into the final latent feature $z_{std} \in \mathbb{R}^D$, we use *cross-entrpy* (CE) loss $L_{sto}$ for label classification.

## 5.4 MODEL SUMMARY AND THEORETICAL INSIGHTS

**Proposition 5.1.** *Assuming time series data $x_t \in \mathbb{R}^{L \times C}$ through frequency decomposition into time-deterministic components $A_{det} = \{a_i\} \in \mathbb{R}^{K \times C}$ and time-stochastic components $A_{sto} = \{a_i\} \in \mathbb{R}^{(L/2-K) \times C}$, the invariant minimization objective formula suitable for OOD time series classification is as follows:*

$$\min_{\boldsymbol{w}} \mathcal{L}(\boldsymbol{w}) := \sum_{e \in \mathcal{E}_{tr}} \mathcal{R}^e(\Phi_{det}(A_{det})) + \lambda_{det}\mathcal{P}(\Phi_{det}(A_{det})) + \lambda_{sto}\mathcal{J}_{\phi_{\boldsymbol{w}}}(A_{sto}) \qquad (15)$$

*where $\mathcal{R}^e$ denotes the risk in domain environment $e$, $\Phi_{det}$ denotes invariant feature extractor. $\mathcal{P}$ is the regularization for invariant feature, $\mathcal{J}_{\phi_{\boldsymbol{w}}}$ capture auxiliary features avoiding information lose.*

FEDNet decoupled the feature into time-deterministic and time-stochastic features to solve domain distribution shift and temporal distribution shift respectively and concat $z_{inv}$, $z_{sto}$ from two parts for final classification $\mathcal{L}_{cls}$. The model's total loss $\mathcal{L}$ consists of three parts:

$$\begin{aligned} \mathcal{L} &= \lambda_{det}\mathcal{L}_{det} + \lambda_{sto}\mathcal{L}_{sto} + \mathcal{L}_{cls}, \\ \mathcal{L}_{det} &= \mathcal{L}_{ELBO} + \mathcal{L}_{con}, \end{aligned} \qquad (16)$$

where $\mathcal{L}_{det}$ denotes time-deterministic loss, $\mathcal{L}_{sto}$ denotes time-stochastic loss, $\mathcal{L}_{cls}$ denotes final classification loss, $\lambda_{det}$ and $\lambda_{sto}$ are hyperparameter weights for two parts.

**Proposition 5.2 (Frequency Perspective Risk Bound on Unseen Time series Domain).** *Let $\mathcal{H}$ be a hypothesis space built from a set of source time series domains $D = \{D_i\}_{i=1}^{N_d}$. Suppose $q > 0$ is a constant, for any unseen time series domain $D_U$ from the convex hull $\Lambda_D$, we have its closest element $D_{\bar{U}}$ related to source domains in $\Lambda_D$, i.e.,$D_{\bar{U}} = \arg\min_{\pi_1,...,\pi_{N_d}} \beta_q(D_{\bar{U}} \| \sum_{i=1}^{N_d} \pi_i D_i)$. Then the risk of $D_U$ on any label function $h \in \mathcal{H}$ is,*

$$R_{D_U}[h] \leq \frac{1}{2} d_{D_U}(h) + \rho \cdot \left[e_{D_{\bar{U}}}(h)\right]^{1-\frac{1}{q}}, \qquad (17)$$

*where $\rho = 2^{\frac{q-1}{q} \sup_{i,j \in [N_d]} RD_q(D_i \| D_j)}$, $d_D(h)$ and $e_D(h)$ are ideal and empirical risk of domain $D$,*

$$RD_q(D_i \| D_j) = \frac{1}{q-1} \log \int \left[P_F(D_i)\right]^q \left[P_F(D_j)\right]^{1-q} da = \frac{1}{q-1} \log \left[\frac{\mu_j^{2(q-1)}}{\mu_i^{2q}} \cdot \frac{\sqrt{\pi}^{2n+1} n!!}{2^{2n+1}\sqrt{\gamma}^{n+1}}\right] \qquad (18)$$

*where $\mu_i = E(D_i) = \prod_{k=1}^n \sqrt{\frac{\pi}{2}}\sigma_k, \mu_j = E(D_j) = \prod_{k=1}^n \sqrt{\frac{\pi}{2}}\tau_k, \gamma = \sum_{k=1}^n \frac{q}{2\sigma_k^2} + \frac{1-q}{2\tau_k^2}, \{\sigma_k\}_{k=1}^n$ and $\{\tau_k\}_{k=1}^n$ denote frequency scale parameters in $D_i, D_j$, $n$ represents the number of components.*

The individual $\sigma_k$ is a linear unbiased estimate of $E(a_k)$, The first term can be regarded as constant and bounded. The second term perfectly aligns with our motivation for decoupling the frequency domain. By keeping top $E(a_k)$, we can reduce $n$ while making $\gamma$ within a controllable range to decrease $\rho$ and whole generalization bound. proofs for two proposition are provided in Appendix A

## 6 EXPERIMENT

**Datasets.** We conduct serveral datasets to evaluate the performance and efficiency of FEDNet. We used the synthetic dataset Spurious Fourier and the real datasets HHAR (Gagnon-Audet et al., 2022) provided by WOODS (Gagnon-Audet et al., 2022), three open-source datasets UCIHAR (Anguita et al., 2012), UniMiB-SHAR (Micucci et al., 2017), and Opportunity (Chavarriaga et al., 2013) used in GILE(Qian et al., 2021), as well as DSADS (Barshan & Yüksek, 2014), PAMAP (Reiss & Stricker, 2012),processed according to the domain division strategy provided by Diversify (Lu et al., 2022). The details of these datasets are listed in Appendix B.

**Baselines.** We evaluate the proposed FEDNet with various significant baselines, which can be divided into three types. We provide detailed hyperparameter implementation, dataset settings and comparing rules in Appendix D.

- **General time series methods.** We compare with the mainstream time series models in recent years such as PatchTST (Nie et al., 2023), and FreTS (Yi et al., 2023).
- **General OOD methods.** We choose some important baselines for OOD generalization from other research domains, GroupDRO (Sagawa* et al., 2020), ANDMask (Parascandolo et al., 2021), and VREx (Krueger et al., 2021).
- **OOD time series methods.** Research on OOD generalization for time series classification is limited, and we select three important works, i.e., GILE (Qian et al., 2021), AdaRNN (Du et al., 2021), and Diversify (Lu et al., 2022).

## 6.1 Performance Comparison

Table 1: Accuracy on cross-person generalization. "Target" represents the unseen test domain. Spurious Fourier is a synthetic dataset, with only {d=10%} used as the test domain, while the remains are all real-world datasets. FEDNet$_f$ uses FFT and FEDNet$_w$ uses othogonal function DWT.

| Dataset | Target | VREx | GroupDRO | ANDMask | FreTS | PatchTST | GILE | AdaRNN | Diversify | FEDNet$_f$ | FEDNet$_w$ |
|---|---|---|---|---|---|---|---|---|---|---|---|
| Spurious Fourier | d=10% | 48.19 | 48.66 | 11.16 | 49.09 | 11.03 | 15.90 | 50.12 | 15.37 | **74.56** | 33.34 |
| HHAR | 0 | 89.60 | 88.49 | 91.48 | 93.12 | 79.46 | 94.51 | 72.77 | 91.48 | 96.97 | **97.76** |
| | 1 | 91.00 | 88.81 | 92.65 | 93.76 | 79.90 | 96.94 | 75.67 | 92.65 | **98.26** | 97.27 |
| | 2 | 84.58 | 83.99 | 86.81 | 89.93 | 75.34 | 88.21 | 72.77 | 84.79 | 90.76 | **93.04** |
| | 3 | 65.55 | 65.26 | 61.60 | 62.76 | 39.68 | 63.34 | 50.06 | 54.32 | 63.17 | **67.87** |
| | 4 | 54.05 | 55.74 | 51.69 | 55.40 | 42.22 | 43.92 | 36.49 | 45.14 | **57.09** | 51.35 |
| UCIHAR | 0 | 89.34 | 89.34 | **98.56** | 81.55 | 78.96 | 83.07 | 80.20 | 87.03 | 94.81 | 78.38 |
| | 1 | 66.23 | 57.28 | 69.21 | 56.62 | 73.50 | 75.62 | 76.24 | 76.49 | **80.13** | 72.84 |
| | 2 | 97.65 | 96.77 | 97.94 | 92.30 | 70.67 | 86.19 | 86.45 | 90.91 | **97.95** | 90.61 |
| | 3 | 83.28 | 84.54 | 88.33 | 79.49 | 78.86 | 91.25 | 87.50 | 89.27 | **91.79** | 80.12 |
| | 4 | 70.86 | 66.23 | 89.74 | 89.07 | 80.46 | 85.62 | 87.81 | 92.38 | **98.34** | 90.06 |
| UniMiB-SHAR | 1 | 55.99 | 58.33 | 57.55 | 41.14 | 49.21 | 47.39 | 46.88 | 50.26 | 55.98 | **64.58** |
| | 2 | 57.80 | 59.69 | 59.35 | 36.02 | 68.78 | 46.40 | 26.76 | 42.20 | 70.15 | **73.58** |
| | 3 | 63.16 | 63.82 | 64.14 | 60.85 | 71.38 | 62.18 | 46.05 | 60.20 | 71.71 | **75.65** |
| | 5 | 41.28 | 42.62 | 40.94 | 38.25 | 36.91 | 38.43 | 35.57 | **44.30** | 40.26 | 44.29 |
| Opportunity | S1 | 53.73 | 59.62 | 77.85 | 81.87 | 52.00 | 84.02 | 80.64 | 82.23 | **84.86** | 83.82 |
| | S2 | 37.17 | 55.79 | 78.57 | 81.21 | 66.62 | 81.39 | 78.97 | 79.96 | 81.45 | **81.53** |
| | S3 | 36.77 | 56.31 | 74.67 | 75.94 | 46.76 | 77.91 | 76.36 | 76.79 | **79.11** | 76.32 |
| | S4 | 46.41 | 58.15 | 78.32 | 77.58 | 52.48 | 80.91 | 78.85 | 80.74 | **81.77** | 80.60 |
| EMG | 0 | 70.25 | 70.66 | 71.60 | 71.77 | 33.92 | 62.96 | 54.11 | 67.55 | **73.00** | 64.55 |
| | 1 | 85.50 | 83.08 | 82.52 | 80.15 | 36.82 | 68.02 | 57.44 | 81.09 | **87.10** | 59.59 |
| | 2 | 73.62 | 77.03 | 76.91 | 74.88 | 22.66 | 66.02 | 57.83 | 74.64 | **79.66** | 77.51 |
| | 3 | 77.14 | 78.62 | 77.50 | 77.96 | 36.62 | 69.99 | 53.87 | 77.32 | 77.43 | **79.85** |
| DSADS | 0 | 80.26 | 84.69 | 82.50 | 80.26 | 82.24 | 89.64 | 83.11 | 77.19 | **92.80** | 92.41 |
| | 1 | 76.54 | 78.03 | 73.42 | 70.13 | 74.07 | 78.20 | 79.78 | 77.28 | **84.86** | 83.64 |
| | 2 | 86.40 | 85.96 | 83.03 | 84.29 | 82.67 | 86.75 | 83.46 | 85.22 | **93.24** | 90.65 |
| | 3 | 74.61 | 74.39 | 78.46 | 73.46 | 78.85 | 79.56 | 70.35 | 71.80 | **87.71** | 80.52 |
| PAMAP | 0 | 62.88 | 61.75 | 61.84 | 55.22 | 60.40 | 65.01 | 63.30 | 61.98 | 64.94 | **67.48** |
| | 1 | 54.88 | 52.00 | 53.04 | 60.41 | 66.36 | 51.46 | 54.24 | 54.38 | **68.16** | 67.08 |
| | 2 | 22.68 | 25.69 | 28.02 | 34.98 | **50.06** | 25.23 | 23.35 | 24.32 | 34.39 | 35.27 |
| | 3 | 62.10 | 65.22 | 67.86 | **68.69** | 63.39 | 68.06 | 61.04 | 57.79 | 67.13 | 68.55 |

We evaluate the generalizability of our proposed FEDNet by comparing its performance with baselines on these publicly accessible datasets in Table 16. We choose accuracy as the main evaluation metric. For our experiments, we select one domain as the "target domain" for testing, while the remaining domains serve as the "source domains" for training.

**Cross-person generalization.** The frequency-domain separation design of FEDNet achieves the best performance across multiple datasets compared to other methods. General time series methods use ERM (Empirical Risk Minimization) as the optimization objective, resulting in greater performance fluctuations across different domains. These methods are significantly affected by domain shifts and segmentation. FreTS, which also extracts frequency-domain information, performs better than PatchTST in some domains, indicating that frequency information is robust, by providing a global perspective on time series. Additionally, FEDNet outperforms three different representa-

tive types of General OOD that fail to consider the impact of temporal distribution shifts within the time series window on marginal probabilities. Methods like AdaRNN and Diversify, which are carefully designed to account for temporal shifts through reweighting or relabeling, directly extract information from the pure time domain. Essentially, these are data augmentation techniques and are inevitably affected by sample noise when achieving domain generalization. We also provide results for other metrics, as detailed in the Appendix C.9.

**Cross-position generalization** We did more diffcult experiments explore model generalization focus on General OOD and time series OOD methods. The details of the dataset-specific processing are described in the Appendix B.2. From Table 2, We can see our method is better and stable than other OOD method, it shows frequency decompostion can hold more generalize sitiuations.

Table 2: Accuracy on cross-position generalization for DSADS.

| Dataset | Target | VREx | GroupDRO | ANDMask | IB-IRM | IRM | GILE | AdaRNN | Diversify | FEDNet$_f$ |
|---------|--------|------|----------|---------|--------|-----|------|--------|-----------|-----------|
| DSADS | 0 | 27.12 | 29.99 | 26.64 | 29.82 | 25.30 | 38.85 | 38.19 | **47.70** | 40.54 |
| | 1 | 20.33 | 23.86 | 24.62 | 23.17 | 20.18 | 21.45 | 29.04 | 32.90 | **36.05** |
| | 2 | 27.17 | 38.20 | 33.22 | 35.72 | 25.62 | 38.33 | 32.72 | **44.50** | 36.25 |
| | 3 | 24.78 | 24.00 | 26.95 | 19.96 | 27.34 | 20.09 | 24.61 | 31.60 | **33.93** |
| | 4 | 17.24 | 26.01 | 20.18 | 22.15 | 18.25 | 23.14 | 19.50 | 30.40 | **33.04** |

## 6.2 MODEL ANALYSIS

**Frequency Decomposition Study.** We add a fully connected linear layer behind two components separately after the Frequency Filter to analyze the linear weight changes. We chose different domains as the target domains for training and computed the coefficient of variation of the weights of the two linear layers since the linear weights can reflect the dependence between the time series points. Formally, the coefficient of variation is defined as follows:

$$\text{coefficient of variation} = \frac{\sigma_{weights}}{\mu_{weights}}, \tag{19}$$

where $\sigma_{weight}$ and $\mu_{weights}$ represent the standard deviation and mean of the linear layer weights.

As shown in Figure 3, we find that the coefficient of variation of the deterministic linear weights during training in different domains is always lower than that in stochastic part. It shows that the deterministic component could vary less among different periods, controlled by specific frequency.

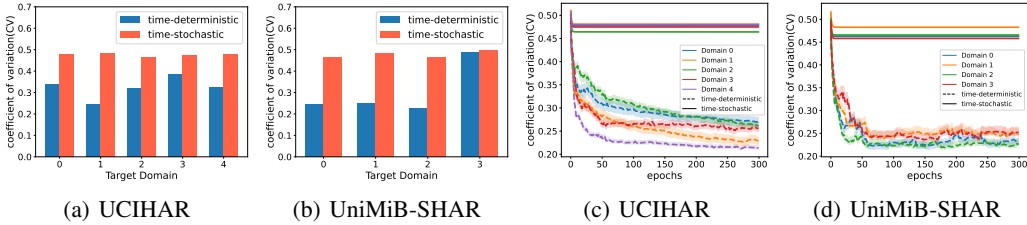

| (a) UCIHAR | (b) UniMiB-SHAR | (c) UCIHAR | (d) UniMiB-SHAR |

Figure 3: Coefficient of variation of time-deterministic and time-stochastic linear weights in model.

**Invirant Feature Study.** We use $\mathcal{A}$-distance (Schölkopf et al., 2007) to measure the domain discrepancy of invariant features obtained by different methods. It can be approximated as $d = 2(1 - 2\sigma_{\mathcal{A}})$, where $\sigma_{\mathcal{A}}$ is the risk of a binary classifier distinguishing features between source and target domains. Figure 4 shows that our method consistently outperforms other approaches, and using contrastive learning achieves better results compared to the standard cross label loss. The smaller the indicator, the more invariant features are.

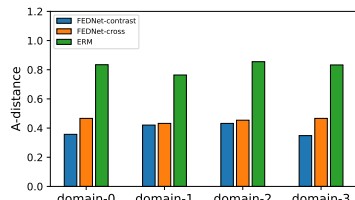

Figure 4: $\mathcal{A}$-distance on invariant features with EMG dataset.

**Ablation Study.** we conduct ablations on FEDNet with several degenerate variants to analyze model components: (1) w/o $\mathcal{L}_{con}$: we remove the constraint loss in Time-Determistic Block. (2) w/o $\mathcal{L}_{det}$: we remove the time-determistic block, only use $x_{sto}$ for classification. (3) w/o $\mathcal{L}_{sto}$: we remove the time-stochastic block, only use $x_{det}$ for classification. We present some of the results in Figure 5, with detailed results available in Appendix C.1. We observe that our proposed Frequency Filter is effective in disentangling distributional shifts, and it obtains stable time-invariant components that eliminate the effect of temporal shifts:

(1) w/o $\mathcal{L}_{con}$: The time-deterministic block essentially degenerates into a dimensionality reduction module for $x_{det}$ without considering domain distribution shift. The final results are not much different from the general time series methods.

(2) w/o $\mathcal{L}_{det}$: Lack of $x_{det}$ leads to significant performance degradation. At the same time, we find that time-stochastic features still have some classification potential in some domains.

(3) w/o $\mathcal{L}_{sto}$: without $x_{sto}$, it also has a small effect on model performance. It shows that dynamic changes in the time series can improve the robustness of the model in unseen domains to some degree.

Figure 5: Ablation study. The Y-axis shows average accuracy.

(4) cross vs contrast: contrastive learning obviously outperforms cross loss.

**Empirical Domain Divergence Study.** As Proposition 5.2 shows, $\gamma$ can be seen as a meaningful metric to measure the volatility of time series in frequency. We adopted it to be an empirical absolute value $\hat{\gamma} = \sum_{k=1}^{n} q |\frac{1}{2\sigma_k^2} - \frac{1}{2\tau_k^2}|$ avoid redundant frequency to estimate the domain divergence. Figure 3 (d) changes of domain divergence in the dataset consistents with the IRM gains experiments.

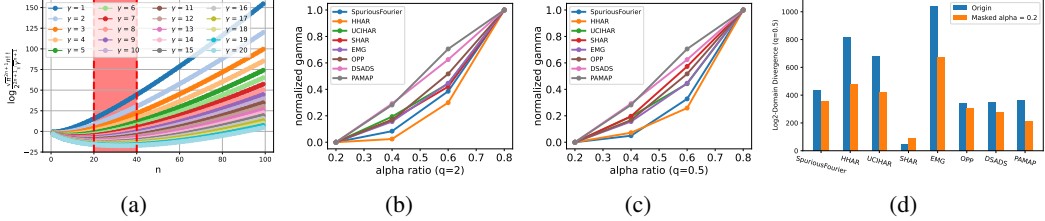

(a)        (b)        (c)        (d)

Figure 6: (a) denotes the second log value with the number of frequencies $n$ and $\gamma$. It shows that we could get the lowest upper bound on generalization when $\alpha \in [0.2, 0.4]$, $\gamma \in [1, 20]$ is usually the datasets' range after reducing frequencies. (b) and (c) denote the trend of $\hat{\gamma}$ when $q = 2, 1/2$, shows that reducing the number of frequencies reduces the volatility of time series. (d) denotes dataset domain divergence with mask $\alpha$=0.2. It demonstrates that keep $\alpha$ ratio high amplitude frequency reduces domain divergence. The anomaly in SHAR stems from denominator $\hat{\gamma}(< 0.01)$ very small.

## 7 CONCLUSION

In this paper, we focus on OOD time series classification and propose a novel method called FEDNet. Our method incorporates frequency information as a prior and utilizes a decomposition framework to separate time series into time-deterministic components and time-stochastic components. We address both domain distribution shift and temporal distribution shift to extract invariant features for domain generalization. Extensive experiments demonstrate that FEDNet achieves superior performance and effectively exploits frequency information for OOD time series classification.

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

# A  THEORETICAL INSIGHTS

## A.1  BACKGROUND

Here, we first provide some background knowledge to illustrate the widespread occurrence of marginal probability shift in time series.

**OOD Generalization Problem.** Given datasets $D_e = \{(x_i^e, y_i^e)\}_{i=1}^{n_e}$ collected from multiple training environments $e \in \mathcal{E}_{tr}$. Each dataset $D_e$ contains a group of examples according to a certain probability distribution $x \sim \mathbb{P}^e(x)$. We hold on the basic covariate shift assumption (Shimodaira, 2000) that the optimal label function $h^* \in \mathcal{H}$ representation space or conditional probability distribution $\mathbb{P}^e(y|x)$ are the same within different environments while $\mathbb{P}^e(x) \neq \mathbb{P}^{e'}(x), \forall e, e' \in \mathcal{E}_{tr}$.

We define the prediction risk of model $f$ under $e$ environment as $R^e(f) := \mathbb{E}^e[\ell(f(x^e), y^e)]$. Our goal is to obtain a optimal model $f_\theta^*$ that can generalize to unseen domain distribution $\mathcal{E}_{unseen} = \mathcal{E}_{all}/\mathcal{E}_{tr}$.

$$f_\theta^* = \arg \min_{f_\theta} \mathbb{E}_{\mathcal{E}_{unseen}}[\ell(f_\theta(x), y)]. \tag{20}$$

where $\mathbb{E}$ denotes expectation and $\ell(\cdot, \cdot)$ denotes loss function.

**OOD Time Series Problem.** The basic goal of the Time Series Out of Distribution Problem is the same as that of the Out of Distribution Generalization Problem. However, the major difference is that there exists temporal distribution shift in each domain, which causes the data distribution broken the i.i.d assumption with $D_e(x, y)$, i.e. $\mathbb{P}^{D_k^i}(x, y) \neq \mathbb{P}^{D_k^j}(x, y) \neq \mathbb{P}^{D_k}(x, y), \exists i \neq j \in [1, T]$.

**IRM.** Existing IRM methods encourage to elicit an invariant predictor $f(:, w)$ which is a composite function of $w \circ \Phi$ across environments $\mathcal{E}_{all}$. The feature extractor $\Phi : \mathcal{X} \to \mathcal{H}$ maps $\mathcal{X}$ to representation space $\mathcal{H}$ to extract invariant features $h$ from $\mathcal{E}_{tr}$ which support $\mathbb{E}[y^e \mid \Phi(x^e) = h] = \mathbb{E}[y^{e'} \mid \Phi(x^{e'}) = h], \forall e, e' \in \mathcal{E}_{all}$, while the classifier $w : \mathcal{H} \to \mathcal{Y}$ simultaneously optimal the prediction in among $\mathcal{E}_{tr}$. Here is the formula minimization objective:

$$\min_{\boldsymbol{w}} \mathcal{L}(\boldsymbol{w}) := \sum_{e \in \mathcal{E}_{tr}} \mathcal{R}^e(\boldsymbol{w}) + \lambda \mathcal{P}(\boldsymbol{w}) \tag{21}$$

where $\mathcal{R}^e(\boldsymbol{w}) = \frac{1}{n_e} \sum_{i=1}^{n_e} \ell(f(\mathbf{x}_i^e, \boldsymbol{w}), \mathbf{y}_i^e)$ and $\ell$ is loss function. $\mathcal{P}(\boldsymbol{w})$ is a regularization encourage $f(:, w)$ to optimal all environments. IRM is mainly used to solve the problem of distributional shifts due to conditional probabilities $\mathbb{P}^e(y|x)$ in different environments, hoping to find stable and invirant features in different environments to solve the OOD problem, but IRM will face challenges with huge marginal probability shifts $\mathbb{P}^e(x)$.

## A.2  PRELIMINARY

**Lemma 2 (Temporal Covariate Shift).** *The root cause of Temporal Distribution Shift is the $\mathbb{P}(x)$ marginal distribution changes, while the $\mathbb{P}(y|x)$ conditional distribution in each domain remains unchanged.*

**Lemma 3 (Phase Congruency).** *In Frequency distribution shift, the change in the phase is small and can be ignored.*

**Lemma 4 (Distribution of Fourier Component).** The distributions of Fourier amplitude and phase can be modeled as Rayleigh distribution and uniform distribution respectively (He et al., 2023), the probabilistic density function of fourier component can be formulated:

$$f(a, p) = \text{Rayleigh}(a|\sigma) \cdot \text{U}(p|0, 2\pi) = \frac{a}{2\pi\sigma^2} \cdot \exp(-\frac{a^2}{2\sigma^2}), \tag{22}$$

where $a$ denotes the amplitude while $p$ represents phase, $\sigma$ is a variance parameter to scale the distribution of frequency component.

The marginal probabilistic density function of one fourier component amplitude can be viewed as the integral of $f(a, p)$ over $p \in [0, 2\pi], a \in (0, +\infty)$

$$f(a) = \int_0^{2\pi} f(a, p)dp = \frac{a}{\sigma^2} \cdot \exp(-\frac{a^2}{2\sigma^2})da \tag{23}$$

This also confirms the validity of the influence of linear phase changes on the probability distribution as stated in **Lemma 3**.

Then we can get the expectation $E(a)$ and variance $D(a)$ of the frequency components

$$E(a) = \sqrt{\frac{\pi}{2}}\sigma, D(a) = \frac{4 - \pi}{2}\sigma^2. \tag{24}$$

We can easily calculate the conclusion that $E(a)$ is an unbiased estimate of the statistic, which means that we can directly reflect the change on the $\sigma$ parameter by the average amplitude of frequency component, It is consistent with our motivation for magnitude decoupling.

**Corollary 1 (Frequency Marginal Probability Distribution of Time Series Data).** Combining **Lemma 3** and **Lemma 5** with frequency components independent assumption(Kegel et al., 2018), we can obtain the strict formulation of the frequency marginal probability distribution $\mathbb{P}_F(x)$ given a time series data in any domain,

$$\mathbb{P}_F(x) = \int f(F[1], ...F[n])dF = \int f(a_1, .., a_n) = \int \prod_{k=1}^n f(a_k)da = \int \prod_{k=1}^n \frac{a}{\sigma_k^2} \cdot \exp(-\frac{a^2}{2\sigma_k^2})da, \tag{25}$$

Naturally, we can use the principle of probabilistic independence to obtain the overall expectation and variance of the time series in the frequency domain perspective.

$$E(x) = \prod_{k=1}^n E(a_k) = (\frac{\pi}{2})^{\frac{n}{2}} \prod_{i=k}^n \sigma_k, D(x) = \sum_{k=1}^n D(a_k) = \frac{4 - \pi}{2} \sum_{k=1}^n \sigma_k^2, \tag{26}$$

**Lemma 5 (Domain Divergence).**(Germain et al., 2016) Suppose any domain $D_1$, $D_2$ are built on input variable $x$ and label variable $y$. Let $q > 0$ be a constant, the domain divergence between $D_1$ and $D_2$ is defined as

$$\beta_q(D_1\|D_2) = \left[\mathbb{E}_{(x,y)\sim D_2}\left(\frac{D_1(x, y)}{D_2(x, y)}\right)^q\right]^{\frac{1}{q}} = 2^{\frac{q-1}{q}RD_q(D_1\|D_2)}, \tag{27}$$

where $RD_q(\cdot)$ is Rényi Divergence.

**Corollary 2 (Bounding Domain Divergence in A Convex Hull).** Let $D$ be a set of source domains, denoted as $D = \{D_i\}_{i=1}^{N_d}$. A convex hull $\Lambda_D$ considered here consists of a mixture of distributions $\Lambda_D = \left\{\bar{D} : \bar{D}(\cdot) = \sum_{i=1}^{N_d} \pi_i D_i(\cdot), \pi_i \in \Delta_{N_d}\right\}$, where $\Delta_{N_d}$ is the $N_d - 1$-th dimensional simplex $(\forall \pi_i > 0, \sum_i^{N_d} \pi_i = 1)$. Let $\rho = 2^{\frac{q-1}{q}\sup_{i,j\in[N_d]} RD_q(D_i\|D_j)}$ then we have the following relation for the domain divergence when $q > 1$ between any pair of two domains $D', D'' \in \Lambda_S$ in the convex hull,

$$\beta_q(D' \| D'') \le \rho. \tag{28}$$

*Proof.* Suppose two unseen domains $D'$ and $D''$ on the convex hull $\Lambda_S$ of $N_d$ source domains with support $\mathcal{O}$. More specifically, let these two domains be $D' = \sum_{i=1}^{N_d} \pi_i D_i(\cdot)$ and $D'' = \sum_{i=1}^{N_d} \pi_j D_j(\cdot)$, then the domain divergence between $D'$ and $D''$ is

$$\beta_q(D'\|D'') = 2^{\frac{q-1}{q}RD_q(D'\|D'')}. \tag{29}$$

Let us consider the part of $RD_q(\cdot)$ as follows first,

$$RD_q(D'\|D'') = \frac{1}{q - 1}\ln\int_{\mathcal{O}}\left[\sum_{i=1}^{N_d} \pi_i D_i(x)\right]^q\left[\sum_{j=1}^{N_d} \pi_j D_j(x)\right]^{1-q} dx \tag{30}$$

Since $\forall \pi_i \in \Delta_{N_d}$, We knows that $\sum_{i=1}^{N_d} \pi_i = 1, \forall \pi_i \geq 0$, when $q > 1$ we could hold $f(x) = x^q$ and $g(x) = x^{1-q}$ are convex function, their second-order derivative coefficients are the same and positive $q \times (q-1) > 0$.

Thus the original equation satisfies Jensen's Inequality (Jensen, 1906) and we get the following inequality

$$RD_q(D'\|D'') = \frac{1}{q-1} \ln \int_{\mathcal{O}} \left[ \sum_{i=1}^{N_d} \pi_i D_i(x) \right]^q \left[ \sum_{j=1}^{N_d} \pi_j D_j(x) \right]^{1-q} dx \tag{31}$$

$$\leq \frac{1}{q-1} \ln \int_{\mathcal{O}} \sum_{i=1}^{N_d} \pi_i \left[ D_i(x) \right]^q \sum_{i=j}^{N_d} \pi_j \left[ D_j(x) \right]^{1-q} dx \tag{32}$$

$$\leq \frac{1}{q-1} \ln \sum_{i=1}^{N_d} \sum_{i=j}^{N_d} \pi_i \pi_j \int_{\mathcal{O}} \left[ D_i(x) \right]^q \left[ D_j(x) \right]^{1-q} dx \tag{33}$$

$$\leq \frac{1}{q-1} \ln \int_{\mathcal{O}} \left[ D_i(x) \right]^q \left[ D_j(x) \right]^{1-q} dx \tag{34}$$

$$\leq \sup_{i,j \in [N_d]} RD_q(D_i \| D_j) \tag{35}$$

Then we have

$$\beta_q(D'\|D'') = 2^{\frac{q-1}{q} RD_q(D'\|D'')} \leq 2^{\frac{q-1}{q} \sup_{i,j \in [N_d]} RD_q(D_i\|D_j)} = \rho. \tag{36}$$

$\square$

## A.3 PROOF OF PROPOSITION 5.1

**Proposition 5.1.** *Assuming time series data $x_t \in \mathbb{R}^{L \times C}$ through frequency decomposition into time-deterministic components $A_{det} = \{a_i\} \in \mathbb{R}^{K \times C}$ and time-stochastic components $A_{sto} = \{a_i\} \in \mathbb{R}^{(L/2-K) \times C}$, the invariant minimization objective formula suitable for OOD time series classification is as follows:*

$$\min_{\boldsymbol{w}} \mathcal{L}(\boldsymbol{w}) := \sum_{e \in \mathcal{E}_{tr}} \mathcal{R}^e(\Phi_{det}(A_{det})) + \lambda_{det}\mathcal{P}(\Phi_{det}(A_{det})) + \lambda_{sto}\mathcal{J}(A_{sto}), \tag{37}$$

*where $\mathcal{R}^e$ denotes the risk in domain environment $e$, $\Phi_{det}$ denotes invariant feature extractor. $\mathcal{P}$ is the regularization for invariant feature, $\mathcal{J}_{\phi_{\boldsymbol{w}}}$ capture auxiliary features avoiding information lose.*

*Proof.* Since the frequency components obtained through the Fourier transform are orthogonal, we can consider these frequencies to be probabilistically independent of each other.

$$F[k] = \mathcal{F}[f(t)] = \int_{-\infty}^{\infty} f(t)e^{-i2\pi kt} dt = A_k e^{jP_k} \tag{38}$$

where $A_k$ is the amplitude and $P_k$ is the phase of the $k$-th frequency conponent, thus we obtain:

$$\forall i, j, F[i] \perp F[j] \rightarrow \mathbb{P}_F(F[i]) \perp\!\!\!\perp \mathbb{P}_F(F[j]) \tag{39}$$

From **Lemma 2**, **Lemma 3**, the phase is not influenced by distributional shifts, so the probability of frequency components is determined by the amplitude frequency.

$$\mathbb{P}_F(x) = \prod_{k=1}^{K} \mathbb{P}_F(F[k]) = \prod_{k=1}^{K} \mathbb{P}_F(A_k) \tag{40}$$

After Frequency Filter, the time series decomposed into the time-deterministic components $A_{det} = \{a_1, a_2, \ldots, a_k\}$ and time-stochastic components $A_{sto} = \{a_{k+1}, \ldots, a_{L/2}\}$. Then we could rewrite the $\mathbb{P}_F(x)$

$$\mathbb{P}_F(x) = \mathbb{P}_F(A_{det}) \times \mathbb{P}_F(A_{sto}) \tag{41}$$

The probability of $\mathbb{P}_F(A_{det})$ can be considered unaffected by temporal influences cross domains, i.e $\forall 1 \le i \ne j \le T, \text{SUPP}(\mathbb{P}_F^{D_k^i}(A_{det})) = \text{SUPP}(\mathbb{P}_F^{D_k^j}(A_{det}))$, thus making this feature suitable for the IRM invariance theory. thus making this feature suitable for the IRM invariance theory by holding on the conidtion assumption,

$$\mathbb{E}[y^e \mid \Phi(A_{det})] = \mathbb{E}[y^{e'} \mid \Phi(A_{det})], \forall e, e' \in \mathcal{E}_{all} \tag{42}$$

At the same time, despite $\forall 1 \le i \ne j \le T, \text{SUPP}(\mathbb{P}_F^{D_k^i}(A_{sto})) \ne \text{SUPP}(\mathbb{P}_F^{D_k^j}(A_{sto}))$ could lead to the frequency distribution shift, the behavior on the time series is influenced by moving average weights, suppose a learnable feature reweight extractor could alleviate the shift optimized with ERM $J_{\phi_w}(\cdot)$ since $\mathbb{P}^{D_k}(y|x)$ don't change ensure its practicability to help generalization space without affecting $\Phi_{det}$ extractor and losing information.

$$\min_{\boldsymbol{w}} \mathcal{L}(\boldsymbol{w}) := \sum_{e \in \mathcal{E}_{tr}} \mathcal{R}^e(\Phi_{det}(A_{det})) + \lambda_{det}\mathcal{P}(\Phi_{det}(A_{det})) + \lambda_{sto}\mathcal{J}_{\phi_w}(A_{sto}) \tag{43}$$

Therefore, we complete the proof. $\qquad\square$

### A.4 THE RISK FOR DOMAIN GENERALIZATION OF TIME SERIES FROM FREQUENCY VIEW

**Theorem 1 (PAC-Bayesian Risk Bound on Unseen Time series Domain).**(Germain et al., 2016) Let $\mathcal{H}$ be a hypothesis space built from a set of source domains, denoted as $D = \{D_i\}_{i=1}^{N_d}$. Suppose $q > 0$ is a constant, for any unseen domain $D_U$ from the convex hull $\Lambda_D$, we have its closest element $D_{\bar{U}}$ in $\Lambda_D$, i.e.,$D_{\bar{U}} = \arg\min_{\pi_1,\dots,\pi_{N_d}} \beta_q(D_{\bar{U}} \| \sum_{i=1}^{N_d} \pi_i D_i)$. Then the risk of $D_U$ on any label function $h \in \mathcal{H}$ is,

$$R_{D_U}[h] \le \frac{1}{2}d_{D_U}(h) + \epsilon \cdot \left[e_{D_{\bar{U}}}(h)\right]^{1-\frac{1}{q}} + \eta_{T/S}, \tag{44}$$

where $d_D(h)$ and $e_D(h)$ are ideal and expected risk of a domain $D$ respectively, $\epsilon = \beta_q(D_U \| \sum_{i=1}^{N_d} \pi_i D_i)$ is an ideal distance since we can't have access to $D_U$, while $\eta_{T/S}$ denotes the distribution of $(x, y) \in \text{SUPP}(Test) \backslash \text{SUPP}(Source)$, it is usually a small value.

Suppose $(x, y)$ between the unseen domain for testing and source doamins have been fully covered by $\Lambda_S$, then $\eta_{T/S} = 0$ and there exists a finite upper bound $\rho = \sup_{i,j \in [N_d]} \beta_q(D_i \| D_j), \forall q > 0$ for any convex combinatorial domains $D_i$ and $D_j$.

$$R_{D_U}[h] \le \frac{1}{2}d_{D_U}(h) + \rho \cdot \left[e_{D_{\bar{U}}}(h)\right]^{1-\frac{1}{q}}, \tag{45}$$

where $\rho$ denotes the maximum domain divergence of source domains we could minimize, and $e_{D_{\bar{U}}}(h)$ represents that the empirical risks of source domains to be minimized.

**Proposition 5.2 (Frequency Perspective Risk Bound on Unseen Time series Domain).***Let $\mathcal{H}$ be a hypothesis space built from a set of source time series domains $D = \{D_i\}_{i=1}^{N_d}$. Suppose $q > 0$ is a constant, for any unseen time series domain $D_U$ from the convex hull $\Lambda_D$, we have its closest element $D_{\bar{U}}$ related to source domains in $\Lambda_D$, i.e.,$D_{\bar{U}} = \arg\min_{\pi_1,\dots,\pi_{N_d}} \beta_q(D_{\bar{U}} \| \sum_{i=1}^{N_d} \pi_i D_i)$. Then the risk of $D_U$ on any label function $h \in \mathcal{H}$ is,*

$$R_{D_U}[h] \le \frac{1}{2}d_{D_U}(h) + \rho \cdot \left[e_{D_{\bar{U}}}(h)\right]^{1-\frac{1}{q}}, \tag{46}$$

*where $\rho = 2^{\frac{q-1}{q}\sup_{i,j\in[N_d]} RD_q(D_i\|D_j)}$, $d_D(h)$ and $e_D(h)$ are ideal and empirical risk of domain D,*

$$RD_q(D_i\|D_j) = \frac{1}{q-1}\log\int [P_F(D_i)]^q [P_F(D_j)]^{1-q}\,da = \frac{1}{q-1}\log\left[\frac{\mu_j^{2(q-1)}}{\mu_i^{2q}} \cdot \frac{\sqrt{\pi}^{-2n+1}n!!}{2^{2n+1}\sqrt{\gamma}^{n+1}}\right] \tag{47}$$

where $\mu_i = E(D_i) = \prod_{k=1}^{n} \sqrt{\frac{\pi}{2}}\sigma_k, \mu_j = E(D_j) = \prod_{k=1}^{n} \sqrt{\frac{\pi}{2}}\tau_k, \gamma = \sum_{k=1}^{n} \frac{q}{2\sigma_k^2} + \frac{1-q}{2\tau_k^2}, \{\sigma_k\}_{k=1}^{n}$ and $\{\tau_k\}_{k=1}^{n}$ denote frequency scale parameters in $D_i, D_j, n$ represents the number of components.

*Proof.* Suppose the overall distribution supports condition is $\text{SUPP}(D_i) = \text{SUPP}(D_j)$, their corresponding overall probability distributions on the frequency domain $P_F(x)$ also satisfy the condition, then it is feasible for us to use the generalized Rényi Divergence to estimate its whole probabilistic density function $f(a) = \prod_{k=1}^{n} f(a_k)$, It is a reasonable extension for the individual frequency domain components mentioned in Raincoat(He et al., 2023) do not satisfy the KL-Divergence in the range of $a_k \in (0, +\infty)$. We define the probability density functions $f_i(a), f_j(a)$ corresponding to the two distributions $D_i$ and $D_j$,

$$f_i(a) = \prod_{k=1}^{n} \frac{a}{\sigma_k^2} \cdot \exp(-\frac{a^2}{2\sigma_k^2}), f_j(a) = \prod_{k=1}^{n} \frac{a}{\tau_k^2} \cdot \exp(-\frac{a^2}{2\tau_k^2}) \tag{48}$$

where $\sigma$ and $\tau$ denotes the scale parameters in $D_i, D_j$ respectively, while $n$ is the number of frequency components. Then we can calculate the formulation of $\frac{f_i(a)}{f_j(a)}$,

$$\frac{f_i(a)}{f_j(a)} = \prod_{k=1}^{n} \frac{\frac{a}{\sigma_k^2} \cdot \exp(-\frac{a^2}{2\sigma_k^2})}{\frac{a}{\tau_k^2} \cdot \exp(-\frac{a^2}{2\tau_k^2})} = \prod_{k=1}^{n} \frac{\tau_k^2}{\sigma_k^2} \exp\left(-a^2(\frac{1}{2\sigma_k^2} - \frac{1}{2\tau_k^2})\right) \tag{49}$$

Following these, we can continue to derive $RD_q(D_i\|D_j)$,

$$RD_q(D_i\|D_j) = \frac{1}{q-1} \log \int [f_i(a)]^q [f_j(a)]^{1-q} da \tag{50}$$

$$= \frac{1}{q-1} \log \int \left[\frac{f_i(a)}{f_j(a)}\right]^q f_j(a) da \tag{51}$$

$$= \frac{1}{q-1} \log \int \left[\prod_{k=1}^{n} \frac{\tau_k^2}{\sigma_k^2} \exp\left(-a^2(\frac{1}{2\sigma_k^2} - \frac{1}{2\tau_k^2})\right)\right]^q \cdot \prod_{k=1}^{n} \frac{a}{\tau_k^2} \cdot \exp(-\frac{a^2}{2\tau_k^2}) da \tag{52}$$

$$= \frac{1}{q-1} \log \int \prod_{k=1}^{n} \frac{\tau_k^{2q-2}}{\sigma_k^{2q}} a^n \cdot \exp\left(-qa^2 \sum_{k=1}^{n}(\frac{1}{2\sigma_k^2} - \frac{1}{2\tau_k^2}) - a^2 \sum_{k=1}^{n} \frac{1}{2\tau_k^2}\right) \tag{53}$$

$$= \frac{1}{q-1} \log \int \prod_{k=1}^{n} \frac{\tau_k^{2q-2}}{\sigma_k^{2q}} a^n \cdot \exp\left(-a^2 \sum_{k=1}^{n}(\frac{q}{2\sigma_k^2} + \frac{1-q}{2\tau_k^2})\right) \tag{54}$$

Let $A = \prod_{k=1}^{n} \frac{\tau_k^{2q-2}}{\sigma_k^{2q}}, B = \sum_{k=1}^{n}(\frac{q}{2\sigma_k^2} + \frac{1-q}{2\tau_k^2})$,

$$RD_q(D_i\|D_j) = \frac{1}{q-1} \log A \int_0^{+\infty} a^n \cdot exp(-Ba^2) da \tag{55}$$

$$= \frac{1}{q-1} \log A \frac{\sqrt{\pi}n!!}{(2\sqrt{B})^{n+1}} \tag{56}$$

Let $\mu_i = E(D_i) = \prod_{k=1}^{n} \sqrt{\frac{\pi}{2}}\sigma_k, \mu_j = E(D_j) = \prod_{k=1}^{n} \sqrt{\frac{\pi}{2}}\tau_k, \gamma = B$,

$$RD_q(D_i\|D_j) = \frac{1}{q-1} \log \left(\left[\prod_{k=1}^{n} \frac{\tau_k^{2q-2}}{\sigma_k^{2q}}\right] \cdot \left[\frac{\sqrt{\pi}n!!}{(2\sqrt{\gamma})^{n+1}}\right]\right) \tag{57}$$

$$= \frac{1}{q-1} \log \left(\frac{\mu_j^{2q-2}}{\mu_i^{2q}} \cdot (\frac{\pi}{2})^n \cdot \frac{\sqrt{\pi}n!!}{(2\sqrt{\gamma})^{n+1}}\right) \tag{58}$$

$$= \frac{1}{q-1} \log \left[\frac{\mu_j^{2(q-1)}}{\mu_i^{2q}} \cdot \frac{\sqrt{\pi}^{2n+1}n!!}{2^{2n+1}\sqrt{\gamma}^{n+1}}\right] \tag{59}$$

It is easy to determine $\lim_{n \to +\infty} \frac{\sqrt{\pi}^{2n+1}n!!}{2^{2n+1}\sqrt{\gamma}^{n+1}} \to +\infty$, At the same time, we plotted its monotonicity for $n \in \mathbb{N}^+, 1 \le n \le 100, \gamma \in \mathbb{N}^+ 1 \le n \le 20$ in Figure 6 (a). it usually shows a monotonically increasing trend.

Let's consider $R_{D_U}[h]$ where $\rho = 2^{\frac{q-1}{q} \sup_{i,j \in [N_d]} RD_q(D_i \| D_j)}$, $RD_q(\cdot \| \cdot) \geq 0$

$$R_{D_U}[h] \leq \frac{1}{2} d_{D_U}(h) + \rho \cdot \left[ e_{D_{\bar{U}}}(h) \right]^{1-\frac{1}{q}}, \tag{60}$$

(1) $q > 1$, both $\rho$ and $\left[ e_{D_{\bar{U}}}(h) \right]^{1-\frac{1}{q}}$ monotonically increasing $\uparrow$, if we reduce $\rho$ and $e_{D_{\bar{U}}}$, The upper bound of $R_{D_U}[h] \downarrow$. We could decrease $n$ and the $\log$ part will decrease.

(2) $0 < q < 1$, both $\rho$ and $\left[ e_{D_{\bar{U}}}(h) \right]^{1-\frac{1}{q}}$ monotonically decreasing $\downarrow$, When we minimize the $e_{D_{\bar{U}}}$, $\left[ e_{D_{\bar{U}}}(h) \right]^{1-\frac{1}{q}} \uparrow$, so we should decrease the $\rho$ to let the upper bound of $R_{D_U}[h] \downarrow$, it requires $RD_q(\cdot \| \cdot)$ larger, in this case, $\frac{1}{q-1} < 0$ so we also need to decrease the $\log$ part by decreasing $n$. $\qquad \square$

# B DATASET

## B.1 DATASET INFORMATION

We list detail introduction of the datasets we used with FEDNet:

- Spurious Fourier (Gagnon-Audet et al., 2022) dataset is designed to study the impact of spurious correlations in one-dimensional signals under distribution shifts. It involves binary classification tasks based on the frequency characteristics of the signals. Each signal is constructed from Fourier spectra with one low-frequency peak and one high-frequency peak. The dataset comprises different domains, which are 10%, 80%, and 90%, representing the correlation between the low-frequency signal and the label. In contrast, the high-frequency signal maintains a consistent 75% correlation with the label across all domains.

- HHAR (Gagnon-Audet et al., 2022) dataset is used to study human activity recognition across different smart devices, such as smartphones and smartwatches. This dataset includes five source domains, each containing data gathered from a different device. The goal is to train models that can generalize to unseen devices, effectively ignoring spurious information from complex signals.

- UCIHAR (Anguita et al., 2012) dataset captures daily activities of 30 volunteers aged 19 to 48 using mobile phone sensors. It features a sampling frequency of 50 Hz and contains 1,318,272 time series samples, each with 9 initial features. The classification task involves identifying one of six activities: walking, sitting, lying down, standing, going upstairs, and going downstairs. The dataset is organized into 5 domains based on participants, with each domain comprising data from 6 volunteers.

- UniMiB-SHAR (Micucci et al., 2017) dataset consists of activity data gathered from three mobile phone sensors at a sampling frequency of 50 Hz, involving 30 participants aged 18 to 60. These participants performed 17 detailed actions, including 9 everyday activities and 8 types of falls. For evaluation, the dataset is divided into 4 domains. It includes a total of 1,569 time series samples, each with 453-dimensional features derived from the three sensors.

- Opportunity (Chavarriaga et al., 2013) dataset contains data from 4 volunteers performing 18 daily activities in a home environment, such as opening and closing the dishwasher, refrigerator, drawers, and etc. It uses various inertial sensors to enhance the generalization of OOD data. The dataset is sampled at 30 Hz, resulting in a total of 869,387 time series samples, each with 77 features. It is divided into 4 domains for evaluation purposes.

- DSADS (Barshan & Yüksek, 2014) dataset consists of 19 activities collected from 8 subjects wearing body-worn sensors on 5 different body parts. It captures a variety of daily and sports activities, providing comprehensive data for human activity recognition research.

- PAMAP (Reiss & Stricker, 2012) dataset includes data on 18 activities performed by 9 subjects, each wearing 3 sensors. This dataset focuses on physical activities and is designed to aid in the development of models for recognizing a wide range of movements.

- EMG (Lobov et al., 2018) consists of 6 types of gestures with 8 channels recorded from 36 participants sampled at 200 Hz.

- EEG (Goldberger et al., 2000) is a single channel EEG dataset collected from 20 subjects to classify 5 sleep stages.

Table 3: Dataset statistics.

| Dataset | Shape | Classes | Domains | samples | Subjects | Sensors | Frequency |
|---------|-------|---------|---------|---------|----------|---------|-----------|
| Spurious Fourier | (50, 1) | 2 | 3 | 12,000 | - | - | - |
| HHAR | (500, 6) | 6 | 5 | 13,674 | 9 | 5 | 25 Hz |
| UCIHAR | (125, 45) | 6 | 5 | 1,318,272 | 30 | 1 | 50 Hz |
| UniMiB-SHAR | (151, 3) | 18 | 4 | 11,771 | 30 | 1 | 50 Hz |
| Opportunity | (30, 77) | 17 | 4 | 869,387 | 4 | 72 | 30 Hz |
| DSADS | (125, 45) | 19 | 4 | 1,140,000 | 8 | 5 | 25 Hz |
| PAMAP | (200, 27) | 18 | 4 | 3,850,505 | 9 | 3 | 100 Hz |
| EMG | (200, 8) | 6 | 4 | 33,903,472 | 36 | 1 | 200 Hz |
| EEG | (3000, 1) | 5 | 4 | - | 20 | 1 | - |

## B.2 DATA PROCESSING

We will provide preprocessing code for all datasets that need to be processed. The preprocessing methods will be consistent with those mentioned in other works and with open-source code. The detailed preprocessing procedures for some datasets are as follows:

**DSADS Cross-Position.** The DSADS dataset consists of 5 sensors positioned at torso (T), right arm (RA), left arm (LA), right leg (RL), and left leg (LL). Each sensor records 9-dimensional variables (x, y, z accelerometers; x, y, z gyroscopes; x, y, z magnetometers) representing the position in space. Originally, the data had dimensions of 125x1x45. We split it into 5 domains based on the positions, resulting in final data dimensions of 125x1x9. This dataset is used to study more challenging domain generalization issues across different body parts.

**PAMAP.** For the PAMAP dataset, we followed the processing method of Diversify, selecting all sample records and categories. The original data records were used with a fixed window size of 200 and a window overlap ratio of 50%.

**Initial Domain splitting** We list detail initial domains of the datasets we used in Table 4.

Table 4: Initial domain information.

| Dataset | Domains | Infomation |
|---------|---------|------------|
| Spurious Fourier | 3 | {d=10%, d=80%, d=90%} |
| HHAR | 5 | {Nexus 4, Galaxy S3, Galaxy S3 Mini, LG watch, Gear watch} |
| UCIHAR | 5 | {0,1,2,3,4,5} |
| UniMiB-SHAR | 4 | {1,2,3,5} |
| Opportunity | 4 | {S1,S2,S3,S4} |
| DSADS | 4 | {(0,1), (2,3), (4,5), (6,7)} |
| PAMAP | 4 | {(0,1,2,11), (3,5,6,9), (7,8,10,13), (4,12)} |
| EMG | 4 | {(0-8), (9-17), (18-26), (27-35)} |
| EEG | 4 | {(0,1,2,3,4),(5,6,7,8,9),(10,11,12,13,14),(15,16,17,18,19)} |

## C SUPPLEMENTARY EXPERIMENTAL RESULTS

### C.1 FULL ABLATION RESULTS

Table 5: Ablation Settings of FEDNet.

| Model | Time-Deterministic | | | Time-Stochastic | Classification |
|-------|--------------------|--|--|-----------------|----------------|
| | $\mathcal{L}_{ELOB}$ | $\mathcal{L}_{con1}$ | $\mathcal{L}_{con2}$ | $\mathcal{L}_{sto}$ | $\mathcal{L}_{cls}$ |
| w/o $\mathcal{L}_{con}$ | ✓ | ✗ | ✗ | ✓ | ✓ |
| w/o $\mathcal{L}_{det}$ | ✗ | ✗ | ✗ | ✓ | ✓ |
| w/o $\mathcal{L}_{sto}$ | ✓ | ✗ | ✓ | ✗ | ✓ |
| FEDNet-cross | ✓ | ✓ | ✗ | ✓ | ✓ |
| FEDNet-contrast | ✓ | ✗ | ✓ | ✓ | ✓ |

We conduct ablation study to verify the impact of each technique in FEDNet on performance, and the details of our setup are provided in Table 5. As shown in Table 6, it is evident that $\mathcal{L}_{con}$ and

$\mathcal{L}_{det}$ significantly affect the model's performance. For all datasets, after removing the constraints of the $x_{det}$ part, the performance generally shows a significant decrease, which indicates that there is Domain Shift in this part of the features, and removing $x_{det}$ shows a very large decrease in the performance on all datasets. On the other hand, $\mathcal{L}_{sto}$ has a smaller impact on performance and even results in negative optimization in some cases (e.g., when the target domain is S3 on the Opportunity dataset), this is due to the low signal-to-noise ratio in the information of the $x_{sto}$ features. Additionally, we can see that FEDNet outperforms its three variants in most results, demonstrating the soundness of our design.

**Abnormal results in Opportunity dataset.** due to the original time series sequence window length $L = 30$ being much shorter than that of other time series containing limited information, after applying the Fourier transform, only $L/2$ frequency components remain, and further decomposition results in only 2-3 stable frequency components extracted, making prediction difficult in w/o $\mathcal{L}_{con}$.

Table 6: Ablation study of FEDNet.

| Dataset | Target | w/o $\mathcal{L}_{con}$ | w/o $\mathcal{L}_{det}$ | w/o $\mathcal{L}_{sto}$ | FEDNet-cross | FEDNet-contrast |
|---|---|---|---|---|---|---|
| Spurious Fourier | 0 | 50.12 | 50.78 | 74.38 | 74.22 | **75.31** |
| HHAR | 0 | 83.18 | 22.23 | 96.34 | 99.18 | **99.48** |
| | 1 | 83.98 | 22.44 | 96.88 | 98.98 | **98.98** |
| | 2 | 55.45 | 22.27 | 87.64 | 90.97 | **91.59** |
| | 3 | 45.65 | 17.17 | 60.56 | 58.18 | **67.87** |
| | 4 | 42.57 | 20.27 | 43.92 | 58.11 | **60.81** |
| UCIHAR | 0 | 60.80 | 15.27 | 90.21 | **99.42** | 95.67 |
| | 1 | 41.72 | 17.88 | 76.82 | **83.74** | 80.13 |
| | 2 | 70.96 | 18.18 | 96.48 | 97.15 | **98.53** |
| | 3 | 51.10 | 18.92 | 89.58 | 93.21 | **94.64** |
| | 4 | 54.63 | 17.76 | 97.79 | 98.34 | **99.02** |
| UniMiB-SHAR | 1 | 23.69 | 10.93 | 53.42 | 57.55 | **58.85** |
| | 2 | 19.21 | 24.52 | 44.25 | 63.63 | **70.15** |
| | 3 | 18.75 | 10.52 | 68.09 | 70.06 | **71.05** |
| | 5 | 31.87 | 18.79 | 39.93 | **44.63** | 44.29 |
| Opportunity | S1 | 12.21 | 63.23 | 84.62 | 84.80 | **85.02** |
| | S2 | 15.26 | 62.38 | 81.20 | 79.75 | **81.45** |
| | S3 | 29.34 | 51.42 | 78.77 | 77.89 | **79.21** |
| | S4 | 21.15 | 53.39 | 81.20 | 81.36 | **81.96** |
| DSADS Cross Position | 0 | 11.14 | 5.26 | **40.89** | 37.54 | 40.54 |
| | 1 | 11.97 | 5.26 | 32.47 | 35.13 | **36.05** |
| | 2 | 13.66 | 5.26 | 36.25 | 36.25 | **36.25** |
| | 3 | 9.52 | 5.26 | 33.28 | 32.16 | **33.93** |
| | 4 | 13.53 | 5.26 | **35.08** | 32.78 | 33.04 |
| EMG | 0 | 46.00 | 17.01 | 71.77 | 64.84 | **73.00** |
| | 1 | 41.28 | 16.48 | 85.39 | 82.13 | **87.10** |
| | 2 | 41.38 | 16.44 | 78.82 | 71.53 | **79.66** |
| | 3 | 45.12 | 16.42 | 78.08 | 69.87 | **79.85** |
| DSADS | 0 | 84.60 | 5.26 | 88.20 | 90.74 | **92.80** |
| | 1 | 76.84 | 5.26 | 83.11 | 82.63 | **84.86** |
| | 2 | 86.31 | 5.26 | 87.89 | 87.36 | **93.24** |
| | 3 | 82.67 | 5.26 | 86.85 | 71.88 | **87.71** |
| PAMAP | 0 | 65.24 | 44.56 | 66.93 | 66.53 | **67.48** |
| | 1 | 63.71 | 59.32 | 54.81 | 47.93 | **67.08** |
| | 2 | 35.02 | 37.57 | 22.91 | 28.32 | **35.27** |
| | 3 | 66.45 | 68.03 | 68.08 | 64.73 | **69.80** |

## C.2    TOP AVERAGE AMPLITUDE RATIO EFFECT

We further conduct key experiments on the proportion $\alpha$ of the acquired high average amplitude. The results in Figure 7 show that we only need to extract 5-20% of the original frequency as a time-deterministic component to reach state-of-the-art performance, which suggests that the use of the frequency-domain component as prior information is robust and conducive to the model's generalization ability.

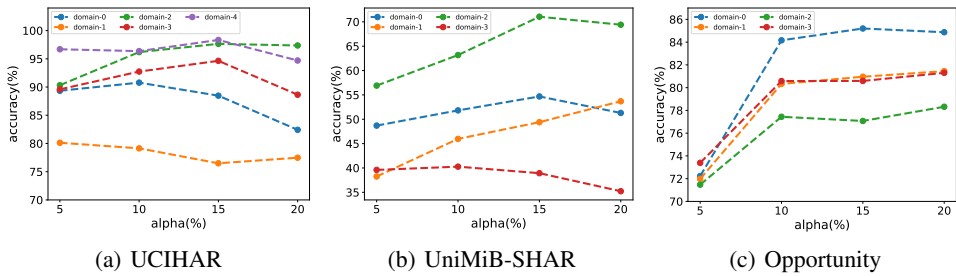

(a) UCIHAR       (b) UniMiB-SHAR       (c) Opportunity

Figure 7: Hyper-parameter study of $\alpha$ on three datasets. The X-axis represents $\alpha\%$ of top average amplitude frequency components, and the Y-axis represents accuracy in time series classification.

## C.3 TIME-STOCHASTIC FEATURE STUDY

We conducted experimental analysis on the features of the Time-stochastic component across three different dimensions, demonstrating that these features can serve as auxiliary information to effectively enhance the model's generalization ability, with potential for further optimization.

**Patch condition number (feature level).** We analyzed the original patch matrix after dividing the Time-stochastic feature into patches. We used the condition number $\text{cond}(x_{patch}) = \sigma_{max}/\sigma_{min}$, where $\sigma_{max}$ and $\sigma_{min}$ represent the maximum and minimum singular values of $x_{patch}$ to evaluate this part of the features, as shown in Table 7. We found that using non-overlapping patches effectively reduces the condition number, ensuring the numerical stability of the original matrix. It allows the module learning more effectively and further speeding up the training process.

Table 7: Time-stochastic patch matrix condition number with different mask ratio $\alpha$.

| Dataset | $\alpha = 0.2$ | | $\alpha = 0.4$ | | $\alpha = 0.6$ | |
| --- | --- | --- | --- | --- | --- | --- |
| | $stride = P/2$ | $stride = P$ | $stride = P/2$ | $stride = P$ | $stride = P/2$ | $stride = P$ |
| Spurious Fourier | 377.80 | **14.57**↓ | 1211.58 | **41.70**↓ | 200.43 | **24.53**↓ |
| HHAR | 57.31 | 93.70 | 6702.17 | **5661.14**↓ | 4149784.84 | **3688642.06**↓ |
| UCIHAR | 7600.79 | **30.31**↓ | 159401 | **37.88**↓ | 147293444 | **161.05**↓ |
| SHAR | 47.13 | **11.21**↓ | 9288.10 | **17.402**↓ | 830192 | **38.3774**↓ |
| EMG | 27.26 | 84.076 | 2065.86 | **152.440**↓ | 1561521 | **12334.38**↓ |
| OPP | 151.68 | **2.0032**↓ | 59.24 | **1.8594**↓ | 81.89 | **1.78146**↓ |
| DSADS | 351.83 | **26.7469**↓ | 2260.30 | **15.1602**↓ | 3226488.87 | **37.6795**↓ |
| PAMAP | 49.98 | 71.068 | 4921.72 | **245.448**↓ | 2968754.15 | **26859.82**↓ |

**Time series missing value study (data level).** We further analysed a more realistic scenario, i.e., we tried to add different missing rates of values to the training domains. We found it will affect the extraction of time-deterministic features with Freqency Filter. In this case, the supplementation of Time-stochastic brought significant gains. Table 8 shows that retaining time-stochastic features can improve model's generalization when the invariant features are not sufficiently extracted.

Table 8: time-stochastic gains with different time series missing rate.

| Dataset | missing rate = 20% | | missing rate = 40% | | missing rate = 60% | | missing rate = 80% | |
| --- | --- | --- | --- | --- | --- | --- | --- | --- |
| | w/o $L_{sto}$ | FEDNet | w/o $L_{sto}$ | FEDNet | w/o $L_{sto}$ | FEDNet | w/o $L_{sto}$ | FEDNet |
| UCIHAR | 94.52 | **98.55** | 94.52 | **97.41** | 64.55 | **70.60** | 69.74 | **74.06** |
| UniMiB-SHAR | 52.60 | **52.86** | 50.52 | **51.56** | 38.54 | **40.36** | 30.98 | **38.80** |
| EMG | 67.43 | **67.49** | 38.62 | **40.73** | 16.55 | **27.88** | 16.55 | **23.59** |

**EEG Long signal case study (domain level).** We investigated the OOD generalization for the classification of EEG signals with an extremely long sequence (L=3000). As shown in Figure 8, the spectrum maps produced by ultra-long sequences are more likely to contain unknown frequency distributions. When we use domain 2 as the target domain, our model is unable to detect frequency

variations specific to the high-frequency part from the training domain, which may also belong to invariant features.

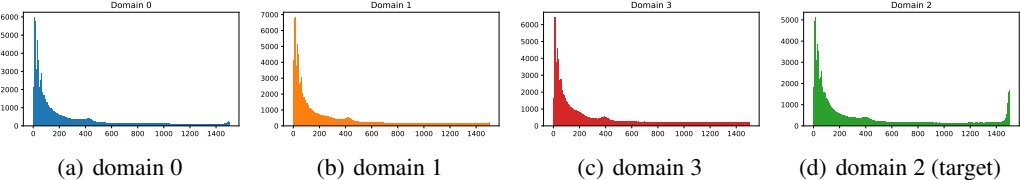

(a) domain 0      (b) domain 1      (c) domain 3      (d) domain 2 (target)

Figure 8: EEG training domain mean frequency spectrum.

This example shows us the shortcomings of FEDNet, i.e., the invariant features learnt from stable frequencies are not sufficient if we rely exclusively on the training domain to extract them, especially in the case of ultra-long sequences where many of the frequency components are lost. This is why we need to retain the time-stochastic module. From Table 9, there is a 3% difference between w/o $L_{sto}$ and FEDNet, while general domain generalization methods are always the lowest. We have also tried to simplify the attention mechanism with the MLP and the deep separable convolution.

Table 9: EEG target domain 2 performance.

| Method | Accuracy | F1-score | Precision | Recall |
|---|---|---|---|---|
| VREx | 68.58 | 56.95 | 57.44 | 59.02 |
| GroupDRO | 69.40 | 55.83 | 58.31 | 56.53 |
| ANDMask | 69.44 | 57.47 | 59.40 | 57.97 |
| FEDNetw/o $L_{sto}$ | 69.99 | 57.97 | **63.26** | 59.31 |
| FEDNet+ self-attention | **72.90** | 62.16 | 61.74 | 63.01 |
| FEDNet+ DwConv | 70.21 | 57.78 | 56.33 | 62.09 |
| FEDNet+ MLP | 71.85 | **62.94** | 61.80 | **65.19** |

## C.4 EMPIRICAL DOMAIN DIVERGENCE STUDY WITH $\alpha$ ISOLATED LEVEL

We theoretically analyzed the changes in the maximum distance of the dataset under four retention ratios: $\alpha = [0.2, 0.4, 0.6, 0.8]$. From Figure 10 and 9 We found that the maximum distance is generally smallest when $\alpha$ is 0.2 or 0.4, and the overall trend shows that as the retention ratio decreases, the domain generalization distance also decreases. The only exception is the UniMiB-SHAR dataset, where the calculated denominator contains a very small value of $\hat{\gamma} < 0.01$, which causes the overall result to be significantly large. However, we conducted mask ratio experiments on UniMiB-SHAR and found that the optimal frequency is also concentrated at 0.2-0.4 From Table 13.

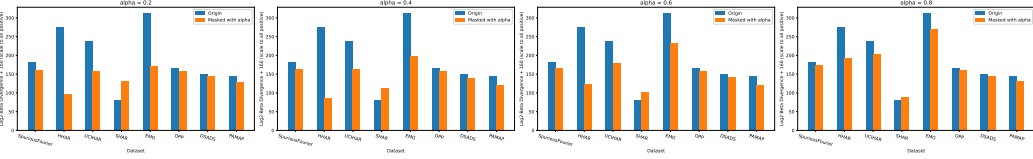

Figure 9: empirical max domain divergence with different $\alpha$ when $q = 2$.

## C.5 INTRINSIC RELATIONSHIP STUDY OF ORTHOGONAL FREQUENCY DECOMPOSITION

In addition to the orthogonal FFT, We conducted experiments using orthogonal wavelet mother functions and non-orthogonal wavelet mother functions in Table 10 and found that orthogonal wavelet functions generally outperform the non-orthogonal case, which is consistent with our theoretical

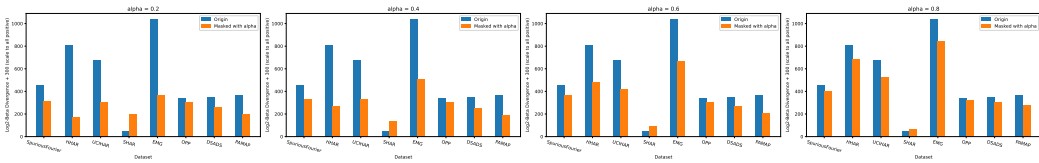

Figure 10: empirical max domain divergence with different $\alpha$ when $q = 1/2$.

insight Eq. (39). It is the orthogonality of the frequency components that allows us to treat the probability distributions of each frequency component as independent of each other. This independence allows us to model the effects of these distributions separately.

Table 10: Performance on different wavelet mother functions for frequency decomposition.

| Dataset | Target | orthogonal | | | | non-orthogonal | |
|---|---|---|---|---|---|---|---|
| | | db2 | | coif1 | | rbio1.3 | |
| | | Accuracy | F1 | Accuracy | F1 | Accuracy | F1 |
| Spurious Fourier | d=10% | 18.00 | 17.81 | **33.34** | **33.33** | 22.38 | 22.37 |
| HHAR | 0 | **97.49** | **97.20** | 97.30 | 97.02 | 96.12 | 95.75 |
| | 1 | **97.27** | **96.89** | 96.97 | 96.58 | 95.23 | 94.84 |
| | 2 | 88.99 | 88.08 | **89.30** | **88.59** | 89.20 | 88.42 |
| | 3 | 59.22 | 55.80 | 59.74 | 57.32 | **61.43** | **59.82** |
| | 4 | **43.92** | **53.52** | 40.88 | 50.70 | 41.89 | 50.59 |
| UCIHAR | 0 | 75.50 | 76.80 | **78.38** | 80.45 | 78.09 | **80.78** |
| | 1 | 69.86 | 67.61 | **72.84** | **70.10** | 71.52 | 67.75 |
| | 2 | **90.61** | **90.49** | 89.44 | 88.97 | 89.73 | 89.49 |
| | 3 | **80.12** | **79.27** | 77.60 | 75.58 | 75.39 | 72.64 |
| | 4 | 90.06 | 90.35 | 90.06 | 89.88 | **92.38** | **92.56** |
| UniMiB-SHAR | 1 | 62.50 | 50.78 | **64.58** | **52.57** | 62.50 | 50.14 |
| | 2 | 59.86 | 47.92 | **60.72** | **55.72** | 60.54 | 51.65 |
| | 3 | 75.65 | 50.35 | 74.67 | 45.85 | 73.02 | 43.85 |
| | 5 | 42.61 | **29.52** | **44.29** | 28.04 | 41.94 | 29.09 |
| Opportunity | 0 | 82.86 | 56.23 | **83.82** | **58.06** | 83.53 | 55.42 |
| | 1 | 81.41 | 51.73 | 81.53 | 54.27 | **81.93** | **55.75** |
| | 2 | 74.14 | 36.99 | **76.32** | **39.95** | 74.52 | 36.25 |
| | 3 | **81.00** | 47.96 | 80.60 | **51.67** | 80.67 | 47.25 |

## C.6 TIME-DETERMINISTIC GAIN PHENOMENON FOR IRM

We observed that VREx performs the worst in this temporal OOD scenario. This is mainly because the principle behind IRM series methods is based on the assumption of domain feature invariance. Studies have shown that these methods typically fail when there is a significant marginal shift in the data itself.we selected representative IRM-related methods incorporated the Frequency Filter module. We use only the separated time-deterministic components to complete the OOD task. As shown in Table 12, we find that using time-deterministic features as a prior resulted in stable improvements across various IRM variants and it fits our theoretical insights in Appendix A. We verified that filtering Time-Deterministic features learnt with the help of IRM correlation yields smaller invariant domain distances than learning the original features directly by $\mathcal{A}$-distance in Table 11.

Table 11: $\mathcal{A}$-distance on IRM-based invariant features on DSADS.

| Dataset | Target | IRM | | IB-IRM | | VREx | | IIB | |
|---|---|---|---|---|---|---|---|---|---|
| | | full | $\alpha_{0.2/0.4}$ | full | $\alpha_{0.2/0.4}$ | full | $\alpha_{0.2/0.4}$ | full | $\alpha_{0.2/0.4}$ |
| DSADS | 0 | 0.8338 | **0.8235**↓ | 0.9329 | **0.8235**↓ | 0.8524 | **0.8235**↓ | 0.8833 | **0.8235**↓ |
| | 1 | 0.8482 | 0.8513 | 0.8679 | **0.8390**↓ | 0.86377 | **0.8307**↓ | 0.9195 | **0.8235**↓ |
| | 2 | 1.0319 | **0.8421**↓ | 0.9391 | 0.9473 | 0.8235 | 0.8431 | 1.0061 | **0.8338**↓ |
| | 3 | 0.8235 | 0.8534 | 0.9287 | **0.9102**↓ | 0.9752 | **0.8235**↓ | 0.8482 | **0.8235**↓ |
| | avg. | $0.8841_{\pm 0.09}$ | $\textbf{0.8425}_{\pm 0.01}$ | $0.9171_{\pm 0.03}$ | $\textbf{0.8800}_{\pm 0.05}$ | $0.8787_{\pm 0.06}$ | $\textbf{0.8302}_{\pm 0.01}$ | $0.9142_{\pm 0.06}$ | $\textbf{0.8260}_{\pm 0.01}$ |

Table 12: Time-Deterministic Enhancement for IRM-based variants.

| Dataset | Target | IRM full | α=20% | α=40% | IB-IRM full | α=20% | α=40% | VREx full | α=20% | α=40% | IIB full | α=20% | α=40% |
|---|---|---|---|---|---|---|---|---|---|---|---|---|---|
| Spurious Fourier | d=10% | 48.84 | 48.97↑ | **49.91**↑ | 49.66 | **51.34**↑ | 50.81↑ | 49.66 | **50.69**↑ | 47.38 | 50.34 | **51.34**↑ | 50.16 |
| HHAR | 0 | 90.14 | 88.39 | **90.33**↑ | 85.23 | 83.10 | **85.64**↑ | 89.60 | 94.40↑ | **95.28**↑ | 85.42 | 85.17 | **86.02**↑ |
|  | 1 | 89.41 | **91.57**↑ | 89.83↑ | 86.17 | 84.97 | **87.34**↑ | 91.00 | 93.70↑ | **95.02**↑ | 87.37 | **87.49**↑ | 87.25 |
|  | 2 | 88.06 | 90.29↑ | **90.65**↑ | 80.63 | **82.81**↑ | 81.88↑ | 84.58 | **90.03**↑ | 87.23↑ | 85.10 | **85.72**↑ | 84.79 |
|  | 3 | 65.95 | 61.02 | 62.47 | 61.60 | 62.88↑ | **63.92**↑ | 65.55 | 67.34↑ | **69.90**↑ | 61.77 | 64.10↑ | **64.62**↑ |
|  | 4 | 52.03 | **53.38**↑ | 53.38↑ | 42.57 | 43.24↑ | **43.58**↑ | 54.05 | 53.04 | **56.42**↑ | 48.99 | 49.66↑ | **53.04**↑ |
| UCIHAR | 0 | 93.95 | 95.97↑ | **97.69**↑ | 95.97 | **99.71**↑ | 97.98↑ | 89.34 | 96.83↑ | **98.85**↑ | 98.85 | 96.83 | 95.68 |
|  | 1 | 67.55 | **85.43**↑ | 61.92 | 61.26 | **74.89**↑ | 72.41↑ | 66.23 | **86.09**↑ | 62.25 | 79.14 | **87.75**↑ | 82.12↑ |
|  | 2 | 98.83 | **99.71**↑ | 98.83 | 98.83 | **99.71**↑ | 99.71↑ | 97.65 | **99.71**↑ | 98.53↑ | 99.41 | 96.48 | 95.60 |
|  | 3 | 95.27 | 94.64 | **97.79**↑ | 96.21 | 95.27 | **97.79**↑ | 83.28 | 85.43↑ | **86.38**↑ | 90.22 | 94.01↑ | **95.27**↑ |
|  | 4 | 89.40 | **98.01**↑ | 96.03↑ | 83.44 | 88.41↑ | **88.74**↑ | 70.86 | **89.34**↑ | 85.70↑ | 96.36 | **96.69**↑ | 89.07 |
| Opportunity | S1 | 61.36 | **65.99**↑ | 62.49↑ | 52.87 | 57.76↑ | **58.74**↑ | 53.73 | **55.12**↑ | 52.60 | 72.22 | **72.23**↑ | 52.17 |
|  | S2 | 43.31 | 44.38↑ | **44.43**↑ | 47.61 | **47.71**↑ | 45.84 | 37.17 | 47.15↑ | **50.76**↑ | 71.47 | 53.15 | **71.98**↑ |
|  | S3 | 35.17 | **39.06**↑ | 38.01↑ | 44.09 | 40.28 | 43.31 | 36.77 | **40.86**↑ | 40.41↑ | 71.47 | 33.44 | 37.29 |
|  | S4 | 46.07 | 52.08↑ | **54.10**↑ | 48.52 | 52.06↑ | **54.01**↑ | 46.41 | 53.48↑ | **59.78**↑ | 44.70 | **51.02**↑ | 44.37 |
| EMG | 0 | 72.77 | 70.60 | 71.65 | 69.31 | **71.48**↑ | 70.13↑ | 70.25 | 71.77↑ | **71.89**↑ | 64.85 | **68.19**↑ | 66.84↑ |
|  | 1 | 84.56 | 83.24 | **84.84**↑ | 86.44 | **87.76**↑ | 87.71↑ | 85.50 | 84.12 | 84.23 | 77.18 | **79.44**↑ | 76.79 |
|  | 2 | 77.75 | 77.81↑ | **78.95**↑ | 80.44 | 79.96 | **80.74**↑ | 75.36 | 77.51↑ | **78.47**↑ | 75.84 | 71.89 | 73.68 |
|  | 3 | 77.20 | 77.67↑ | **77.97**↑ | 80.09 | 79.92 | **80.57**↑ | 77.14 | 76.79 | **77.20**↑ | 62.08 | 71.35↑ | **74.42**↑ |
| DSADS | 0 | 83.63 | 83.56 | **85.21**↑ | 86.22 | 83.83 | 83.52 | 83.27 | 83.41↑ | 78.86 | 79.04 | **90.04**↑ | 88.46↑ |
|  | 1 | 71.77 | **75.98**↑ | 71.68 | 86.99 | **88.09**↑ | 86.38 | 75.56 | 73.80 | **75.83**↑ | 80.70 | **82.19**↑ | 81.27↑ |
|  | 2 | 83.54 | 89.83↑ | **90.45**↑ | 86.98 | 84.43 | **90.32**↑ | 88.27 | 87.56 | **90.86**↑ | 85.22 | 83.95 | **90.18**↑ |
|  | 3 | 78.57 | **83.36**↑ | 78.59↑ | 80.29 | **85.70**↑ | 84.12↑ | 77.81 | 83.16↑ | 82.81↑ | 74.61 | **75.48**↑ | 70.13 |
| PAMAP | 0 | 63.44 | **63.69**↑ | 62.47 | 60.04 | **63.00**↑ | 61.54↑ | 62.88 | **63.85**↑ | 62.93↑ | 63.21 | **63.66**↑ | 62.74 |
|  | 1 | 50.31 | **51.55**↑ | 51.35↑ | 49.81 | 50.26↑ | **51.26**↑ | 54.88 | **56.30**↑ | 54.52 | 57.39 | **63.66**↑ | 53.57 |
|  | 2 | 25.68 | 22.97 | 23.96 | 26.01 | **26.03**↑ | 25.21 | 22.68 | **24.98**↑ | 23.80↑ | 22.87 | 23.79↑ | **27.11**↑ |
|  | 3 | 62.62 | 62.68↑ | **63.71**↑ | 62.97 | **63.82**↑ | 60.22 | 62.10 | 62.15↑ | **62.49**↑ | 65.79 | 66.16↑ | **68.97**↑ |
| UniMiB-SHAR | 1 | 60.16 | 53.91 | 58.07 | 59.11 | 55.73 | 58.85 | 55.99 | 53.39 | **59.64**↑ | 60.42 | 55.47 | 57.81 |
|  | 2 | 61.92 | 43.40 | 52.66 | 63.64 | 48.54 | 57.12 | 57.80 | 45.11 | 53.69 | 62.09 | 49.74 | 52.14 |
|  | 3 | 69.41 | 60.86 | 67.11 | 66.45 | 54.61 | 59.87 | 63.16 | 60.20 | **67.11**↑ | 65.79 | 66.45↑ | **71.05**↑ |
|  | 5 | 36.91 | 36.58 | **38.26**↑ | 42.28 | 40.60 | **42.62**↑ | 41.28 | 38.93 | 40.60 | 46.64 | 44.63 | 44.30 |

## C.7 VISUALIZATION STUDY

We provide some t-SNE and FFT masking spectrum visualizations as shown in Figure 11 and 12. We have chosen several methods for comparison: FreTS and Diversify, one ablation version FEDNet w/o $L_{con}$, and FEDNet. From the results, it can be seen that our proposed FEDNet has a more compact potential representation and the division between different labels is clear. In addition, Diversify's division between labels is clear, but the representation is scattered, potentially causing confusion from redundant information and random noise.

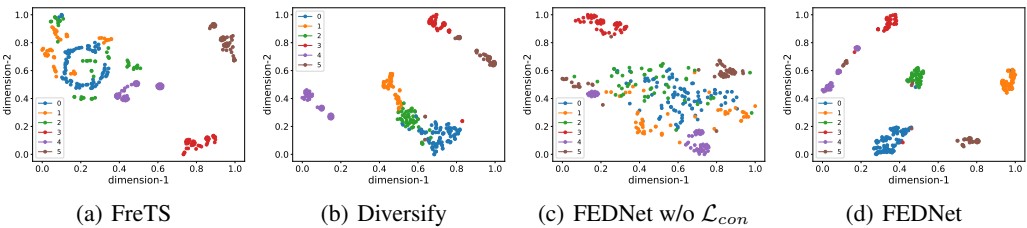

(a) FreTS     (b) Diversify     (c) FEDNet w/o $\mathcal{L}_{con}$     (d) FEDNet

Figure 11: t-SNE visualizations on the UCIHAR dataset with target domain 0. The X-axis represents the first dimension and the Y-axis represents the second dimension. The colors denote class labels.

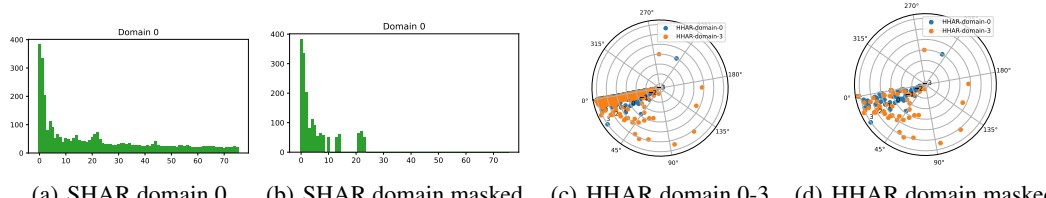

(a) SHAR domain 0    (b) SHAR domain masked    (c) HHAR domain 0-3    (d) HHAR domain masked

Figure 12: Figures (a) and (b) show the spectrograms before and after masking for domain 0 of the SHAR dataset. It can be observed that the retained frequencies are not exclusively low-frequency signals. Figures (c) and (d) display the polar plots before and after masking between domain 0 and domain 3 of the HHAR dataset. It can be seen that, through masking, the diversity between domains is more pronounced, while many redundant frequencies are eliminated.

Table 13: Performance on dataset UniMiB-SHAR with different $\alpha$ ratio level.

| Target | FFT | | | | DWT | | | |
|---|---|---|---|---|---|---|---|---|
| | 0.2 | 0.4 | 0.6 | 0.8 | 0.2 | 0.4 | 0.6 | 0.8 |
| 0 | 57.29 | **61.19** | 58.59 | 58.59 | 55.20 | 56.77 | **59.89** | 58.07 |
| 1 | **67.75** | 65.00 | 67.06 | 62.55 | 58.83 | **73.58** | 66.72 | 66.03 |
| 2 | 67.10 | **69.73** | 65.13 | 68.42 | **75.65** | 75.65 | 74.01 | 73.68 |
| 3 | 35.23 | **42.95** | 38.25 | 38.92 | 40.26 | **44.29** | 40.26 | 38.59 |

## C.8 MODEL COMPLEXITY

Given a time series data $L \times C$, where $L$ denotes its input length and $C$ represents the number of channels, FFT complexity is $O(L \log L)$. The time-deterministic complexity is $O(2 \times N \times L \times C \times C_{out} \times K)$ comes from the encoder and decoder, where $N$ is the number of hidden layers, $K$ is the kernel size of the 1D-Conv and $C_{out}$ is the output channels. The time-stochastic part uses patchify to reduce tokens in attention layer, with complexity $O(M^2)$ where $M = L/S \ll L$, $S$ is patch stride.

## C.9 FULL METRIC RESULTS

In order to more fully illustrate the effectiveness of our method, we provide results for other metrics in table 14, and it can be seen that our method achieves the best performance on most of the datasets.

## D IMPLEMENTATION DETAILS

### D.1 EXPERIMENTAL ENVIRONMENT

We implement FEDNet and the baselines based on WOODS(Gagnon-Audet et al., 2022) Benchmark and Time-Series-Library on a server equipped with an Intel(R) Xeon(R) Gold 5117 CPU and a Tesla V100 (32 GB) GPU, with 256 GB of memory. The server runs on Ubuntu 18.04 with CUDA 12.4, and the codes are implemented in PyTorch 2.3.0+cu121, Python 3.9.12. To reduce randomness, we conducted each experiment 3 times and reported the best results.

### D.2 COMPARISON METHOD INTRODUCTION

We list detail introduction of the methods we compared with FEDNet:

- IRM (Arjovsky et al., 2019) proposes a framework named Invariant Risk Minimization that aims to find representations where the optimal classifier remains invariant across different environments, thereby improving generalization to unseen domains.

Table 14: Macro-F1 on cross-person generalization.

| Dataset | Target | VREx | GroupDRO | ANDMask | FreTS | PatchTST | GILE | AdaRNN | Diversify | FEDNet$_f$ | FEDNet$_w$ |
|---|---|---|---|---|---|---|---|---|---|---|---|
| Spurious Fourier | d=10% | 32.52 | 32.73 | 32.93 | 11.15 | 50.32 | 11.03 | 33.39 | 15.37 | **74.56** | 33.33 |
| HHAR | 0 | 88.57 | 88.24 | 90.73 | 92.25 | 78.25 | 93.75 | 90.73 | 91.82 | 96.63 | **97.51** |
| | 1 | 90.31 | 86.66 | 91.87 | 93.00 | 78.60 | 96.57 | 68.58 | 91.99 | **97.96** | 96.89 |
| | 2 | 83.61 | 82.80 | 85.83 | 88.67 | 74.68 | 87.33 | 67.69 | 86.87 | 89.75 | **92.25** |
| | 3 | 60.32 | 64.07 | 61.13 | 58.42 | 36.53 | 62.67 | 46.30 | 51.67 | 61.53 | **65.94** |
| | 4 | 56.16 | 59.49 | 49.74 | 54.47 | 45.60 | 49.70 | 41.63 | 49.29 | **59.59** | 53.52 |
| UCIHAR | 0 | 89.23 | 91.70 | **98.73** | 84.01 | 67.52 | 86.44 | 85.29 | 79.67 | 95.62 | 80.45 |
| | 1 | 63.33 | 51.40 | 63.00 | 52.58 | 74.50 | 75.88 | 56.36 | 66.78 | **77.49** | 70.10 |
| | 2 | 97.51 | 96.81 | **97.85** | 92.10 | 66.59 | 97.31 | 75.05 | 92.39 | 97.80 | 90.49 |
| | 3 | 83.18 | 81.81 | 87.78 | 76.67 | 80.05 | 87.59 | 74.25 | 85.61 | **88.80** | 79.27 |
| | 4 | 68.91 | 60.30 | 89.75 | 88.89 | 77.65 | 92.89 | 65.23 | 92.18 | **98.35** | 90.35 |
| UniMiB-SHAR | 1 | 43.37 | 45.99 | 45.89 | 27.96 | 30.85 | 33.80 | 29.10 | 36.21 | 41.05 | **52.57** |
| | 2 | 51.37 | 53.55 | 52.55 | 32.41 | 54.49 | 47.09 | 17.75 | 28.66 | 55.97 | **61.87** |
| | 3 | 31.59 | 35.52 | 35.69 | 41.62 | 51.71 | 48.20 | 15.73 | 40.50 | 49.81 | **50.35** |
| | 5 | 23.70 | 24.74 | 23.56 | 23.10 | 24.85 | 31.82 | **34.02** | 20.10 | 22.39 | 28.04 |
| Opportunity | S1 | 39.98 | 40.16 | 28.67 | 55.41 | 27.18 | 60.76 | 47.48 | 53.75 | **63.22** | 58.06 |
| | S2 | 32.06 | 33.56 | 30.52 | 54.47 | 24.75 | **55.69** | 40.71 | 46.72 | 55.66 | 54.27 |
| | S3 | 26.53 | 32.15 | 21.79 | 41.20 | 23.51 | **45.64** | 31.58 | 34.39 | 42.35 | 39.95 |
| | S4 | 29.30 | 32.67 | 29.20 | 46.75 | 24.66 | 49.51 | 33.98 | 46.03 | **51.79** | 51.67 |
| EMG | 0 | 71.59 | 70.55 | 71.25 | 71.57 | 32.07 | 62.97 | 52.58 | 65.18 | **72.71** | 64.03 |
| | 1 | 82.99 | 82.71 | 82.10 | 80.16 | 35.66 | 67.97 | 56.39 | 78.61 | **87.06** | 59.57 |
| | 2 | 72.88 | 78.49 | 76.60 | 74.62 | 14.35 | 65.88 | 56.04 | 70.63 | **79.46** | 77.07 |
| | 3 | 77.41 | 78.85 | **79.87** | 78.26 | 36.14 | 70.02 | 52.76 | 76.50 | 77.69 | 79.80 |
| DSADS | 0 | 76.83 | 82.88 | 79.73 | 78.28 | 81.82 | 89.41 | 82.58 | 76.58 | **92.65** | 92.57 |
| | 1 | 77.91 | 76.51 | 74.48 | 69.76 | 73.32 | 78.26 | 79.74 | 77.29 | 82.74 | **83.73** |
| | 2 | 85.39 | 82.75 | 81.08 | 83.35 | 82.50 | 85.37 | 83.07 | 85.12 | **93.21** | 90.27 |
| | 3 | 73.33 | 74.93 | 77.58 | 72.36 | 78.28 | 77.38 | 69.54 | 70.04 | **87.64** | 80.52 |
| PAMAP | 0 | 45.22 | 48.18 | 47.49 | 48.79 | 48.66 | **55.52** | 53.61 | 53.78 | 55.43 | 53.26 |
| | 1 | 46.14 | 41.24 | 43.83 | 51.49 | 55.46 | 41.45 | 43.92 | 41.31 | **59.90** | 49.69 |
| | 2 | 22.70 | 26.21 | 27.71 | 35.50 | **49.65** | 26.52 | 23.13 | 25.40 | 35.34 | 37.15 |
| | 3 | 50.60 | 52.77 | 57.60 | **68.23** | 51.54 | 55.06 | 47.57 | 48.67 | 56.68 | 55.34 |

- IIB (Li et al., 2022) proposes a method to enhance domain generalization by extracting features with information bottleneck that are invariant across different domains, improving model robustness and performance.
- VREx (Krueger et al., 2021) introduces a method to enhance model robustness by penalizing the variance of risks across different environments, encouraging the model to perform consistently across diverse settings.
- IB-IRM (Ahuja et al., 2021) presents a method that combines the principles of empirical risk minimization with the information bottleneck approach to encourage models to focus on relevant features, enhancing robustness and generalization across different environments.
- GroupDRO (Sagawa* et al., 2020) is a method that seeks a global distribution with the worst performance within a range of the raw distribution for better generalization. Ours study the internal distribution shift instead of seeking a global distribution close to the original one.
- ANDMask (Parascandolo et al., 2021) is another gradient-based optimization method that belongs to special learning strategies. Ours focuses on representation learning.
- GILE (Qian et al., 2021) is a disentanglement method designed for cross-person human activity recognition. It is based on VAEs and requires domain labels.
- AdaRNN (Du et al., 2021) is a method with a two-stage that is non-differential and it is tailored for RNN. A specific algorithm is designed for splitting. Ours is universal and is differential with better performance.
- Diversify (Lu et al., 2022) is a time series OOD generalization method for dynamic distribution representation learning. It constructs a set of latent domain labels to better adapt to downstream tasks by employing a min-max adversarial approach to divide the original time series data distribution.

Table 15: Precision on cross-person generalization

| Dataset | Target | VREx | GroupDRO | ANDMask | FreTS | PatchTST | GILE | AdaRNN | Diversify | FEDNet$_f$ | FEDNet$_w$ |
|---|---|---|---|---|---|---|---|---|---|---|---|
| Spurious Fourier | d=10% | 24.33 | 24.33 | 24.55 | 11.17 | 50.32 | 11.03 | 25.06 | 15.36 | **74.56** | 33.34 |
| HHAR | 0 | 89.14 | 88.57 | 90.84 | 92.41 | 79.14 | 94.39 | 90.84 | 91.14 | 96.65 | **97.73** |
| | 1 | 90.21 | 86.79 | 91.76 | 92.99 | 78.44 | 96.45 | 77.17 | 91.55 | **97.98** | 97.32 |
| | 2 | 84.09 | 83.29 | 86.17 | 89.13 | 75.07 | 88.03 | 68.03 | 84.79 | 89.43 | **92.38** |
| | 3 | 65.41 | 65.45 | 61.90 | 63.63 | 36.48 | 65.93 | 46.46 | 54.32 | 62.82 | **69.86** |
| | 4 | 54.24 | 56.35 | 48.61 | 57.18 | 46.99 | 51.45 | 60.84 | 51.49 | 59.85 | 53.28 |
| UCIHAR | 0 | 91.19 | 93.04 | **98.53** | 87.24 | 78.13 | 89.70 | 89.29 | 80.78 | 95.36 | 80.89 |
| | 1 | 79.53 | 74.97 | 61.07 | 69.66 | 75.30 | 83.72 | 54.34 | 67.39 | **86.99** | 79.52 |
| | 2 | 97.86 | 97.24 | 97.82 | 92.47 | 71.86 | 97.31 | 80.91 | 93.14 | 97.80 | 90.79 |
| | 3 | 88.69 | 89.01 | 91.13 | 83.49 | 81.09 | 90.44 | 81.06 | 87.85 | 92.64 | 81.24 |
| | 4 | 82.45 | 62.54 | 91.01 | 90.26 | 80.89 | 94.07 | 68.94 | 93.27 | **98.36** | 91.53 |
| UniMiB-SHAR | 1 | 47.60 | 48.37 | 49.15 | 28.49 | 38.93 | 33.11 | 35.86 | 39.31 | 43.21 | **52.90** |
| | 2 | 54.57 | 57.70 | 57.22 | 35.72 | 54.74 | 46.52 | 18.06 | 32.61 | 57.58 | **63.03** |
| | 3 | 35.65 | 42.31 | 37.17 | 40.15 | 49.69 | 55.08 | 13.63 | 43.10 | **57.77** | 56.00 |
| | 5 | 24.03 | 24.70 | 24.21 | 24.84 | 23.53 | 29.04 | 17.85 | **36.41** | 25.63 | 23.83 |
| Opportunity | S1 | 34.96 | 34.87 | 39.85 | 63.66 | 23.50 | 68.60 | 58.21 | 59.25 | **70.79** | 68.09 |
| | S2 | 28.09 | 28.76 | 45.41 | 61.58 | 24.75 | 62.78 | 55.83 | 60.05 | 64.57 | **64.84** |
| | S3 | 24.86 | 30.01 | 26.85 | 52.73 | 20.60 | 55.80 | 43.54 | 47.11 | **56.41** | 51.36 |
| | S4 | 25.20 | 27.36 | 39.40 | 45.92 | 25.36 | 59.30 | 43.99 | 53.81 | **59.96** | 58.61 |
| EMG | 0 | 72.43 | 71.12 | 71.92 | 72.52 | 31.48 | 64.20 | 53.48 | 66.91 | **73.19** | 67.19 |
| | 1 | 84.37 | 84.61 | 84.21 | 81.49 | 35.34 | 69.04 | 56.88 | 80.67 | **87.31** | 60.34 |
| | 2 | 73.51 | 78.66 | 77.30 | 74.99 | 10.81 | 66.04 | 56.80 | 71.28 | **80.32** | 78.00 |
| | 3 | 78.27 | 79.48 | **80.05** | 78.95 | 37.74 | 70.56 | 54.84 | 76.79 | 79.39 | 78.43 |
| DSADS | 0 | 81.79 | 88.22 | 86.59 | 82.35 | 84.00 | 91.23 | 85.26 | 80.33 | 93.65 | **93.93** |
| | 1 | 87.21 | 82.31 | 82.16 | 77.73 | 76.07 | 84.72 | 83.58 | 83.51 | 85.51 | **88.57** |
| | 2 | 90.89 | 85.59 | 84.49 | 88.50 | 86.29 | 91.47 | 89.24 | 89.83 | 93.75 | 92.58 |
| | 3 | 79.48 | 81.53 | 81.54 | 80.42 | 81.81 | 85.30 | 75.67 | 77.72 | **88.67** | 83.35 |
| PAMAP | 0 | 44.60 | 49.92 | 47.62 | 53.45 | 53.78 | 56.25 | **59.30** | 57.37 | 58.89 | 57.37 |
| | 1 | 56.77 | 49.20 | 50.45 | 52.49 | 57.26 | 49.01 | 44.12 | 42.01 | **64.78** | 58.01 |
| | 2 | 25.36 | 29.64 | 30.35 | 33.46 | **49.55** | 30.11 | 27.57 | 26.73 | 42.35 | 39.23 |
| | 3 | 50.76 | 54.28 | 62.41 | **73.39** | 55.34 | 55.41 | 48.42 | 50.61 | 59.73 | 55.53 |

## D.3 EVALUATION

For UCIHAR, UniMiB-SHAR, and Opportunity, we directly utilized the open-source original datasets processed by GILE. To faithfully replicate the performance of GILE on these datasets, our experimental evaluation follows the same methodology as GILE. Specifically, for each domain, we treat it as the testing set while the remaining domains serve as the training set. We then select the model with the highest accuracy on the testing set.

For Spurious Fourier and HHAR, both are derived from the Benchmark dataset provided by WOODS (Gagnon-Audet et al., 2022). We adopt WOODS' default methods and domain partitioning as our baseline. This involves splitting 20% of the data in each domain for evaluation, using the evaluation data from the training domain as the validation set, and the evaluation data from the testing domain as the testing set. We calculate the average accuracy of each checkpoint on all validation sets across domains and save the results corresponding to the model with the highest average accuracy. The evaluation metric primarily relies on the Train-domain validation from WOODS, as it aligns with the evaluation methodology used in other datasets.

For DSADS, PAMAP, and we improved the architecture provided by WOODS and extended it to integrate these datasets for evaluation. The dataset partitioning method for evaluation follows the same approach as employed in Diversify, where only 20% of the training data is set aside as the validation set, and the remaining data from each domain is used as the testing set. We record the results on the testing set corresponding to the model that performed best on the validation set.

## D.4 HYPERPARAMETER SETTING

For hyper-parameter settings, if baselines provide hyper-parameters for the used datasets, we keep their default settings. Otherwise, we adjust the hyper-parameters to ensure a fair comparison as much as possible. For our proposed FEDNet, we leverage a 4-layer CNN with max pooling for the time-deterministic block and a 2-layer transformer encoder for the time-stochastic block, attention

Table 16: Recall on cross-person generalization.

| Dataset | Target | VREx | GroupDRO | ANDMask | FreTS | PatchTST | GILE | AdaRNN | Diversify | FEDNet$_f$ | FEDNet$_w$ |
|---|---|---|---|---|---|---|---|---|---|---|---|
| Spurious Fourier | d=10% | 50.00 | 50.00 | 50.00 | 11.15 | 50.32 | 11.03 | 50.00 | 15.38 | **74.56** | 33.33 |
| HHAR | 0 | 88.38 | 88.39 | 90.89 | 92.22 | 78.48 | 93.53 | 90.89 | 91.20 | 96.64 | **97.34** |
| | 1 | 90.79 | 86.96 | 92.08 | 93.33 | 79.21 | 96.75 | 71.20 | 91.54 | **97.94** | 96.57 |
| | 2 | 83.72 | 83.02 | 86.01 | 88.46 | 74.87 | 87.74 | 70.48 | 83.41 | 90.78 | **92.17** |
| | 3 | 64.30 | 63.99 | 61.40 | 59.73 | 38.71 | 62.29 | 48.36 | 53.42 | 61.86 | **66.79** |
| | 4 | 64.50 | 67.58 | 56.46 | 61.40 | 50.52 | 55.90 | 52.32 | 59.33 | **69.21** | 58.95 |
| UCIHAR | 0 | 88.65 | 92.02 | **98.96** | 85.37 | 76.12 | 88.14 | 85.53 | 80.01 | 96.66 | 80.35 |
| | 1 | 66.59 | 57.98 | 67.74 | 56.82 | 72.67 | 75.88 | 61.43 | 70.99 | **78.97** | 72.15 |
| | 2 | 97.35 | 96.79 | **97.94** | 91.92 | 70.84 | 97.30 | 75.49 | 92.26 | 97.81 | 90.35 |
| | 3 | 82.92 | 84.39 | 88.80 | 79.22 | 78.89 | 87.59 | 73.09 | 85.73 | **91.72** | 78.95 |
| | 4 | 72.84 | 67.48 | 90.15 | 89.53 | 80.18 | 92.89 | 69.62 | 92.05 | **98.39** | 90.61 |
| UniMiB-SHAR | 1 | 47.35 | 47.81 | 49.19 | 28.49 | 33.54 | 39.79 | 31.75 | 37.96 | 45.61 | **54.84** |
| | 2 | 55.12 | 56.41 | 56.63 | 36.32 | 60.54 | 52.94 | 25.34 | 40.04 | 60.39 | **64.89** |
| | 3 | 36.90 | 39.58 | 40.78 | 46.51 | 54.62 | 52.14 | 21.75 | 44.40 | 49.33 | **50.68** |
| | 5 | 30.95 | 33.12 | 31.57 | 29.50 | 31.54 | 38.41 | 19.97 | 40.91 | 27.31 | 37.19 |
| Opportunity | S1 | **59.88** | 56.63 | 25.86 | 50.77 | 39.82 | 55.96 | 43.19 | 50.60 | 58.95 | 52.03 |
| | S2 | 53.15 | 48.13 | 25.95 | 50.30 | 27.77 | **54.19** | 35.38 | 41.20 | 53.07 | 49.94 |
| | S3 | **43.82** | 42.81 | 19.74 | 38.82 | 33.52 | 42.28 | 29.24 | 34.73 | 38.31 | 36.80 |
| | S4 | 48.42 | 51.01 | 28.10 | 45.92 | 35.68 | 49.83 | 43.72 | 43.42 | 51.10 | **51.55** |
| EMG | 0 | 72.02 | 70.75 | 71.69 | 71.87 | 34.37 | 63.21 | 54.32 | 66.23 | **73.11** | 64.73 |
| | 1 | 83.28 | 83.19 | 82.61 | 80.25 | 36.94 | 68.26 | 57.59 | 79.13 | **87.17** | 59.98 |
| | 2 | 73.21 | 78.55 | 76.62 | 74.59 | 22.55 | 65.77 | 57.39 | 71.11 | **79.39** | 77.14 |
| | 3 | 77.16 | 78.64 | 79.86 | 78.02 | 36.46 | 70.12 | 53.82 | 76.43 | 77.54 | **79.97** |
| DSADS | 0 | 80.26 | 84.69 | 82.50 | 80.26 | 82.24 | 89.64 | 83.11 | 77.19 | **92.80** | 92.41 |
| | 1 | 76.54 | 76.58 | 73.42 | 70.13 | 74.07 | 78.20 | 79.78 | 77.28 | **84.86** | 83.64 |
| | 2 | 86.40 | 84.69 | 83.03 | 84.29 | 82.67 | 86.75 | 83.46 | 85.22 | **93.24** | 90.65 |
| | 3 | 74.61 | 76.62 | 78.46 | 73.46 | 78.85 | 79.56 | 70.35 | 71.80 | **87.71** | 80.52 |
| PAMAP | 0 | 49.10 | 52.46 | 52.46 | 51.64 | 48.71 | 58.56 | **58.59** | 57.79 | 57.85 | 56.66 |
| | 1 | 45.03 | 41.99 | 43.18 | 52.63 | 54.89 | 42.09 | 51.22 | 44.97 | **61.34** | 49.73 |
| | 2 | 27.03 | 30.12 | 30.72 | 41.72 | **55.83** | 30.26 | 25.94 | 27.10 | 36.56 | 40.06 |
| | 3 | 51.47 | 52.72 | 56.43 | **70.00** | 50.93 | 57.91 | 47.65 | 49.58 | 58.11 | 57.79 |

layer with 8 multi-head, patch length set to 16. We set the temperature for contrastive learning to 0.07 or 0.2.

For UCIHAR, UniMiB-SHAR, Opportunity, we follow the same settings with GILE The hidden dimension is set to 50 for UCIHAR and UniMiB-SHAR datasets, and 128 for Opportunity datasets as they are more complex.

We used the *Adaptive Moment Estimation* (Adam) optimizer for all our training processes, with learning rates primarily adjusted 1e-2 ~1e-5. The weight decay parameter is typically set to {0, 1e-5, 5e-4}. For the UCIHAR, UniMiB-SHAR, and Opportunity datasets, we employed the WeightedRandomSampler consistent with GILE to balance label distribution. For other datasets without specific requirements, we utilized the RandomSample method provided by WOODS (Gagnon-Audet et al., 2022) for class balancing strategies.

Table 17: Training hyperparameter settings for specific objective.

| Object | Hyperparameter | Value | Object | Hyperparameter | Value |
|---|---|---|---|---|---|
| VREx | penalty weight | 1e4 | Diversify | latent_domains | [5,10] |
| | annealing iterations | [500,1000,2000,4000] | | alpha | [0.1,1.0,10] |
| GroupDRO | $\eta$ | 1e-2 | ANDMask | $\tau$ | 1.0 |

# E  FURTHER DISCUSSION

**Finer-grained analysis with temporal stochastic components.** Our method effectively separates the influence of two types of shifts and models the relationship between time-stable modules and domain shifts well. We have verified that this approach can effectively mitigate the generalization performance of IRM methods under data conditions where the marginal probabilities within domains dynamically change over time. However, finer-grained research on the impact of temporal shift components still requires further investigation. From the current experimental results, better

Table 18: Model hyper-parameter settings.

| Objective | Dataset | Model | Model-parameter |
|---|---|---|---|
| VREx
GroupDRO
ANDMask | Spurious Fourier | LSTM | hidden_dept=3,
hidden_width=20,
recurrent_layers=2,
state_size=32 |
| VREx
GroupDRO
ANDMask | HHAR | Deep4Net | n_filters_time=32,
n_filters_spat=32,
n_filters = [64,128,256] |
| VREx
GroupDRO
ANDMask | DSADS
PAMAP
UCIHAR
UniMiB-SHAR
Opportunity | ActNetwork | kernel_size=(1,6)
bottleneck_dim=256
MaxPool2d_kernel_size=(1,2)
MaxPool2d_stride=2 |
| ERM | ALL | FreTS | embed_size=128, hidden_size=256 |
| ERM | ALL | PatchTST | patch_len=16, d_model = 128, n_heads = 4 |
| GILE | ALL | GILE | kernel_size=9, d_AE=50 |
| AdaRNN | ALL | AdaRNN | n_hiddens=[64,64], trans_loss='mmd' |
| Diversify | ALL | ActNetwrok | kernel_size=6, alpha1=1.0, alpha=1.0 |
| FEDNet | ALL | FEDNet | kernel_size=9, hidden_size=50,
patch_len=16, d_model=512, n_heads=8 |

modeling of dynamic changes is beneficial for improving model robustness. Currently, there are related works on large language models attempting to map the changes in time series to state tokens that describe temporal trends. In the future, we can use the results of temporal shift components as guideline to model trends between patches, capturing the patterns of different domains change over time.

**multi-domain datasets training phenomenon.** We observed a phenomenon in the training of multi-domain datasets, which mainly arises from differences in the training processes of diversify and GILE compared to WOODS. For the training processes of diversify and GILE on UCIHAR, UniMiB-SHAR, and Opportunity datasets, 20% of the data is randomly divided from the entire training data. This can lead to an imbalance in the number of different training domains. In contrast, WOODS uses a method where 20% is divided from each domain as a validation set, and the multi-domain data is simultaneously loaded for training. This training strategy is more balanced compared to the previous method, but it also reduces the model's performance because, after balancing each domain, the originally smaller number of samples becomes even fewer. We hope to discuss which of these two training strategies is more reasonable, or if there is a better evaluation, as this will be helpful for our subsequent research.

