# OpenReview forum: "FEDNET: FREQUENCY ENHANCED DECOMPOSED NETWORK FOR OUT-OF-DISTRIBUTION TIME SERIES CLASSIFICATION"
_ICLR.cc/2025/Conference — Submitted to ICLR 2025_

### Official Review · Reviewer_Fq1k · 2024-10-22

**Soundness:** 2
**Presentation:** 2
**Contribution:** 2
**Rating:** 5
**Confidence:** 2

**Summary:**

The paper proposes a method called FEDNet for out-of-distribution (OOD) time series classification. FEDNet extracts the information from frequency domain to decompose the time series into time-deterministic and time-stochastic components, and obtains the domain-invariant features for OOD data. By presenting both theoretical  and experimental insights, the authors claims the superiority of FEDNet for OOD time series classification.

However, the mathematical formulation (e.g., which references their definition relies on) is rather confusing, and it is hard to justify the authors' claim. But the results and their proposal are interesting.

**Strengths:**

(1) By specifying the notions of domain,  temporal, and frequency distribution shift, the paper addresses the issues of inconsistent distribution between time series from different domains and a solid analysis is provided for an insight into the improvement of OOD time series classification tasks.

(2) The experiments are conducted comprehensively along with well-illustrated insights.

(3) The proposed schemes of constraint loss for learning domain-invariant and domain-specific space is novel.

**Weaknesses:**

(1) It is not clear what are the shortcomings of the method. For example, to what degree of noises in time series can FEDNet can tolerate (still perform well)?

(2) Just suggestions typos: the first letter in line 239 should be capitalized.

(3) The ablation study only concerns the constraint loss, study on time-stochastic block is lacking. Would be nice to explain more empirically/theoretically on how time-stochastic block actually contributes the performance compared to the other model components.

**Questions:**

(1) What are the disadvantages of the method? It is not clear when the method will fail.

(2) How is the performance for general data instead of OOD data compared to existing SOTA methods?

(3) What is the time complexity for learning the time deterministic and stochastic features? e.g., space complexity and any potential trade-offs between computational requirements and performance gains compared to existing methods?

---

> ### Author Response · Authors · 2024-11-24
>
> **limitation and time stochastic study(W1, Q1, W3)**:
> We complement the time-stochastic feature study with a three-level(feature, data, and domain) analysis, discovered some of the problems that the model would face. The shortcomings of our approach mainly stem from two aspects; on the one hand, the results usually become poor when the domain invariant features are insufficient or corrupted by deletions. On the other hand, the over-reliance on the spectral distribution of the source domain leads to bias in the invariant features that may be learned. A detailed analysis can be found in the response of **Reviewer b2Zv time-stochastic study (W4)**.
>
> **writing error(W2)**:
> Thank you for your very careful review! we have fixed it.
>
> **comparison on general classification task(Q2)**
> Thanks for mentioning this interesting issue! We find that our method can be trained and predicted directly in generalized temporal classification and can get close to the sota level. We chose the methods PatchTST, Dlinear,iTransformer,FEDformer,TimesNet for comparison control, epochs = 100, and the rest of the parameters were generically not turned on. We found that this VAE form of modeling also achieves good results on regular datasets.
>
> |dataset |PatchTST| Dlinear| iTransformer | FEDformer | TimesNet| FEDNet|
> |---|---|---|---|---|---|---|
> |EthanolConcentration|28.13|30.41|25.85|31.21|$\underline{32.71}$|**34.60**|
> |Handwriting|28.19|27.87|28.31|28.01|$\underline{32.1}$|**36.94**|
> |Heartbeat|72.14|75.12|26.12|73.27|**78.30**|$\underline{75.61}$|
>
> By the way, we have added a lot of theoretical and experimental content in this new version, including the role of the two components, the effect of facing missing timing data, theoretical generalization upper bounds and frequency domain isolation levels, etc. We hope that these contents will be of interest to you as well :)

---

> > ### Comment · Reviewer_Fq1k · 2024-11-27
> >
> > How do you do the validation and choose the parameters, it looks like the results are overfitting

---

> > > ### Author Response · Authors · 2024-12-01
> > >
> > > ## Precison
> > > |Dataset| Target| VREx| GroupDRO| ANDMask| FreTS| PatchTST| GILE| AdaRNN| Diversify| FEDNet|
> > > |---|---|---|---|---|---|---|---|---|---|---|
> > > |UCIHAR|0|**0.9778±0.0078**|	0.9422±0.0321|	0.9404±0.0359|	0.8897±0.0194|	0.7278±0.0404|	0.9416±0.0294|	0.8834±0.0819|	0.8521±0.0155|$\underline{0.9506±0.0102}$|
> > > ||1|0.7752±0.0698|	0.6117±0.1414|	0.6601±0.0053|	0.563±0.0194|	0.7678±0.0094|	$\underline{0.8968±0.0079}$|	0.5188±0.0944|	0.7371±0.1052|	**0.9362±0.0057**|
> > > ||2|0.9748±0.0119|	0.9631±0.0185|	0.9723±0.0148|	0.8655±0.017|	0.6603±0.0153|	$\underline{0.9875±0.0026}$|	0.9351±0.0448|	0.8986±0.0008|	**0.9892±0.0024**|
> > > ||3|0.8711±0.045|	0.8831±0.002|	0.9139±0.0163|	0.782±0.0053|	0.816±0.0172|	0.8621±0.0655|	0.7837±0.0591|	$\underline{0.888±0.0473}$|	**0.9154±0.0228**|
> > > ||4|0.8505±0.0326|	0.6314±0.012|	0.7691±0.106|	0.8748±0.0099|	0.7295±0.0232|$\underline{0.9165±0.0198}$|	0.7651±0.144|	0.8876±0.0144|	**0.9331±0.0074**|
> > > |SHAR|0|0.497±0.0282	|0.4999±0.0293|	$\underline{0.5288±0.0313}$|	0.2944±0.0285|	0.36±0.0256|	0.4086±0.015|	0.3013±0.0875|	0.4375±0.0494|	**0.5471±0.0535**|
> > > ||1|0.4386±0.0959|	0.5164±0.0374|	0.5224±0.026|	0.3329±0.0147|	0.5067±0.0044|	$\underline{0.5548±0.0186}$|	0.2074±0.0619|	0.4256±0.0389|	**0.5802±0.0449**|
> > > ||2|0.3701±0.0394	|0.3817±0.0274	|0.3484±0.0058	|0.3186±0.0195|	**0.4982±0.0232**|	0.4228±0.0321|	0.2868±0.0225|	0.3837±0.0504|	$\underline{0.4235±0.0472}$|
> > > ||3|0.2237±0.0235|	0.2294±0.0161|	0.237±0.0109|	0.2032±0.0307|	0.2189±0.0303|	0.248±0.0182|	0.2344±0.0774|	**0.2964±0.0411**|	$\underline{0.2854±0.0162}$|
> > > ## Recall
> > > |Dataset| Target| VREx| GroupDRO| ANDMask| FreTS| PatchTST| GILE| AdaRNN| Diversify| FEDNet|
> > > |---|---|---|---|---|---|---|---|---|---|---|
> > > |UCIHAR|0|**0.969±0.0136**|	0.9263±0.0416|	0.9243±0.0649|	0.8267±0.0195|	0.7183±0.0388|	0.9406±0.0251|	0.8847±0.0682|0.8197±0.0149| $\underline{0.9446±0.007}$|
> > > ||1|0.701±0.0802|	0.5961±0.0249|	0.6981±0.0679|	0.5076±0.0093|	0.7418±0.0319|	$\underline{0.8266±0.0094}$	|0.5351±0.0411	|0.6766±0.0678|	**0.9091±0.031**|
> > > ||2|0.9703±0.014|	0.9549±0.0234|	0.966±0.0179|	0.8562±0.0177|	0.6533±0.0248|	$\underline{0.9851±0.0038}$|	0.9192±0.059	|0.8777±0.0199	|**0.9876±0.0037**|
> > > ||3|0.8636±0.0519|	0.8644±0.0264|	$\underline{0.8815±0.0215}$|	0.7244±0.0145|	0.7929±0.0263|	0.8454±0.0915|	0.7056±0.0603|	0.8724±0.0666|	**0.911±0.0172**|
> > > ||4|0.8247±0.041|	0.6434±0.0157|	0.7228±0.0161|	0.8382±0.0222|	0.7099±0.0024|	$\underline{0.8956±0.0461}$|	0.681±0.0624|	0.8451±0.0365	|**0.9182±0.0064**|
> > > |SHAR|0|0.4644±0.0355	|0.4596±0.0341|	$\underline{0.4724±0.024}$	|0.3019±0.0193|	0.3443±0.018|	0.4188±0.0313|	0.2907±0.0389|	0.4339±0.0411|	**0.5431±0.0461**|
> > > ||1|0.4608±0.0912|	0.5146±0.0106	|0.4888±0.0159|	0.3443±0.0227	|0.5064±0.0257	|$\underline{0.5708±0.03}$	|0.2238±0.0236	|0.4553±0.0374	|**0.5748±0.0313**|
> > > ||2|0.3588±0.0218|	0.3603±0.026|	0.3565±0.0157	|0.3208±0.0545|	**0.4654±0.0412**|	0.4064±0.012	|0.2801±0.0538|	0.3904±0.0079|	$\underline{0.4173±0.0296}$|
> > > ||3|0.2865±0.0215|	0.3112±0.0173	|0.2989±0.0413|	0.2081±0.029	|0.2889±0.0212	|$\underline{0.3175±0.0141}$|	0.2658±0.053	|0.2942±0.0339|	**0.3192±0.0246**|

---

> ### Author Response · Authors · 2024-11-24
>
> **model complexity analysis(Q3)**
> We have provide model commplexity in appendix D.8 MODEL COMPLEXITY.
> > Given a time series data $L \times C$, where $L$ donates its input length and $C$ represents the number of channels, FFT complexity is $O(L\log L)$. The time-deterministic complexity is $O(2 \times N \times L \times C \times C_{out} \times K)$ comes from the encoder and decoder, where $N$ is the number of hidden layers, $K$ is the kernel size of the 1D-Conv and $C_{out}$ is the output channels. The time-stochastic part uses patchify to reduce tokens in attention layer, with complexity $O(M^2)$ where $M = L/S \ll L$, $S$ is patch stride.
>
> 1. **We have performed some lightweight MLP Encoder processing time-deterministic,it is feasible.**
>
> |Dataset|Target|FEDNet-CNNEncoder|FEDNet-MLPEncoder|
> |---|---|---|---|
> |UCIHAR|0|94.81|**95.62**|
> ||1|80.13|**81.81**|
> ||2|**97.95**|94.49|
> ||3|91.79|**92.81**|
> ||4|98.35|**98.68**|
>
> |Method|Complexity|
> |---|---|
> |FEDNet-CNNEncoder|$O(2 \times N \times L \times C \times C_{out} \times K)$|
> |FEDNet-MLPEncoder|$O(2 \times N \times L \times C \times C_{out})$|
>
> 2. **from time-stochastic study, we found unfold patches for time-stochastic featrues achieves the fast speed for self-attention and reduce $x_{patch}$ condition number.** P donates patch length while S represents stride.
>
> |dataset|S=P/2|S=P|
> |---|---|---|
> |HHAR|0.34075 s/iter|**0.2321** s/iter|
> |UCIHAR|0.4724 s/iter|**0.3672** s/iter|
> |UniMiB-SHAR|0.3685 s/iter|**0.3157** s/iter|
> |OPP| 0.5676 s/iter|**0.4272** s/iter|
> |EMG|0.2785 s/iter|**0.1951** s/iter|
> |DSADS|0.8803 s/iter|**0.3177** s/iter|
> |PAMAP|0.5272 s/iter|**0.3276** s/iter|
> |EEG|0.8316 s/iter|**0.6683** s/iter|
>
>
>
> 3. **we also provide different time-stochastic encoders like MLP and depthwise separable convolution on EEG target domain-2.**
>
> | Method   | Accuracy | F1-score | Precision | Recall |
> |----------|----------|----------|-----------|--------|
> | FEDNet w/o $L_{sto}$   | 69.99 |57.97| **63.26**| 59.31|
> |FEDNet+ self-attention| **72.90**| 62.16| 61.74| 63.01|
> |FEDNet+ DwConv| 70.21| 57.78| 56.33| 62.09|
> |FEDNet+ MLP| 71.85| **62.94**| 61.80| **65.19**|

---

> ### Comment · Reviewer_Fq1k · 2024-11-27
>
> Thanks for your clarification.
>
> It seems like the Definition 2 is not mathematically rigorous, and not practical, do you have any related work built on this definition? maybe it is good to clarify more on how you made this definition.

---

> > ### Author Response · Authors · 2024-11-28
> > **Benifts from Frequency**
> >
> > Dear Reviewer Fq1k
> > we clearify more details about the benefits of the frequency information and our theory contributions, which is also related to your question and you can be seen in the comments Frequency perspective Benifts with **Reviewer b2Zv**.

---

> ### Author Response · Authors · 2024-11-27
>
> Thank you for your response!
> 1. For General time series classification("EthanolConcentration", "Handwriting", "Heartbeat") and UCIHAR,SHAR,Opportinity, we follows the origin paper settings[1], they choose validation and testing the same and get the best results, you can check the code in `Time-Series-Library`.
>
> 2. For the rest 6 dataset, we keep 6:2:2 spilting settings and we save the best validation results model for testings follows the origin paper settings in diversify[2], and our code uses the same dataset and dataloader of their implementation added into Time-Series-Library Framework.
>
> 3. For domain dividing, we follows the origin code implementations without change anything.  All the datasets we use are open source, only PAMAP is reproduced by us as they do not have an open-source version may have some different, we provided all the links in our code README.
>
> 3. For some important hyperparameters, such as temperature we set 0.07 or 0.2, and for $\alpha$ we test with $ [0.2, 0.4, 0.6, 0.8]$, we mainly tested 0.2 and 0.4, the latter 0.6 and 0.8 due to time constraints did not test to cover all datasets, and then some individual dataset domain we tried 0.5 and 0.1, for $\lambda_{det}$ and $\lambda_{sto}$ we set to default 1.0, and we have tested it with $[0.1, 1.0, 10.0]$ it doesn't change much. The other model's parameters we just followed the default code settings in the benchmark, we didn't change them.
>
> We have give all the detail settings in the appendix **D IMPLEMENTATION DETAILS** and all the settings and evaluation are follow the origin papers and code, we don't change it.
>
> From the results you can see that some datasets in point1 may face your concern and their performance gap between different methods is smaller than the rest dataset, especially on the performance of contrastive learning.
>
> Has your confusion been resolved? If you have any questions or suggestions, we will continue to add the experiment. We still have one week to discuss :)
>
> [1] Wu H, Hu T, Liu Y, et al. Timesnet: Temporal 2d-variation modeling for general time series analysis[J]. arXiv preprint arXiv:2210.02186, 2022.
>
> [2] Lu W, Wang J, Sun X, et al. Out-of-distribution representation learning for time series classification[J]. arXiv preprint arXiv:2209.07027, 2022.

---

> ### Author Response · Authors · 2024-11-27
> **Basic Definition of Domain Generalization**
>
> We apologize for the confusion caused by the introduction of too many concepts in the new commit, in fact, the concepts we mentioned are the basic concepts within domain generalization.
>
> To further clear your confusion, we will give you a detail introduction about domain generalization.
>
> 1. **Domain** [1]
> > Let $\mathcal{X}$ denote a nonempty input space and $\mathcal{Y}$ an output space. A domain is composed of data that are sampled from a distribution. We denote it as $D = [(x_i, y_i)] \sim P_{XY}$ contains $n$ samples, where $x \in \mathcal{X} \subseteq \mathbb{R}^d$, $y \in \mathcal{Y} \subseteq \mathbb{R}$ denotes the label, and $P_{XY}$ denotes the joint distribution of the input sample and output label. $X$ and $Y$ denote the corresponding random variables.
>
> 2. **Covariate Shift Assumption** [2]
>
> > Given the training andthe test data from two distributions $P_{XY1}(x, y)$, $P_{XY2}(x, y)$, covariate shift is referred to the case that the marginal probability distributions are different, and the conditional distributions are the same, i.e.,$P_{XY1}(x) \neq P_{XY2}(x)$, and $P_{XY1}(y|x)= P_{XY2}(y|x)$.
>
> 3. **Domain Adaptation (DA)** [3]
>
> Before we get into domain generalization, we'd like to introduce you to a simplified version of it, which is one of the earliest types of OOD problems you'll have heard of. DA aims to maximize the performance on a given target domain using existing training source domain(s).
>
> DA problems will usually only have 2 domains and hold on the **Covariate Shift Assumption** i.e. $P_{source}(X) \neq P_{target}(X), P_{source}(y|x)= P_{target}(y|x)$. **Most importantly, it allows you to use the input data $X_{test}$ from the test set to align.**
>
> 4. **Domain Generalization (DG)** [1]
>
> > In domain generalization task, we are given $M$ training (source) domains $D_{train} = \{ D^i \mid i = 1, \cdots, M \}$, where $D^i = [(x_j^i, y_j^i)]$ denotes the $i$-th domain contains $j=[1 ... n_i]$ samples. The joint distributions between each pair of domains are different: $P_{XY}^i \neq P_{XY}^j, \quad 1 \leq i \neq j \leq M.$ (usually hold on the **Covariate Shift Assumption** $P_{X}^i \neq P_{X}^j, P_{Y|X}^i = P_{Y|X}^j$ ) The goal of domain generalization is to learn a robust and generalizable predictive function $h : \mathcal{X} \to \mathcal{Y}$ from the $M$ training domains to achieve a minimum prediction error on an unseen test domain $D_{test}$
>
> $$
> \min_h \mathbb{E}_{(x, y)}[\ell(h(x), y)], (x,y) \in Dtest
> $$
>
> where $\mathbb{E}$ is the expectation and $\ell(·, ·)$ is the loss function. **In addition, we can not access to the test data $D_{test}$ which is different from DA.**
>
> 5. **Temporal Covariate Shift** [4]
> > Given a time series data $D$ with $n$ labeled segments. Suppose it can be split into $K$ periods or intervals,
> $D = [D_1, \cdots, D_K]$, where $D_k = [(x_i, y_i)]$, where $i=n_{k}+1, \cdots n_{k+1}$ , and $n_1 = 0$ and $n_{k+1} = n$. Temporal Covariate Shift (TCS) is referred to the case that all the segments in the same period $i$ follow the same data distribution $P_{D_i}(x, y)$, while for different time periods $1 \leq i \neq j \leq K$, $ P_{D_i}(x) \neq P_{D_j}(x) \quad \text{and} \quad P_{D_i}(y \mid x) = P_{D_j}(y \mid x)$.
>
> **This is in fact the same as our definition of Temporal Distribution Shift**, which is also followed in the latest paper Diversify[5], but none of them gives such a detail definition as ours, either in the time series or in the frequency domain.
>
> And we would like to emphasize that in both **Adarnn** [4] and **diversify** [5] it is in fact impractical to address the problem of shifting time distributions by using $K$ as a hyperparameter for modeling the shifts to be possible categories, while **in our definition we define $K$ as unpredictable** to explore more essentially the impact of such shifts on the time series.
>
> The formulas have been slightly adjusted due to some issues with markdown syntax display, and the full formulas can be found directly from the original references.
>
> [1] J. Wang et al., "Generalizing to Unseen Domains: A Survey on Domain Generalization," in IEEE Transactions on Knowledge and Data Engineering, vol. 35, no. 8, pp. 8052-8072, 1 Aug. 2023, doi: 10.1109/TKDE.2022.3178128.
>
> [2] Shimodaira H. Improving predictive inference under covariate shift by weighting the log-likelihood function[J]. Journal of statistical planning and inference, 2000, 90(2): 227-244.
>
> [3] Wang M, Deng W. Deep visual domain adaptation: A survey[J]. Neurocomputing, 2018, 312: 135-153.
>
> [4] Du Y, Wang J, Feng W, et al. Adarnn: Adaptive learning and forecasting of time series[C]//Proceedings of the 30th ACM international conference on information & knowledge management. 2021: 402-411.
>
> [5] Lu W, Wang J, Sun X, et al. Out-of-distribution representation learning for time series classification[J]. ICLR 2023.

---

> ### Author Response · Authors · 2024-11-27
> **Domain Adaptation Error Bound**
>
> These upper bound definitions are also all based on the **Covariate Shift Assumption**.
>
> ### Domain Adaptation Error Bound
>
> **Theorem 1 Domain adaptation error bound** [1]
> > Let $d$ be the Vapnik–Chervonenkis (VC) dimension [2] of $\mathcal{H}$, and $D^s$ and $D^t$ be unlabeled samples of size $n$ from the two domains. Then for any $h \in \mathcal{H}$ and $\delta \in (0, 1)$, the following inequality holds with probability at least $1 - \delta$:
>
> $$
> \epsilon^t(h) \leq \epsilon^s(h) + \hat{d}_{\mathcal{H}\Delta\mathcal{H}}(D^s, D^t) + \lambda_H + 4 \sqrt{\frac{2d \log(2n) + \log(2/\delta)}{n}}
> $$
>
> where $\hat{d}_{\mathcal{H}\Delta\mathcal{H}}(D^s, D^t)$ is the estimate of
>
> $d_{\mathcal{H}\Delta\mathcal{H}}(P_X^s, P_X^t)$ on the two sets of finite data samples.  $\epsilon^s(h)$ denotes ideal target risk and $\lambda_H$ denotes ideal domain divergence, they are both unpredictable.
>
> **In fact, $\hat{d}_{\mathcal{H}\Delta\mathcal{H}}(D^s, D^t)$ is the same measture $\mathcal{A}$-distance often used for estimate domain divergence, so that the smaller the distance is, the risk bound will be smaller.** We also use it in our experiments in **Invirant Feature Study**.
>
> **Theorem 2 PAC-Bayesian Perspective on Domain Adaptation** [3]
> > Let $\mathcal{H}$ be a hypothesis space, let $\mathcal{S}$ and $\mathcal{T}$ respectively be the source and the target domains on $\mathcal{X} \times \mathcal{Y}$. Let $q > 0$ be a constant. We have, for all $\rho$ on $\mathcal{H}$,
>
> $$
> \mathcal{R}_{\mathcal{T}}(G(\rho))  \leq \frac{1}{2} d_T(\rho) + \beta_q(\mathcal{T} \| \mathcal{S}) \times [e_S(\rho) ]^{1 - \frac{1}{q}} + \eta,
> $$
>
> where $d_{\mathcal{T}}(\rho)$ is denote to ideal target risk (unpredictable), $e_{\mathcal{S}}(\rho)$ is empirical source risk, $\beta_q(\mathcal{T} \| \mathcal{S})$ denote the domain divergence, and $\eta$ estimate the joint error when $SUPP(\mathcal{T})$  is not entirely included in $SUPP(\mathcal{S})$.
>
> [1] Ben-David S, Blitzer J, Crammer K, et al. A theory of learning from different domains[J]. Machine learning, 2010, 79: 151-175.
>
> [2] Vapnik V, Levin E, Le Cun Y. Measuring the VC-dimension of a learning machine[J]. Neural computation, 1994, 6(5): 851-876.
>
> [3] Germain P, Habrard A, Laviolette F, et al. A new PAC-Bayesian perspective on domain adaptation[C]//International conference on machine learning. PMLR, 2016: 859-868.

---

> ### Author Response · Authors · 2024-11-27
> **Domain Generalization Error Bound**
>
> Domain generalization Error Bound itself are all conclusions of the extension of the domain adaptive upper bound in the case of multiple domains.
>
> **Theorem 3 Domain generalization error bound** [1]
> > Let $\gamma := \min_{\pi \in \Delta_M} d_{\mathcal{H}\Delta\mathcal{H}}\left(P_X^t, \sum_{i=1}^M \pi_i P_X^i\right)$ with minimizer $\pi^*$ be the distance of $P_X^t$ from the convex hull $\Lambda$, and $ P_X^* := \sum_{i=1}^M \pi_i^* P_X^i$ be the best approximator within $\Lambda$. Let $ \rho := \sup_{P_X', P_X'' \in \Lambda} d_{\mathcal{H}\Delta\mathcal{H}}(P_X', P_X'')$ be the diameter of $\Lambda$.
>
> Then it holds that
> $$
> \epsilon^t(h) \leq \sum_{i=1}^M \pi_i^* \epsilon^i(h) + \frac{\gamma + \rho}{2} + \lambda_\mathcal{H}(P_X^t, P_X^*),
> $$
>
> where $\lambda_\mathcal{H}(P_X^t, P_X^*)$ is the ideal joint risk across the target domain and the domain with the best approximator distribution $P_X^*$.
>
> **Theorem 4 PAC-Bayesian Domain generalization error bound** [2]
> > Let $\mathcal{H}$ be a hypothesis space built from a set of source domains, denoted as $D = \{D_i\}_{i=1}^{N_d}$.
>
> Suppose $q > 0$ is a constant, for any unseen domain $D_{U}$ from the convex hull $\Lambda_D$,
>
> $$
> R_{D_U}[h] \leq \frac{1}{2} d_{D_U}(h) + \epsilon \cdot \left[ e_{D_{\bar{U}}} (h) \right]^{1-\frac{1}{q}} + \eta_{T/S},
> $$
>
> where $d_D(h)$ and $e_D(h)$ are ideal and expected risk of a domain $D$ respectively, $\epsilon = \beta_{q}(D_U \| \sum_{i=1}^{N_d} \pi_i D_i)$ is an ideal distance since we can't have access to $D_U$, while $\eta_{T/S}$ denotes the distribution of $(x, y) \in \text{SUPP}(Test) \backslash \text{SUPP}(Source)$, it is usually a small value.
> Suppose $(x, y)$ between the unseen domain for testing and source doamins have been fully covered by $\Lambda_S$, then $\eta_{T/S} = 0$ and there exists a finite upper bound $\rho = \sup_{i,j \in [N_d]} \beta_q(D_i \| D_j), \forall q > 0$ for any convex combinatorial domains $D_i$ and $D_j$.
>
>
> **We introduced the PAC-Bayesian error bound to justify the conclusion because here $\beta_q(D_i \| D_j)$ can be estimated using the marginal probabilities P(X) in any two domains, such as $D_i$ and $D_j$, or $D_{k}^i$ and $D_{k}^j$, while the original method can only be be tested by feature and cannot be linked to the frequency domain or any data prior. I hope this gives you a better understanding and contribution of Proposition 5.2 Frequency Perspective Risk Bound on Unseen Time series Domain**.
>
> [1] Albuquerque I, Monteiro J, Falk T H, et al. Adversarial target-invariant representation learning for domain generalization[J]. CoRR, 2019.
>
> [2] Germain P, Habrard A, Laviolette F, et al. A new PAC-Bayesian perspective on domain adaptation[C]//International conference on machine learning. PMLR, 2016: 859-868.

---

> > ### Author Response · Authors · 2024-11-27
> > **Motivation**
> >
> > **Finally, I would like to emphasize to you the motivation for our paper, that we are addressing a more realistic problem, which is that we cannot get the ideal domain distribution, and that the internal data of well-divided domains still suffers from inconsistent data distributions.**
> >
> > This is not just a problem that exists with time series, although our current work is thinking about the non-stationarity of time series in terms of domain generalization. In contrast to the recent methods we mentioned, we think in a completely different way; instead of assuming that the data may follow multiple distributions, we try to eliminate this effect from analyzing the a priori components of the data itself, and purify to get a more robust characterization of domain invariance.
> >
> > **The reason why specific distributions cannot be given is that we do not want to make certain assumptions about the data distribution, which is consistent and canonical with all these theoretical articles on domain generalization that we mentioned.**

---

> ### Author Response · Authors · 2024-11-27
> **Thank you for the review! Have we clearly addressed the concerns?**
>
> Thank you very much for your support of the theoretical part of our work, we sincerely hope that these replies can let you regain confidence in our work. If you still have any questions, you do not hesitate to tell us, if it is still on the experimental aspects of the existence of some doubts, we will do our best to add all the results of the experiments in the next period of time, and we still have time to modify the pdf tomorrow, if you If you think these theoretical contents need to be added into the pdf, we will add them in time, your advice is really important to us and wish you a happy Thanksgiving!

---

> ### Author Response · Authors · 2024-11-28
> **About Experiment concerns**
>
> our previous setup on UCIHAR, SHAR, and Opportunity was mainly to ensure a consistent way of comparison with the original paper. To further address your confusion regarding some of the experimental results, we have currently performed a 6:2:2 data split on the UCIHAR, SHAR, and Opportunity datasets, and adapted them into the same framework as Diversify. We will provide you with these experimental results here, hoping to address your concerns about these datasets. We will provide you with feedback as soon as we have the complete experimental results.

---

> ### Author Response · Authors · 2024-12-01
> **Experiment concerns**
>
> Dear Fq1k,
> To address your concerns in experiments, we have also split the UCIHAR, SHAR, Opportunity dataset into 6:2:2  for evaluation and have already updated our code and scripts for them named UCIHAR2, SHAR2, OPP2. The conclusion is same as before. Our model achieved always shows the best performance across the domains and ERM methods like FreTS and Patchtst performance are usually worse than ood based methods unlike before.
> Here we list the experiment parameters we tested for all the methods.
> - $learning_rate: 1e-2, 1e-3, 1e-4 $
> - $seed: 2023, 2024, 2025$
> - $\alpha: 0.2, 0.4, 0.6$
> - model parameters we didn't change.
>
> ## Accuracy
> |Dataset| Target| VREx| GroupDRO| ANDMask| FreTS| PatchTST| GILE| AdaRNN| Diversify| FEDNet|
> |---|---|---|---|---|---|---|---|---|---|---|
> |UCIHAR|0|**0.9731±0.0117**|	0.9299±0.0391|	0.927±0.0541|	0.8116±0.0258|	0.7377±0.04|	0.9337±0.0403|	0.901±0.1088|	0.8136±0.0486|	$\underline{0.951±0.0058}$|
> ||1|0.7064±0.0913|	0.5938±0.0301|	0.7042±0.0822|	0.5121±0.0101|	0.7483±0.0271|	$\underline{0.8399±0.0083}$|	0.5287±0.0384|	0.6766±0.0712|	**0.9073±0.0201**|
> ||2|0.9726±0.0122|	0.9551±0.0235|	0.9716±0.0151|	0.8582±0.0167|0.65±0.0249|	$\underline{0.9863±0.0033}$|	0.9247±0.0554|	0.8807±0.0212|	**0.9882±0.003**|
> ||3|0.8612±0.0501|	0.8539±0.0292|	0.8759±0.0215|	0.7234±0.0142|	0.7959±0.0268|	$\underline{0.8927±0.0054}$|	0.695±0.0575|	0.8265±0.0728|	**0.9072±0.0139**|
> ||4|0.8311±0.0422|	0.6402±0.0315|	0.7086±0.0119|	0.8344±0.0172|	0.6832±0.0239|	0.8575±0.0201|	0.7174±0.0978|	$\underline{0.8653±0.0134}$|	**0.9139±0.0088**|
> |SHAR|0|0.5851±0.0148|	0.5625±0.0069|	$\underline{0.5929±0.004}$|	0.375±0.0232|	0.4357±0.004|	0.5078±0.0158|	0.4323±0.0394|	0.5538±0.0266|	**0.5955±0.0007**|
> ||1|0.5689±0.0167|	0.6055±0.0211|	0.5792±0.0275|	0.3012±0.0189|	0.5849±0.0371|	**0.6523±0.0378**|	0.2499±0.0156|	0.5049±0.0335|	$\underline{0.6249±0.0173}$|
> ||2|0.6152±0.0143|	0.6426±0.0219|	0.6151±0.0206|	0.4988±0.0449|	$\underline{0.6776±0.0634}$|	0.6502±0.0342|	0.386±0.0554|	0.6404±0.0095|	**0.6798±0.0106**|
> ||3|0.396±0.0134|	0.4139±0.0151|	0.4083±0.0305|	0.3087±0.0439|	0.3735±0.026|	0.3915±0.0547|	0.3289±0.0484|	$\underline{0.4161±0.0269}$|	**0.4239±0.0118**|
>
> ## F1-score
> |Dataset| Target| VREx| GroupDRO| ANDMask| FreTS| PatchTST| GILE| AdaRNN| Diversify| FEDNet|
> |---|---|---|---|---|---|---|---|---|---|---|
> |UCIHAR|0|**0.9722±0.0114**|	0.9276±0.0405|	0.9265±0.0566|	0.814±0.0199|	0.7135±0.0413|	0.9327±0.0323|	0.8669±0.0882|	0.8108±0.0288|	$\underline{0.9407±0.0086}$|
> ||1|0.683±0.085|	0.5273±0.0524|	0.6307±0.0943|	0.452±0.0071|	0.7443±0.0292|	$\underline{0.8123±0.0108}$|	0.4537±0.0458|	0.6374±0.0689|	**0.8983±0.0326**|
> ||2|0.9792±0.0046|	0.9671±0.0186|	0.9734±0.0201|	0.8566±0.0235|	0.6546±0.0206|	$\underline{0.9849±0.0043}$|	0.9456±0.055|	0.8697±0.0265|	**0.9865±0.0023**|
> ||3|0.8629±0.0489|	0.8593±0.0241|	0.8683±0.0247|	0.6926±0.0226|	0.7926±0.0272|	$\underline{0.8926±0.0054}$|	0.6827±0.0601|	0.8261±0.0718|	**0.9067±0.0129**|
> ||4|0.7651±0.0819|	0.5736±0.0209	|0.6644±0.0223	|0.8258±0.0184	|0.6702±0.0294	|0.8501±0.0204|	0.6768±0.1245|	$\underline{0.8659±0.0135}$|	**0.9120±0.0087**|
> |SHAR|0|0.4583±0.0277|	0.45±0.0318|	$\underline{0.4706±0.026}$|	0.2622±0.0166|	0.3166±0.013|	0.385±0.0139|	0.2702±0.0503|	0.4121±0.0407|	**0.5155±0.0332**|
> ||1|0.4775±0.0277|	0.4796±0.0165|	0.4499±0.0188|	0.3063±0.0147|	0.4693±0.0156|	$\underline{0.5327±0.0231}$|	0.1749±0.0245|	0.4107±0.0304	|**0.5365±0.0357**|
> ||2|0.3096±0.006|	0.336±0.0302|	0.3147±0.0112|	0.2992±0.0362|	**0.4479±0.0203**|	0.4035±0.0155| 0.2561±0.03|	0.3515±0.0434|	$\underline{0.4048±0.0334}$|
> ||3|0.2203±0.0161|	0.233±0.0092|	0.2338±0.026|	0.1661±0.0294|	0.2229±0.0112|	0.2358±0.0124|	0.2328±0.0548|	$\underline{0.2375±0.0277}$|	**0.2667±0.0235**|

---

> ### Author Response · Authors · 2024-12-01
>
> For Opportunity, we only test it with seed=2023 lr=1e-3 and $\alpha$=0.2 **since the dataset is too large, it need very long training time**, but it also shows competitive results. we will continue to working on it if we have time.
> ## Acuuracy
> |Dataset| Target| VREx| GroupDRO| ANDMask| FreTS| PatchTST| GILE| AdaRNN| Diversify| FEDNet|
> |---|---|---|---|---|---|---|---|---|---|---|
> |OPP|0|0.8144|	0.8158|	0.8050|	0.8091|	0.7442|	$\underline{0.8444}$|	0.8158|	0.8044|	**0.8464**|
> ||1|0.7887|	0.7912|	0.7887|	0.8070|	0.7439|	0.7933|	0.7903|	$\underline{0.7937}$|	**0.8014**|
> ||2| 0.7425|	0.7449|	0.7373|	0.7627|	0.7167|	**0.7949**|	0.7553|	0.7319|	$\underline{0.7891}$|
> ||3| 0.7687|	0.7804	|0.7810	|0.7577	|0.7624	|$\underline{0.8037}$|	0.7915|	0.7690|	**0.8191**|
> ## F1-score
> |Dataset| Target| VREx| GroupDRO| ANDMask| FreTS| PatchTST| GILE| AdaRNN| Diversify| FEDNet|
> |---|---|---|---|---|---|---|---|---|---|---|
> |OPP|0|0.514|	0.5246|	0.4585|	0.5477|	0.299|	**0.6256**	|0.4406	|0.5085	|$\underline{0.6121}$|
> ||1|0.3855	|0.3957|	0.3626|	**0.5005**	|0.3064	|0.4375	|0.3266	|0.422	|$\underline{0.4540}$|
> ||2| 0.3257|	0.3001|	0.3021	|0.4131|	0.2601	|0.4406	|0.3074	|0.287|	**0.4430**|
> ||3|0.4062|	0.4125|	0.4015	|0.4723	|0.2608	|$\underline{0.4812}$	|0.4567	|0.3793	|**0.4968**|
> ## Precision
> |Dataset| Target| VREx| GroupDRO| ANDMask| FreTS| PatchTST| GILE| AdaRNN| Diversify| FEDNet|
> |---|---|---|---|---|---|---|---|---|---|---|
> |OPP|0|0.6197|	0.6177	|0.5238|	0.5946|	0.3834	|**0.6928**|	0.6267|	0.5728	|$\underline{0.6824}$|
> ||1|0.5095	|0.5157|	0.5064|	**0.6649**|	0.3998	|0.6096	|0.5575|	0.533|	$\underline{0.6517}$|
> ||2|0.3490|	0.3575|	0.3823|	0.5136	|0.335|	**0.5638**|	0.428|	0.3572|	$\underline{0.5579}$|
> ||3|0.4564|	0.4732|	0.4584	|0.5292	|0.4168	|$\underline{0.5457}$	|0.3567|	0.4186|	**0.5691**|
> ## Recall
> |Dataset| Target| VREx| GroupDRO| ANDMask| FreTS| PatchTST| GILE| AdaRNN| Diversify| FEDNet|
> |---|---|---|---|---|---|---|---|---|---|---|
> |OPP|0|0.4775|	0.4781	|0.4365	|0.5283	|0.2703|	**0.5908**	|0.3812	|0.4797	|$\underline{0.5730}$|
> ||1|0.3475	|0.3728	|0.3489|	**0.4199**|	0.2721|	0.3965	|0.2876|	0.3811	|$\underline{0.4024}$|
> ||2|0.346	|0.3047|	0.3104	|0.3901	|0.2352|	$\underline{0.4056}$	|0.2931|	0.3155|	**0.4222**|
> ||3|0.4108	|0.4145|	0.4308|	0.4736|	0.2194|	**0.4898**|	0.3697|	0.3906|	$\underline{0.4868}$|

---

> ### Author Response · Authors · 2024-12-01
> **Thank you for the review! Have we clearly addressed the concerns?**
>
> If the theory and experiment have addressed your concern, Hopefully, it will give you new confidence in our work and reconsider the contribution of our paper.

---

> > ### Comment · Reviewer_Fq1k · 2024-12-02
> >
> > Thanks for your clarifications.
> >
> > I found the proposed method is very similar to [1, 2] and it seems like just changing a downstream task for existing approaches thus the novelty is very limited.
> >
> > Also, I'm still confused about the mathematical formulation and suspect its correctness, considering the other reviews' suggestions, I made my final decision.
> >
> > [1] FEDformer: Frequency Enhanced Decomposed Transformer for Long-term Series Forecasting
> > [2] TFDNet: Time-Frequency Enhanced Decomposed Network for Long-term Time Series Forecasting

---

> > > ### Author Response · Authors · 2024-12-03
> > >
> > > The formulations we used are all related to the basic domain generalization[1][2] and the time series distribution in frequency is also followed by related works[3].
> > >
> > > [1] J. Wang et al., "Generalizing to Unseen Domains: A Survey on Domain Generalization," in IEEE Transactions on Knowledge and Data Engineering, vol. 35, no. 8, pp. 8052-8072, 1 Aug. 2023, doi: 10.1109/TKDE.2022.3178128.
> > >
> > > [2] Germain P, Habrard A, Laviolette F, et al. A new PAC-Bayesian perspective on domain adaptation[C]//International conference on machine learning. PMLR, 2016: 859-868.
> > >
> > > [3] He H, Queen O, Koker T, et al. Domain adaptation for time series under feature and label shifts[C]//International Conference on Machine Learning. ICML 2023.

---

> ### Author Response · Authors · 2024-12-03
>
> Thank you for your response. **I would like to clarify that our method is completely different from FEDformer and TFDNet** and we also mentioned it in our related work. Here we will show you the details of the difference from **motivation, method, code implementation**.
>
> ## Motivation
> 1.**FEDformer**: tried to **use FFT/Wavelet in accelerating multi-head attention** to modeling temporal dependence, it shows that we use a small set of frequency components that could **low-rank approximate KQV attention matrix**.
>
> 2.**TFDNet**: **didn't use FFT for decomposition, it used moving average pooling to decompose the feature in seasons and trends.** After decomposing the two features, they use STFT (an FFT method with multi-scale sliding windows), which will contain both time and frequency information spitting the features into 2d feature maps so that it could model seasonal features and trend features more carefully and locally.
>
> 3.**FEDNet**: our method **directly uses FFT for precomputing and getting top-k high amplitude frequency component positions in advance across different domains for decomposition**, it is ensured with Wold’s theorem and our work verified the robustness and stability in domain generalization.
>
> ## Method
> 1.**FEDformer** uses **the continuing first 32/64 position** i.e. [0,1,2..until..32...64] of frequency components (they are all low ranges and didn't consider the amplitude of components, the difference shows in **appendix Figure 12(a)(b)**) or **random index frequency component** for approximate KQV self-attention in every attention layer, its complexity will be with FFT $O(N \times d \log d)$, where $N$ is the number of attention layers and $d$ is the dimension after getting low-rank composition.
>
> 2.**TFDNet** uses moving average for decomposition and the STFT they used in FFN encoders  its complexity will be $O(N \times M \times d \log d)$, where $N$ is the number of FFN layers and $d$ is the dimension, $M$ is the number of sliding windows.
>
> 3.**FEDNet** take the **high average amplitude** into consideration and **get the top-k energy components as stable frequency components across the domains** and we **only calculate it once** before we feed them into the VAE or Patch encoders. Our complexity is completely $O(d \log d)$, $d$ is origin time series length $L$. We also use wavelets for decomposition to check the importance of the orthogonality frequency decomposition function, we test both orthogonal (db2,  coif1) and non-orthogonal (rbio1.3) function in **C.5 INTRINSIC RELATIONSHIP STUDY OF ORTHOGONAL FREQUENCY DECOMPOSITION**, whose goal is different from FEDformer and FEDformer only tests the Legendre orthogonal function.
>
> ## Code implementations
> 1. **FEDformer**
> ```
> def get_frequency_modes(seq_len, modes=64, mode_select_method='random'):
>     modes = min(modes, seq_len//2)
>     if mode_select_method == 'random':
>         index = list(range(0, seq_len // 2))
>         np.random.shuffle(index)
>         index = index[:modes]
>     else:
>         index = list(range(0, modes))
>     index.sort()
>     return index
>
> class FourierBlock(nn.Module):
>     def __init__(self, in_channels, out_channels, seq_len, modes=0, mode_select_method='random'):
>         super(FourierBlock, self).__init__()
>         self.index = get_frequency_modes(seq_len, modes=modes, mode_select_method=mode_select_method)
>         self.scale = (1 / (in_channels * out_channels))
>         self.weights1 = nn.Parameter(
>             self.scale * torch.rand(8, in_channels // 8, out_channels // 8, len(self.index), dtype=torch.cfloat))
>     def compl_mul1d(self, input, weights):
>         return torch.einsum("bhi,hio->bho", input, weights)
>
>     def forward(self, q, k, v, mask):
>         B, L, H, E = q.shape
>         x = q.permute(0, 2, 3, 1)
>         x_ft = torch.fft.rfft(x, dim=-1)
>         out_ft = torch.zeros(B, H, E, L // 2 + 1, device=x.device, dtype=torch.cfloat)
>         for wi, i in enumerate(self.index):
>             out_ft[:, :, :, wi] = self.compl_mul1d(x_ft[:, :, :, i], self.weights1[:, :, :, wi])
>         x = torch.fft.irfft(out_ft, n=x.size(-1))
>         return (x, None)
> ```
>
> 2. **TFDNet**
> ```
> class moving_avg(nn.Module):
>     def __init__(self, kernel_size, stride):
>         super(moving_avg, self).__init__()
>         self.kernel_size = kernel_size
>         self.avg = nn.AvgPool1d(kernel_size = kernel_size, stride = stride, padding = 0)
>
>     def forward(self, x):
>         front = x[:, 0:1, :].repeat(1, (self.kernel_size - 1) // 2, 1)
>         end = x[:, -1:, :].repeat(1, (self.kernel_size - 1) // 2, 1)
>         x = torch.cat([front, x, end], dim = 1)
>         x = self.avg(x.permute(0, 2, 1))
>         x = x.permute(0, 2, 1)
>         return x
>
>
> class series_decomp(nn.Module):
>     def __init__(self, kernel_size):
>         super(series_decomp, self).__init__()
>         self.moving_avg = moving_avg(kernel_size, stride = 1)
>
>     def forward(self, x):
>         moving_mean = self.moving_avg(x)
>         res = x - moving_mean
>         return res, moving_mean
> ```

---

> ### Author Response · Authors · 2024-12-03
>
> 3. **FEDNet**
> ```python
> class Exp_Classification(Exp_Basic):
>     def __init__(self, args):
>         super(Exp_Classification, self).__init__(args)
>
>     def _get_mask_spectrum(self, freq_type):
>         """
>         get shared frequency spectrums
>         """
>         train_data, train_loader = self._get_data(flag='TRAIN')
>         amps = 0.0
>         for data in train_loader:
>             lookback_window = data[0]
>             B, L, C = lookback_window.shape
>             # mask
>             # lookback_window = self.precise_random_mask_batch(lookback_window, self.args.mask_rate)
>             frequency_feature = None
>             if freq_type == "fft":
>                 frequency_feature = torch.fft.rfft(lookback_window, dim=1)
>             elif freq_type in ['db2', 'sym2', 'coif1', 'bior1.3', 'rbio1.3']:
>                 wavelet = pywt.Wavelet(freq_type)
>                 # print("ortho=", wavelet.orthogonal)
>                 lookback_window = lookback_window.permute(0,2,1)
>                 device = lookback_window.device
>                 X = lookback_window.numpy()
>                 cA, cD = pywt.dwt(X, wavelet)
>                 frequency_feature = np.concatenate((cA, cD), axis=2).transpose((0,2,1)) # B D C
>                 frequency_feature = torch.from_numpy(frequency_feature).to(device)
>
>             assert frequency_feature != None
>             amps += abs(frequency_feature).mean(dim=0).mean(dim=1)
>         mask_spectrum = amps.topk(int(amps.shape[0]*self.args.alpha)).indices
>         print("mask_spectrum:", mask_spectrum)
>         return mask_spectrum # as the spectrums of time-invariant component
> ```
>
> ```python
> class FrequencyFilter(nn.Module):
>     """
>     Fourier Filter: to time-variant and time-invariant term
>     """
>     def __init__(self, mask_spectrum, freq_type):
>         super(FrequencyFilter, self).__init__()
>         self.mask_spectrum = mask_spectrum
>         self.freq_type = freq_type
>
>     def forward(self, x):
>         B,L,C = x.shape
>         if self.freq_type == "fft":
>             xf = torch.fft.rfft(x, dim=1)
>             mask = torch.ones_like(xf)
>             mask[:, self.mask_spectrum, :] = 0
>             x_sto = torch.fft.irfft(xf*mask, dim=1)
>         elif self.freq_type in ['db2', 'sym2', 'coif1', 'bior1.3', 'rbio1.3']:
>             wavelet = pywt.Wavelet(self.freq_type)
>             device = x.device
>             data = x.permute(0,2,1).cpu().numpy()
>             cA, cD = pywt.dwt(data, wavelet)
>             Dims = cA.shape[-1]
>             frequency_feature = np.concatenate((cA, cD), axis=2)
>             mask = np.ones_like(frequency_feature)
>             mask[:, :, self.mask_spectrum] = 0
>             frequency_feature = frequency_feature * mask
>             cA_sto, cD_sto = frequency_feature[:, :, :Dims], frequency_feature[:, :, Dims:]
>             x_sto = pywt.idwt(cA_sto, cD_sto, wavelet)
>             x_sto = torch.from_numpy(x_sto).to(device).permute(0, 2, 1)
>         x_det = x - x_sto
>         return x_sto, x_det
> ```

---

### Official Review · Reviewer_YpR1 · 2024-10-28

**Soundness:** 3
**Presentation:** 2
**Contribution:** 2
**Rating:** 5
**Confidence:** 4

**Summary:**

This paper presents a significant advancement in handling OOD time-series classification by providing a unified framework that considers both types of distribution shirts, domain and temporal.

The frequency-based approach appears to be particularly effective in capturing invariant features that generalize well to OOD scenarios.

The main novelty of FEDNet is to use frequency information to guide the decomposition of time-series data. By separating deterministic and stochastic components, it can be better identify and leverage invariant features that remain consistent across dirrerent domains.

**Strengths:**

- The FEDNet is a novel approach to address challenges in out-of-distribution time-series classification using frequency domain information for feature decomposition. Unlike existing methods that primarily focus on the time domain, FEDNet uses both frequency and time domains, guided by theoretical insights from Wold’s Theorem.
- This paper is supported by empirical results through experiments.
- Theoretical insights with time-deterministic and time-stochastic components.

**Weaknesses:**

There are a few unresolved clarifications in the question below.

**Questions:**

- The challenge presented in Figure 1 is unclear. It would be helpful to explicitly indicate the specific problem being targeted, such as domain shift or temporal distribution shift, and the key solution proposed. Additionally, clarifying why the frequency domain is used and why the decomposition into deterministic and stochastic components is necessary would improve understanding.
- It is unclear why deterministic features undergo the top-k process and why stochastic features undergo both the top-k and masked frequency processes. A more detailed explanation or a visual diagram illustrating these processes would be beneficial.
- Please provide a clearer rationale for dividing the stochastic and deterministic features in the frequency domain. A brief theoretical justification about this necessity or empirical evidence linking frequency domain characteristics to the properties of time series data would strengthen the argument.
- In the performance comparison section, it is noted that FEDNet is not the best across all datasets. Could you provide an analysis of the conditions or characteristics of datasets where FEDNet does not outperform other methods? Discussing potential limitations or specific scenarios where other methods are superior would provide valuable insights.

---

> ### Author Response · Authors · 2024-11-24
>
> **challenge in time series ood(W1)**: We apologize for the confusion in understanding the article due to the lack of clarity in our picture, and we have revised the figure and the definition of temporal distribution shift in the article. Essentially, the temporal distribution shift depicted in our Figure 1 refers to the fact that the temporal distribution will change due to temporal non-stationarity at different time periods within the domain, leading to a shift in the marginal probabilities, and thus a shift in the joint probability distribution.
>
> **In this case, our artificial division of the domain is not ideal, and we need to solve the problem of how to achieve out-of-distribution generalization under the condition that the data within the source domain does not completely follow its domain distribution.**
>
> **the role of frequency filter(W2,W4)**:
> 1. Thank you for your interest in the principles behind our frequency domain decomposition mechanism! **Specifically, the advantage od frequency orthogonality is the key reason why we can solve the timing OOD problem**. Unlike the time-dependence of different time points in the time domain, the frequency domain view is orthogonal between different frequency components, which allows the effects of these two offsets to be independent of each other, which inspires us to find stable frequency components that are less affected by the offset of the temporal distribution, and thus obtain pure domain-invariant features. At the same time, we retain the random component to prevent information loss as an auxiliary classification feature to increase the generalization space.
>
> 2. We derive the relationship between domain generalization theory and frequency domain components from **Proposition 5.2 (Frequency Perspective Risk Bound on Unseen Time series Domain)**. The proof of the relationship between the domain generalization theory and the frequency domain components is given in the appendix of the paper. And we verify that using $x_{det}$ features instead of the original features further enhances the effectiveness of the  IRM method. For more detailed descriptions, see **Reviewer dy8u Frequency Composition(W4,W5,W6,W11)** and **Reviewer b2Zv motivation(W1)**
>
>
> **frequency filter process(W3)**: We are very sorry for misleading you about the model figure due to our inappropriate expression, here we have modified the formulae and pictures in the text to be consistent with our actual code implementation version, the decomposition process will only be performed once fft to extract the frequencies of topk, and then set the amplitude of these frequencies to 0 (i.e., mask operation), and then we get $x_{sto}$ through the ifft, and then we can use $x - x_{sto}$ to get $x_{det}$. Then we can compute $x_{det}$ using $x - x_{sto}$, the original version is to show you that the process is equivalent.
>
> **limitations (W5)**:
> We analyzed some of the flawed situations of our method.
> 1. We visualize the spectrogram of FEDNet before and after masking, as in Figure 12 (c)(d) in the paper.It can be seen that, through masking, the diversity between domains is more pronounced, while many redundant frequencies are eliminated.But when there are some outliers in the frequencies, this definitely affects the training process of the model on domain invariant features.
>
> 2. We analyzed the OOD generalization for the classification of EEG signals with ultra-long sequences (L=3000), and found that the spectrograms produced by ultra-long sequences are more prone to unknown frequency distributions, as shown in Figure 8, and when we use domain 2 as the target domain, our model is unable to detect such unique frequency variations from the training domain, and it is possible that these frequencies also belong to invariant features, this example shows us the drawbacks that our method may face, that is, if we completely rely on the training domain to extract the invariant features learned from stable frequencies is not sufficient there may be bias, especially in the case of very long sequences will be lost a lot of frequency signals, which is why we need to retain and design the stochastic encoder module.
> 3. We add an interesting experiment of missing value with time series, which suggests a valuable insight for our approach. That is when we extract insufficient domain invariant features, it is crucial to supplement more time-stochastic information and improve the generalisation space. Meanwhile, this also suggests more interesting directions and improvement questions for the future, on the one hand, how to deal with biased or insufficient domain information, and on the other hand, how to do Time Series Imputation with OOD.
> For more analysis, see the **Reviewer b2Zv time-stochastic study (W4)** and see the **C.3. TIME-STOCHASTIC FEATURE STUDY** in the appendix.

---

> ### Author Response · Authors · 2024-11-26
>
> By the way, we have added a lot of theoretical and experimental content in this new version, including the role of the two components, theoretical generalization upper bounds and frequency domain isolation levels, etc. I hope these interesting experimental phenomena have given you more interest, and luckily, we still have a week of discussion and wishing you a Merry Christmas in advance!

---

> ### Comment · Reviewer_YpR1 · 2024-11-26
>
> Thank you for your response. I have reviewed the revised paper, and a new concern has arisen. From additional experiments, it appears that the proposed solution does not have SOTA performance across all datasets. This raises doubts about whether the proposed solution is indispensable from an experimental perspective.
> I will maintain my score.

---

> ### Author Response · Authors · 2024-11-26
>
> Thank you very much for your timely reply, we understand your need for experimental performance, as the training of multiple domains is relatively time consuming, the newly added dataset is relatively large, in order to supplement the new experiments and theories we have not specifically adjusted the various parameters of the model, our next step is to fine tune the model's hidden_size, alpha hyperparameters, num_layers and so on.
>
> **We promise to continue to increase our experimental efforts to address your concern, since we still have one week. We will try our best to improve the performance to address your concern, as soon as we have new experimental results we will give you immediate feedback here**
>
> **Thanks again for your advice to improve our work.**

---

### Official Review · Reviewer_8bmq · 2024-11-04

**Soundness:** 3
**Presentation:** 3
**Contribution:** 3
**Rating:** 6
**Confidence:** 4

**Summary:**

This manuscript presents a new approach to improving domain generalization in deep neural networks. The authors introduce FEDNet, a model designed to enhance frequency-based generalization capabilities, focusing on its application to time series classification. The paper thoroughly evaluates the performance of FEDNet across multiple synthetic and real datasets, including Opportunity, HHAR, UCIHAR, and UniMiB-SHAR, comparing it against baseline methods like VREx, GroupDRO, ANDMask, and others of different flavors.

**Strengths:**

S1. Overall, this paper is well-written and easy to follow.

S2. The design intuition of this paper, i.e., isolating the sift-relevant components to the time-deterministic block only, makes sense and demonstrates its advantage in empirical evaluations.

**Weaknesses:**

W1. The connection between the two major components, i.e., (1) the frequency-domain decomposition and (2) the sift-relevant deep architecture, is not well-justified in the paper. If my understanding in S2 (which I just realized after reading the whole methods section) in the design target of the authors, I would suggest the authors to revise the paper and clearly motivate this.

W2. There are some minor presentation ambiguity and revision suggestions:

W2-1. The equation in line 152 seems a bit weird to me, P(x_i, y_i) and P(x_j, y_j) are the probabilities of two individual examples, while P^Dk(x, y) is the probability distribution of the entire subset D_k. These two cannot be directly compared.

W2-2. It would be very useful if the authors can briefly summarize the impact of Proposition 4.1 in the beginning of Section 5.4. This should include what are introduced (the per-domain constraint) in this paper and how Equation 15 is realized in FEDNet.

**Questions:**

Q1. Despite I agree that the domain/distribution/etc sift information should be more reflected in the time-deterministic decomposed part of the time series, will these sifts also influence the time-stochastic part of the series?

---

> ### Author Response · Authors · 2024-11-24
>
> **writing suggestions(W1, W2)**: Thank you very much for recognizing our work! Your understanding is right to our idea, we have revised the introduction of frequency domain decomposition, the explanation of propotion. We modifiedthe definition of temporal distribution shift to be more clear. Essentially, the root of this shift is the non-stationarity of the series, and we explicitly gave the concept of periods, and we considered that data belonging to the same period inside the domain will follow the same data distribution, while different periods will be different from each other lead to changes in the joint probability distribution due to changes on their marginal probability distribution. We also add the theory related to domain generalization in the appendix **A.4 THE RISK FOR DOMAIN GENERALIZATION OF TIME SERIES FROM FREQUENCY VIEW**, as well as more experimental insights of $x_{det}$ and $x_{sto}$ components, we hope you are interested in!
>
> **shift on time-stochastic(Q1)**: Generally, the time-stochastic part is indeed affected by the domain shift. With the orthogonality of frequencies, both shifts affect each frequency component independently, but our more important goal is still to purify the invariant features that are not affected by time-shift, and in our experiments, we can also see that the random component module plays more of an auxiliary role and prevents the loss of information.
>
> Anyway, there are some works [1] also proved that the generalization space of domain-invariant features can be increased to some extent by using domain-specific features that are beneficial for classification. Meanwhile, for a more detailed study of the role of time-stochastic component, you could see the **TIME-STOCHASTIC FEATURE STUDY in the appendix** of the paper or **the answer to Reviewer b2Zv time-stochastic study (W4)**.
>
> [1] X. Yu, H. -H. Tseng, S. Yoo, H. Ling and Y. Lin, INSURE: An Information Theory iNspired diSentanglement and pURification modEl for Domain Generalization, in IEEE Transactions on Image Processing.

---

> > ### Comment · Reviewer_8bmq · 2024-11-26
> >
> > Thanks for your rebuttal. I will maintain my score.

---

> > > ### Author Response · Authors · 2024-11-26
> > > **Thank you for the review! Have we clearly addressed the concerns?**
> > >
> > > Thank you very much for your timely reply! we have analysed more interesting conclusions and phenomena both from theory and experiment in this new version. Since we still have one week of discussion, if you still have any questions or suggestions about our work, don't hesitate to tell us!
> > >
> > > We will continue to modify and add experiments, thank you again for your contribution, and Merry Christmas in advance!

---

### Official Review · Reviewer_b2Zv · 2024-11-04

**Soundness:** 2
**Presentation:** 2
**Contribution:** 2
**Rating:** 5
**Confidence:** 3

**Summary:**

The paper proposes a novel method from the frequency perspective, FEDNet. The paper identifies the limitations of the IRM-based approach for domain generalization in time series classification tasks. To address this, the proposed FEDNet method utilizes frequency domain information to decompose time series into stable features and applies filtering and denoising techniques from a frequency domain perspective. This approach mitigates the distribution shift in sequence data to some degree. The experiments demonstrate that FEDNet achieves state-of-the-art performance in OOD time series classification tasks.

**Strengths:**

* The paper examines the issues associated with traditional IRM algorithms for time series data and explores out-of-distribution (OOD) time series classification from a frequency standpoint.

* The paper decomposes time series into two components: time-deterministic and time-stochastic, distinguishing the temporal distribution based on frequency information.

* The experiments show that the proposed frequency information is effective and the frequency component exhibits stability to temporal distribution shift.

**Weaknesses:**

* The key motivation of the paper seems weak because temporal distribution shift and frequency distribution shift actually consider the same data distribution shift from different perspectives. The benefits of approaching the distribution shift from the frequency view are unclear. In addition, the paper argues that the temporal distribution shift is influenced by the window division. However, to the reviewer, increasing the window size might help mitigate this shift, as it could provide a more stable representation of the data over time.

* The paper provides two schemes for constraint loss, but no experiments are designed to compare them. The proposed method also involves multiple different loss functions, it is not clear how to balance them in the model optimization.

* The baseline method ERM algorithm actually demonstrates a very competitive performance, which further weakens the motivation of the paper. The adopted datasets in the experiments lack diversity since they only include the human activity datasets.

* In the time-stochastic block, a two-layer transformer encoder is used with a patch length of 16 and a hidden size of 512. However, as shown in Figure 4, such a complex module only brings limited performance gains, which questions the necessity of such complexity. Besides, the symbols are not clearly defined, such as $x_p$ in Eq. 11 and there are also labeling errors in Table 4.

**Questions:**

See the weaknesses.

---

> ### Author Response · Authors · 2024-11-24
>
> **motivation(W1)**: We are very sorry that our description of the temporal distribution shift in figure 1 may have caused you a misunderstanding. I'd like to first clarify that **Temporal distribution shift is essentially a property of the non-stationary time series over time, and is not caused by a sliding window.** More specifically, the temporal distribution shift is a change in the marginal probabilities over time due to the non-stationarity of the time series, and we have revised the description of temporal distribution shift in Figure 1 and the definition of temporal distribution shift in the paper. We have modified the description of temporal distribution shift in figure 1 and reworked the definition of temporal distribution shift in the paper.
>
> In order to give you a better understanding of the background and motivation of our paper, I'll answer your questions in turn from the multiple perspectives you mentioned.
>
> 1. **The role of sliding windows in mainstream time series classification tasks.**
> A sliding window is also necessary for temporal classification tasks. **In fact the original samples contain many invalid time segments[1], these segments can also affect feature learning and distribution if they are not eliminated**, and **the samples are not always keep the same class states at different times[2]**, you can see this in detail in the dataset's introduction and in the code of the open-source dataset we reproduced. Increasing the length will not alleviate this problem, but make it more internally complex and the time complexity of inference will increase because it contains more noise. It is common to separate the data into different time segments after cleaning them using a suitable length, which does not eliminate temporal non-stationarity, but it is a technique similar to data augmentation and makes it easier to capture localised patterns, e.g. some segments features of the same period may follow the same distribution.
>
> 2. **The benefit for modeling distribution shift in frequency.**
>     - **Frequency orthogonality to eliminate time dependence.** The most fundamental reason for using the frequency domain is that changes in the distribution of a time series in the time domain are difficult to model with higher-order moment estimation, and the dependence between times is not conducive to our analysing the sources of distributional shifts, whereas all frequency components in the frequency domain space satisfy orthogonality and do not interfere with each other. Therefore we can assume that both distributions affect each frequency independently, and we can derive expressions for the frequency domain marginal distributions as shown in **Corollary 1 (Frequency Marginal Probability Distribution of Time Series Data)** in the appendix of our paper. **This facilitates us to analyse the relationship between different distributional shifts and individual frequencies.** We verified the relationship between the distribution shift and the frequency domain components, and found that we can use the orthogonality of the frequency domain to separate the two components and weaken the effect of the temporal distribution shift to purify and obtain more robust domain invariant features.
>     - **Reducing redundant frequency components is beneficial for lowering the maximum distance between domains, thus lowering the generalisation upper bound.** We derive **Proposition 5.2 (Frequency Perspective Risk Bound on Unseen Time series Domain)** by combining the theory of PAC-Bayesian domain adaptation[1] with the form of the marginal probability distribution in the time series frequency domain.And we have theoretically and experimentally verified the optimal frequency $\alpha$ ratio ranges 0.2-0.4, the details refer to **Reviewer dy8u empirical domain divergence study with $\alpha$ isolated level(W10)**.

---

> ### Author Response · Authors · 2024-11-24
>
> 3. **Considering temporal distribution shift is gainful for Long time series.**
> To clear up your confusion, we specifically supplemented a case study with a long EEG sequence dataset(L=3000) to show you that very long length sequences considering this offset will work better than traditional domain generalization methods. Due to the very long training time of this dataset, here we have only run through the results for EEG domain 2 for test.
>
> | Method   | Accuracy | F1-score | Precision | Recall |
> |----------|----------|----------|-----------|--------|
> | VREx     | 68.58    | 56.95    | 57.44     | 59.02  |
> | GroupDRO | 69.40    | 55.83    | 58.31     | 56.53  |
> | ANDMask  | 69.44    | 57.47    |  59.40    |  57.97 |
> | FEDNet   | **72.90**| **62.16**| **61.74**|**63.01**|
>
> 4. **Our fundamental motivation and question.**
> **Although it may appear that we are addressing issues specific to time series, our goal is to argue a more fundamental problem: in real-world scenarios, the domains we artificially divide are not perfect, and the data within a domain does not strictly follow an ideal domain distribution**. The key question is how to achieve out-of-distribution generalization under such conditions. One approach, similar to Adarnn[4] and Diversify[5], is to use ensemble weighting or adversarial training to redefine pseudo domain division for better feature learning of time segments. However, this method essentially equates to data augmentation, and the multi-stage training significantly increases the model's training time complexity in our experiments. **On the contrast, our motivation follows a completely different idea: how to leverage existing data itself priors to purify and extract ideal domain-invarint features when facing temporal shift int the domains since the periods could be infinity in realistic, limiting the maximum number of pseudo-domains is not always desirable.**
>
> [1] Reiss A, Stricker D. Introducing a new benchmarked dataset for activity monitoring[C]//2012 16th international symposium on wearable computers. IEEE, 2012.
>
> [2] Supratak A, Dong H, Wu C, et al. DeepSleepNet: A model for automatic sleep stage scoring based on raw single-channel EEG[J]. IEEE transactions on neural systems and rehabilitation engineering, 2017.
>
> [3] Germain P, Habrard A, Laviolette F, et al. A new PAC-Bayesian perspective on domain adaptation. ICML 2015.
>
> [4] Du Y, Wang J, Feng W, et al. Adarnn: Adaptive learning and forecasting of time series[C]//Proceedings of the 30th ACM international conference on information & knowledge management. 2021: 402-411.
>
> [5] Lu W, Wang J, Sun X, et al. Out-of-distribution representation learning for time series classification[J]. ICLR,2023.

---

> ### Author Response · Authors · 2024-11-24
>
> **diverse dataset(W3)**
> 1. The ERM method also performs well, on the one hand, because the signal variations in the human activity dataset are easy to capture, on the other hand, the goodness of the generalization space usually depends on the gap between the training domains and the test domains, if a space consisting of multiple training domains is diverse enough, then the direct gaps between it and the test set are relatively small, at this time it is normal to get a good prediction result with ERM, as we The UCIHAR used is less heterogeneous, so it works well on multiple domains, as mentioned in the diversify[1] paper. In addition, we can also focus on the fact that the ERM method generally has the largest fluctuation of results under different domain tests, which is formally the problem we want to solve in OOD.
>
> 2.To address your concern, we added more diverse datasets for the experiments, we continue to reproduce the **DSADS, PAMAP two datasets with more heterogeneity used by diversify**, as well as **the EMG (electromyogram) signal data**, and **the EEG(Electroencephalogram) signal dataset with the ultra-long sequence of L=3000** as the baseline, and supplemented all the experimental results. It can be seen that on these datasets the ERM method performs overall worse than many ood methods, especially on the EMG and DSADS datasets.
>
> |Dataset| Shape| Classes| Domains|
> |---|---|---|---|
> |DSADS| (125, 45)| 19| 4|
> |PAMAP| (200, 27)| 18| 4|
> |EMG| (200, 8)| 6| 4|
> |EEG| (3000, 1)| 5| 4|
>
> |Dataset| Target| VREx| GroupDRO| ANDMask| FreTS| PatchTST| GILE| AdaRNN| Diversify| FEDNet-f| FEDNet-w|
> |---|---|---|---|---|---|---|---|---|---|-----|---|
> |EMG|0|70.25|70.66|71.60|$\underline{71.77}$|33.92|62.96|54.11|67.55|**73.00**|64.55|
> ||1|$\underline{85.50}$|83.08|82.52|80.15|36.82|68.02|57.44|81.09|**87.10**|59.59|
> ||2|73.62|77.03|76.91|74.88|22.66|66.02|57.83|74.64|**79.66**|$\underline{77.51}$|
> ||3|77.14|$\underline{78.62}$|77.50|77.96|36.62|69.99|53.87|77.32|77.43|**79.85**|
> |DSADS|0|80.26|84.69|82.50|80.26|82.24|89.64|83.11|77.19|**92.80**|$\underline{92.41}$|
> ||1|76.54|78.03|73.42|70.13|74.07|78.20|79.78|77.28|**84.86**|$\underline{83.64}$|
> ||2|86.40|85.96|83.03|84.29|82.67|86.75|83.46|85.22|**93.24**|$\underline{90.65}$|
> ||3|74.61|74.39|78.46|73.46|78.85|79.56|70.35|71.80|**87.71**|$\underline{80.52}$|
> |PAMAP|0|62.88|61.75|61.84|55.22|60.40|$\underline{65.01}$|63.30|61.98|64.94|**67.48**|
> ||1|54.88|52.00| 53.04 |60.41 |$\underline{66.36}$| 51.46| 54.24| 54.38| 61.67| **67.08**|
> ||2| 22.68| 25.69| 28.02| 34.98| **50.06**| 25.23| 23.35| 24.32| 34.39|$\underline{35.27}$|
> ||3| 62.10| 65.22| 67.86| **68.69**| 63.39| 68.06| 61.04| 57.79| 67.13|$\underline{68.55}$|
>
> 3. **By the way, empirically we also agree with this point you mentioned, as you can see that many OOD related works[2] have not been compared with strong baseline models, and it would be valuable for the development of the whole field for us to make this point in our paper.**
>
> Regarding some issues with the performance of the added experiments, the training is more time-consuming due to the somewhat large size of these multi-domain datasets. We are doing our best to solve it, because the previous 2 weeks added more content and did not deliberately adjust the relevant parameters, fortunately, we currently have a week, we are continuing to add.
>
> [1] Lu W, Wang J, Sun X, et al. Out-of-distribution representation learning for time series classification[J]. ICLR,2023.
>
> [2] Gagnon-Audet J C, Ahuja K, Darvishi-Bayazi M J, et al. WOODS: Benchmarks for out-of-distribution generalization in time series[J]. arXiv preprint arXiv:2203.09978, 2022.

---

> ### Author Response · Authors · 2024-11-24
>
> **loss study(W2)**
> 1. **cross vs contrastive loss**. Thank you for pointing out the imperfections of the experiments, we added **performance ablation experiments** and **domain invariant feature $\mathcal{A}$-distance evaluation** experiments for all datasets for both constrained losses.
>
> |Dataset|Target|FEDNet-cross|FEDNet-contrast|
> |---|---|---|---|
> |Spurious Fourier| 0| 74.22| **75.31**|
> |HHAR| 0| 99.18 |**99.48**|
> ||1|98.98|**98.98**|
> ||2|90.97|**91.59**|
> ||3|58.18 |**67.87**|
> ||4| 58.11| **60.81**|
> |UCIHAR| 0| **99.42** |95.67|
> ||1|**83.74**|80.13|
> ||2| 97.15|**98.53**|
> ||3| 93.21|**94.64**|
> ||4| 98.34|**99.02**|
> |UniMiB-SHAR|1|57.55|**58.85**|
> ||1| 63.63|**70.15**|
> ||2|70.06|**71.05**|
> ||5|**44.63**|44.29|
> |Opportunity|S1|84.80 |**85.02**|
> ||S2|79.75|**81.45**|
> ||S3| 77.89|**79.21**|
> ||S4| 81.36|**81.96**|
> |DSADS Cross Position|0|37.54|**40.54**|
> ||1|35.13|**36.05**|
> ||2|36.25|**36.25**|
> ||3|32.16|**33.93**|
> ||4|32.78|**33.04**|
> |EMG|0| 64.84 |**73.00**|
> ||1| 82.13|**87.10**|
> ||2|  71.53|**79.66**|
> ||3|  69.87|**79.85**|
> |DSADS|0|90.74|**92.80**|
> ||1|82.63|**84.86**|
> ||2|87.36|**93.24**|
> ||3|71.88|**87.71**|
> |PAMAP|0| 66.53| **67.48**|
> ||1|47.93|**67.08**|
> ||2| 28.32|**35.27**|
> ||3| 64.73|**69.80**|
>
> We added the results of cross label loss and contrastive loss for all the datasets used and updated the results of the ablation experiments, and we can see that the use of contrastive learning effectively improves the performance of the model.The UCIHAR, UniMiB-SHAR, and Opportunity datasets do not have a validation dataset set up in the implementation of the original paper. We have directly followed the original in order to maintain the accuracy of the reproduction, and there may be some strong overfitting problems. The other datasets all use a 20% validation set, and we can very clearly find that the performance is more robust using contrastive learning.
>
> **In addition, we found that using contrastive learning loss can further reduce the $\mathcal{A}$-distance. This suggests that, compared to standard cross-label loss, contrastive learning can produce more robust invariant features.**  This is likely because contrastive learning inherently possesses properties similar to DRO (distributionally robust optimization) [1]. A more detailed explanation can be found in the **Invirant Feature Study** in the paper and the responses in **Reviewer dy8u time-deterministic with variational inference(W8)**.
>
> |   Model  |  EMG domain-0 | EMG domain-1 |EMG domain-2| EMG domain-2|
> |  ----  | ----  |----  |----  |----  |
> | IRM  $\mathcal{A}$-distance| 0.8344  |0.7638 |0.8551  |0.8327|
> | FEDNet cross  $\mathcal{A}$-distance| 0.4664 | 0.4320  |0.4539 |0.4664|
> | FEDNet contrast $\mathcal{A}$-distance| **0.3570** | **0.4204**  |**0.4320** |**0.3482**|
>
> 2. **hyper-parameters $\lambda_{det}$ and $\lambda_{sto}$ study**. We didn't deliberately adjust these parameters, we used the default 1.0 for all datasets, to further address your concern, we tried $\lambda_{det} = [0.1, 1.0, 10.0]$  and $\lambda_{sto} = [0.1, 1.0, 10.0]$ for some easy adjustments. We performed performance prediction for 9 sets of hyperparameter combinations using UCIHAR's domain-0 as the target domain. **We found that the performance of our approach is robust and does not require very sensitive tuning of them.**
>
> |x_det\x_sto| 0.1 | 1.0 | 10.0|
> |---| ---|---|---|
> |0.1| 0.8841|0.8774|0.8774|
> |1.0| 0.8708|0.8543|0.8774|
> |10.0|0.8841|0.8774|0.8774|
>
> [1] Wu J, Chen J, Wu J, et al. Understanding contrastive learning via distributionally robust optimization. NIPS 2023.

---

> ### Author Response · Authors · 2024-11-24
>
> **time-stochastic study (W4)**
> Thank you for mentioning time-stochastic features! First I need to point out that we used d_model=512 and e_layers=2 in order to align with the default parameters of PatchTST's patch embedding for fair comparisons, and we didn't intentionally adjust them. What's more, we have analyzed the time-stochastic moudle at **three levels (feature, data, domain)** and found some interesting phenomena, **we designed this module most of all to avoid the loss of information, and secondly as an auxiliary feature to increase the generalization space of the domain invariant features, our here design idea is similar to the idea mentioned in Figure 1 of the INSURE**[1] .
>
> 1. **Patch condition number (feature level).** We analyzed the original patch matrix after dividing the
> Time-stochastic feature into patches.We used the condition number $\text{cond}(x_{patch}) =\sigma_{max} / {\sigma_{min}}$, where $\sigma_{max}$ and $\sigma_{min}$ represent the maximum and minimum singular values of $x_{patch}$ to evaluate this part of the features. **We find that using non-overlapping patch matrices effectively reduces the condition number of patch matrices, which makes it easier to train his input patch embedding linear layer stably and can effectively reduce the number of patches and accelerate training.** P donate the patch len in the table bellow:
>
> |Dataset| $\alpha$ = 0.2| | $\alpha$ = 0.4| | $\alpha$ = 0.6| |
> |---|---|---|---|---|---|---|
> ||stride = P/2| stride = P| stride = P/2| stride = P|stride = P/2| stride = P|
> |Spurious Fourier| 377.80 |$\textbf{14.57}\downarrow$ | 1211.58 |$\textbf{41.70}\downarrow$ | 200.43 | $\textbf{24.53}\downarrow$|
> |HHAR| 57.31 |93.70 | 6702.17 |$\textbf{5661.14}\downarrow$ | 4149784.84 | $\textbf{3688642.06}\downarrow$|
> |UCIHAR| 7600.79 |$\textbf{30.31}\downarrow$ | 159401 |$\textbf{37.88}\downarrow$ | 147293444 | $\textbf{161.05}\downarrow$|
> |SHAR| 47.13 |$\textbf{11.21}\downarrow$ | 9288.10|$\textbf{17.402}\downarrow$ | 830192 | $\textbf{38.3774}\downarrow$|
> |EMG| 27.26 |84.076 | 2065.86|$\textbf{152.440}\downarrow$ | 1561521 | $\textbf{12334.38}\downarrow$|
> |OPP| 151.68 | $\textbf{2.0032}\downarrow$ | 59.24|$\textbf{1.8594}\downarrow$ | 81.89 | $\textbf{1.78146}\downarrow$|
> |DSADS| 351.83 | $\textbf{26.7469}\downarrow$ | 2260.30|$\textbf{15.1602}\downarrow$ | 3226488.87 | $\textbf{37.6795}\downarrow$|
> |PAMAP| 49.98 | 71.068 | 4921.72|$\textbf{245.448}\downarrow$ | 2968754.15 | $\textbf{26859.82}\downarrow$|
>
> 2. **Time series missing value study (data level).** We tried an interesting experiment is to missing the original time series to different levels while keep $\alpha= 0.2/0.4$  unchanged, we want to destroy as many time points as possible, so that the invariant features and the main frequencies change drastically, thus making the invariant features problematic, which may be a more realistic kind of scenario, this also provides some valuable insight! **We find that the use of $x_{sto}$ brings more gain when we have more severe data missing, i.e., the worse the invariant features are, which illustrates the necessity of the time-stochastic moudle design, when we face insufficient invariant features, and we need to avoid the loss of information to enhance the generalization ability.**
>
> |Dataset| missing = 20%| | missing = 40%| |missing = 60%| |missing = 80%| |
> |------------|----------|-------|---|--------------|---|---|----------------|--------------|
> |  |w/o $L_{sto}$| FEDNet| w/o $L_{sto}$| FEDNet| w/o $L_{sto}$| FEDNet| w/o $L_{sto}$| FEDNet|
> |UCIHAR|94.52| **98.55**| 94.52| **97.41**| 64.55| **70.60**| 69.74| **74.06**|
> |UniMiB-SHAR| 52.60| **52.86**| 50.52| **51.56**| 38.54| **40.36**| 30.98| **38.80**|
> |EMG| 67.43| **67.49**| 38.62| **40.73**| 16.55| **27.88**| 16.55| **23.59**|
>
> [1] X. Yu, H. -H. Tseng, S. Yoo, H. Ling and Y. Lin, INSURE: An Information Theory iNspired diSentanglement and pURification modEl for Domain Generalization, in IEEE Transactions on Image Processing.

---

> > ### Author Response · Authors · 2024-11-24
> >
> > 3. **EEG Long signal case study (domain level).**
> > We analyzed the OOD (out-of-distribution) generalization performance of EEG signal classification for ultra-long sequences (L=3000). It was observed that spectrograms generated from ultra-long sequences are more likely to exhibit unknown frequency distributions. As an example shown in **Figure 8**, when using domain 2 as the target domain, our model fails to detect this unique frequency variation from the training domains. These frequencies could potentially belong to invariant features.
> >
> > This example highlights a limitation of our approach: solely relying on time-deterministic frequency features learned from the training domains may not be sufficient and could introduce biases, especially in the case of ultra-long sequences, where many frequency signals may be lost. This is why we need to retain and design the time-stochastic encoder module.
> >
> > | Method   | Accuracy | F1-score | Precision | Recall |
> > |----------|----------|----------|-----------|--------|
> > | VREx     | 68.58    | 56.95    | 57.44     | 59.02  |
> > | GroupDRO | 69.40    | 55.83    | 58.31     | 56.53  |
> > | ANDMask  | 69.44    | 57.47    |  59.40    |  57.97 |
> > | FEDNet w/o $L_{sto}$   | 69.99 |57.97| **63.26**| 59.31|
> > |FEDNet+ self-attention| **72.90**| 62.16| 61.74| 63.01|
> > |FEDNet+ DwConv| 70.21| 57.78| 56.33| 62.09|
> > |FEDNet+ MLP| 71.85| **62.94**| 61.80| **65.19**|
> >
> > time-stochastic parts exhibits 3% performance gap compared to FEDNet. Furthermore, we also explored simplifying the attention mechanism module by replacing it with MLPs and depthwise separable convolutions. This approach proved effective, performing significantly better than directly discarding the module.

---

> > > ### Author Response · Authors · 2024-11-27
> > > **Theory Explaination**
> > >
> > > Dear Reviewer b2Zv,
> > > As we have added more theoretical content in this revision, in order to prevent some confusion for you, we have explained it in detail to **Reviewer Fq1k**, and we hope that these explanations will also bring you a better reading experience.

---

> > > > ### Comment · Reviewer_b2Zv · 2024-11-28
> > > > **Rely to the rebuttal**
> > > >
> > > > I appreciate the additional experiments conducted by the authors and their further clarification. They address parts of my concerns. However, the motivation of approaching the problem from the frequency perspective is still not very convincing. In addition, the concerns regarding the presentation, contributions, and efficacy of the proposed method remain, so I adjust my score slightly.

---

> > > > > ### Author Response · Authors · 2024-12-01
> > > > > **Frequency perspective Benifts(2/2)**
> > > > >
> > > > > ## Experiment
> > > > > We also supplemented the analysis by comparing the $\mathcal{A}$-distance before and after applying deterministic IRM gains using different IRM methods. The results showed that incorporating deterministic components can yield purer invariant features.
> > > > > |Dataset|Target|IRM||IB-IRM||VREx||IIB||
> > > > > |---------|--------|--------|--------|--------|--------|---------|--------|--------|--------|
> > > > > |||full|$\alpha_{0.2/0.4}$|full|$\alpha_{0.2/0.4}$|full|$\alpha_{0.2/0.4}$|full|$\alpha_{0.2/0.4}$|
> > > > > |DSADS|0|0.8338|$\textbf{0.8235}\downarrow$|0.9329|$\textbf{0.8235}\downarrow$|0.8524|$\textbf{0.8235}\downarrow$|0.8833|$\textbf{0.8235}\downarrow$|
> > > > > ||1|0.8482|0.8513|0.8679|$\textbf{0.8390}\downarrow$|0.86377|$\textbf{0.8307}\downarrow$|0.9195|$\textbf{0.8235}\downarrow$|
> > > > > ||2|1.0319|$\textbf{0.8421}\downarrow$|0.9391|0.9473|0.8235|0.8431|1.0061|$\textbf{0.8338}\downarrow$|
> > > > > ||3|0.8235|0.8534|0.9287|$\textbf{0.9102}\downarrow$|0.9752|$\textbf{0.8235}\downarrow$|0.8482|$\textbf{0.8235}\downarrow$|
> > > > > ||avg.|$0.8841_{\pm0.09}$|$\textbf{0.8425}_{\pm0.01}\downarrow$|$0.9171_{\pm0.03}$|$\textbf{0.8800}_{\pm0.05}\downarrow$|$0.8787_{\pm0.06}$|$\textbf{0.8302}_{\pm0.01}\downarrow$|$0.9142_{\pm0.06}$|$\textbf{0.8260}_{\pm0.01}\downarrow$|
> > > > >
> > > > > you can see the details in **C.6 TIME-DETERMINISTIC GAIN PHENOMENON FOR IRM**, we tested all the datasets.
> > > > >
> > > > > In addition, we also provide the visualization in **C.7 VISUALIZATION STUDY** it shows how frequency works and  It can be seen that, through masking, the diversity between domains is more pronounced, while many redundant frequencies are eliminated

---

> ### Author Response · Authors · 2024-11-28
> **Frequency perspective Benifts(1/2)**
>
> Thank you for your reply!
> ## Theory and Related Work with Frequemcy
> I would like to clarify to you that frequency domain information has been proved to be robust in many tasks, we have mentioned them in our related work, and I would like to add more details to you here.
>
> 1. For non-staninary time series forcasting, many methods try to use both frequency and time to extractor global and frequency features, these works have mentioned that frequency features are more stable and have global perspective. Some methods try to demonstrate that **removing redundant frequencies by using the orthogonality of frequencies can reduce the variance**[1][2]. (Statistic level)
>
> 2. For time series pre-training task, some works try to compare the learning domain training in both time and frequency domains, and it is proved that **the frequency domain has a more global perspective, and obtains more stable features, which can be migrated to cross-dataset, cross-person, one-to-all task**[3].(Representation level)
>
> 3. For time series domain adaptation, **some works have mentioned that the frequency domain information is more stable than the time information distribution, and proposed a specific form of the frequency domain joint marginal distribution magnitude and phase**[4], which is followed in our paper **Lemma 4 (Distribution of Fourier Component)**,**Corollary 1 (Frequency Marginal Probability Distribution of Time Series Data)**. (Distribution level)
>
> 4. For Domain Generalization, unlike the first three approaches that use frequency domain information at the level of **representation learning**, a common strategy of using the frequency domain is to utilize frequency domain information for **data augmentation**, and **the effect of the magnitude and phase of the frequency domain components of a fine-grained study on data generation**[5][6] is demonstrated in these approaches. (Data Generation level)
>
> In contrast, our approach attempts to introduce frequency domain information to the representation learning level for Domain generalization, demonstrating **the importance of extracting domain-invariant features from frequency domain information, and proving the proof of the relationship between domain divergence and frequency domain**, you can see the details **Proposition 5.2 (Frequency Perspective Risk Bound on Unseen Time series Domain)** and **Empirical Domain Divergence Study**, In addition, Through these experiments we were able to verify that our experimental results were in agreement with the theory and determined that the optimal frequency domain to use is usually in the range of 0.2-0.4. (Fundamental Domain Generalization Error Bound level)
>
> If you feel that these explanations can eliminate your confusion about frequency, we also hope that you will reconsider the contribution of our theory and methods or if you have better suggestions, we will continue to add relevant experiments and text to modify our submitted version, **we still have the opportunity to modify the pdf today, and also have close to a week to improve our work**.
>
> Thank you again for your support of our work!
>
> [1] Ye W, Deng S, Zou Q, et al. Frequency Adaptive Normalization For Non-stationary Time Series Forecasting[J]. arXiv preprint NIPS 2024.
>
> [2] Liu Y, Li C, Wang J, et al. Koopa: Learning non-stationary time series dynamics with koopman predictors[J]. Advances in Neural Information Processing Systems NIPS 2023.
>
> [3] Zhang X, Zhao Z, Tsiligkaridis T, et al. Self-supervised contrastive pre-training for time series via time-frequency consistency[J]. NIPS 2022.
>
> [4] He H, Queen O, Koker T, et al. Domain adaptation for time series under feature and label shifts[C]//International Conference on Machine Learning. ICML 2023.
>
> [5] Demirel B U, Holz C. Finding order in chaos: A novel data augmentation method for time series in contrastive learning[J]. NIPS 2023.
>
> [6] Xu Q, Zhang R, Zhang Y, et al. A fourier-based framework for domain generalization[C]//Proceedings of the IEEE/CVF conference on computer vision and pattern recognition. CVPR 2021.

---

### Official Review · Reviewer_dy8u · 2024-11-12

**Soundness:** 3
**Presentation:** 2
**Contribution:** 3
**Rating:** 5
**Confidence:** 4

**Summary:**

This paper proposes a novel approach (FEDNet) for out-of-distribution (OOD) generalization in time series classification by leveraging frequency decomposition. FEDNet decomposes time series data into two components: a time-deterministic part, which is stable across different domains, and a time-stochastic part, which captures time-dependent variations.

**Strengths:**

- The paper presents an interesting and potentially impactful approach to OOD generalization in time series classification using frequency decomposition.
- The frequency-based separation into time-deterministic and time-stochastic components is theoretically motivated and could offer a meaningful advancement for handling domain and temporal shifts.
- The authors conduct experiments across multiple datasets and compare FEDNet with a wide range of baseline methods, demonstrating its potential strengths in OOD scenarios.

**Weaknesses:**

- The authors claimed that traditional IRM-based domain generalization methods are "not applicable" to time series data. I find this statement inaccurate. They could be applicable but suboptimal.


- The definition of temporal distribution shift (def 3) is unclear. It is not clear why or how different temporal positions in the raw data imply different distributions. It is quite normal the timesteps are different. It should rather be on the temporal segment level.
- Also, definition 4 introduces the connection between temporal shifts and frequency shifts, but the reason why temporal shifts inherently lead to frequency shifts is not clear for me.

- Wold’s Theorem is the basis for separating deterministic and stochastic components, but the connection to FEDNet is weakly established. The decomposition itself is mentioned without context on why Wold’s Theorem specifically supports this approach for OOD time series classification.

- Also, using Hilbert space and Bochner’s theorem is poorly integrated. The authors mentioned that "we can transform such data into a Hilbert space" without explaining why this transformation is needed or how it benefits the model. If Bochner’s theorem and Hilbert space are central to the method, they need more justification and background.

- There are some issues with the usage of notations. For example, in Equation 3, terms like ωλ and λ-th frequency component are introduced without any definition, and their role in the decomposition is unclear. Similarly, the notation for masking (Mask [S_α]) lacks detail.
- Also, λ is reused again in Eq. 13 as a hyperparameter, causing an inconsistency issue.
- I find that variational inference to disentangle domain-invariant and domain-specific features in the time-deterministic block is ok. However, the explanation lacks clarity on how this disentanglement process specifically aids in handling domain distribution shifts in OOD classification.
- In Proposition 4.1, it is not clear how frequency decomposition leads to time-deterministic and time-stochastic components that address OOD challenges.

- It would be great if the authors could study the impact of frequency filter masking levels to assess the impact of different frequency filtering thresholds and the masking operation in isolating stable (time-deterministic) components.

- The authors observe that the coefficient of variation is lower for time-deterministic components than for time-stochastic components. However, they do not discuss the implications of this finding in enough depth.

- The authors state that removing the time-stochastic block has only a "small effect" on performance (which is also clear from the ablation study). Does this mean that this component is less crucial and adds extra unnecessary computations?

- Page 3, sentences 109 to 112 are not clear.
- The authors have an issue with the named citation, which causes a readability issue. Note that ICLR template has several ways of citations. Check the difference in the output of these two statements.
     - Domain Generalization \cite{wang} is a difficult … -->  Domain Generalization (Wang et al. 2022a) is a difficult …
     - \citep{athor_x} mentioned that DG is a difficult … --> Wang et al. (2024) mentioned that DG is a difficult …

Use the first when you want just to cite. Use the second when you want to include the author's name in the text.

**Questions:**

- The authors need to revise:
    - the motivations
    - the definitions
    - the clarity of the writing

based on the above comments.

---

> ### Author Response · Authors · 2024-11-24
>
> **Writing problem(W1,W6,W7,W13,W14)**: Thank you very much for your very useful comments on our thesis writing! This is our first time submit to ICLR, and we ask for your forgiveness for the problems caused by the citation of the first draft.
> We have reworked all the issues you mentioned and changed to suitable citation:
> - We have revised the description of IRM in the introduction, specifically on line 47.
> - We elaborated on the key ideas of the three types of domain generalization methods, in lines 112-118.
> - We updated all citation formats throughout the text for your convenience in reading the details.
> - We resolved the issue of the repeated $\lambda$ hyperparameter, in line 230 and line 343, Eq. 16.
>
> **Definition of Temporal Distribution Shift(W2)**:We sincerely apologize for any misunderstanding caused by our writing. We have revised the definitions and the challenge in figure 1 to be more clear, **the mentioned temporal distribution shift is consistent with your understanding**. **Essentially, it stems from the non-stationarity of time series in the real world**. In the new definition, we explicitly introduce the concept of **period** to illustrate that time-series data across different time periods exhibit **marginal probability distribution** changes. This results in segments within the same domain not being independently and identically distributed. Therefore, the domains we define in practice are not entirely ideal.
>
> This is also a key reason why out-of-distribution (OOD) generalization in time series differs from that in static image datasets. We believe this phenomenon is not limited to time series. **Fundamentally, we aim to argue the broader issue of how to improve a model's generalization ability when the data within a domain does not fully follow a single distribution.** This perspective aligns more closely with real-world scenarios, as the domains we define manually often deviate from perfection.
>
> **Definition of Frequency Distribution Shift(W3)**: The relationship between time series data and frequency components can often be rigorously expressed through Fourier transform as an integral summation $\mathcal{F}\{f(t)\} = F(\omega) = \int_{-\infty}^{\infty} f(t) e^{-i \omega t},dt$. In fact, the intent here is to review this time series distribution problem from frequency perspective. We provide the basic formulation of the Fourier frequency domain distribution **Distribution of Fourier Component** and **Frequency Marginal Probability Distribution of Time Series Data** in **Appendix A.2 PRELIMINARY**. Since the concept of time series distribution cannot be simply defined, if we consider the most fundamental non-stationarity property, such as changes in mean $\mu_{t}$ and variance $\sigma_{t}$.These changes in $x_{t} = \mu_{t} + \sigma_{t} \times z$, where $z$ could be an stationary componets could reflect to variations in the amplitude and phase of $n$ **independent frequencies**. Exploring the magnitude of these changes impact on the frequency domain components is precisely the core focus of our paper's analysis and modeling efforts.

---

> ### Author Response · Authors · 2024-11-24
>
> **Frequency Composition(W4,W5,W6,W11)** : Sorry for the misunderstanding of our decomposition method due to my effectively limited English writing skills. Here we use Wold's theorem, which is in fact based on the basic assumption of the temporal shift problem, i.e., the temporal data belonging to the same time period within the domain will follow the same distribution, it fits the local weak-sense stationarity. **The main benefit of weak-sense stationarity is that any time series can be put into the context Hilbert Space. It means any time series can be decomposed by a set of orthogonal increments in the space.** Bochner's theorem **ensures that there exists** a group of Fourier-type orthogonal complex exponential function $\{e^{-2 \pi i \xi t}\}$ componets to generate $x_t$ in Hilbert Space. We have modified the text to make it more precise. Due to the orthogonality of frequencies and the commom shared space within different time series periods, it inspired us to explore whether the deterministic and stochastic components are dominated by specific frequencies, even in different periods and domains. This idea was verified in our experiments on the coefficient of variation for frequency domain decomposition. **These two components are dominated by two different sets of frequencies, so that we could study and address the two different distribution seperately to get a purified invariant features.** We provide an intuitive example for you: $x_t = \alpha sin(w_0 t) + \beta sin(w_t t) + \epsilon_t $ where $\alpha, \beta, w_0$ do not change with time, $\epsilon_t$ is white noise and $w_t$ varies with peroids. $\alpha sin(w_0t)$ can be regard as time-deterministic components while $\beta sin(w_t t)$ can be seen as time-stochastic components. There are already some related works[1][2] that attempts to use this decomposition idea to solve non-stationary problems.
>
> [1] Ye W, Deng S, Zou Q, et al. Frequency Adaptive Normalization For Non-stationary Time Series Forecasting. NIPS 2024
>
> [2] Liu Y, Li C, Wang J, et al. Koopa: Learning non-stationary time series dynamics with koopman predictors. NIPS 2023

---

> ### Author Response · Authors · 2024-11-24
>
> **time-deterministic with variational inference(W8)**: Thank you very much for mentioning this insightful issue! We analyzed the invariant features in different methods using the classical domain generalization metric $\mathcal{A}$-distance[1]. It can be approximated as $d = 2(1 - 2\sigma_{\mathcal{A}})$, where $\sigma_{\mathcal{A}}$ is the risk of a binary classifier distinguishing features between source and target domains. $d \in [0, 2]$ and the smaller the indicator, the more invariant features are. We measured model invariance features for multiple domains of the EMG dataset. This experiment has been added to the paper.
> |   Model  |  EMG domain-0 | EMG domain-1 |EMG domain-2| EMG domain-2|
> |  ----  | ----  |----  |----  |----  |
> | IRM  $\mathcal{A}$-distance| 0.8344  |0.7638 |0.8551  |0.8327|
> | FEDNet cross  $\mathcal{A}$-distance| 0.4664 | 0.4320  |0.4539 |0.4664|
> | FEDNet contrast  $\mathcal{A}$-distance| **0.3570** | **0.4204**  |**0.4320** |**0.3482**|
>
> **We found that our proposed variational inference approach can learn stronger invariant features compared to IRM.** Theoretically, classical IRM often faces parameter overload failures when addressing domain generalization problems. When the number of model parameters is relatively large, the regularization of IRM tends to cause the model to overfit[2][3]. Some studies have also demonstrated that utilizing variational inference with decoupling enhancements is effective for domain generalization [4].
>
> **In addition, we found that using contrastive learning loss can further reduce the $\mathcal{A}$-distance. This suggests that, compared to standard cross-label loss, contrastive learning can produce more robust invariant features.**  This is likely because contrastive learning inherently possesses properties similar to DRO (distributionally robust optimization) [5].
>
> We also supplemented the analysis by comparing the $\mathcal{A}$-distance before and after applying deterministic IRM gains using different IRM methods. The results showed that incorporating deterministic components can yield purer invariant features.
> | Dataset | Target | IRM    |        | IB-IRM |        | VREx    |        | IIB    |        |
> |---------|--------|--------|--------|--------|--------|---------|--------|--------|--------|
> |         |        | full   | $\alpha_{0.2/0.4}$  | full   | $\alpha_{0.2/0.4}$    | full    |   $\alpha_{0.2/0.4}$   | full   | $\alpha_{0.2/0.4}$   |
> | DSADS   | 0      | 0.8338 | $\textbf{0.8235}\downarrow$ | 0.9329 | $\textbf{0.8235}\downarrow$ | 0.8524  | $\textbf{0.8235}\downarrow$ | 0.8833 | $\textbf{0.8235}\downarrow$ |
> |         | 1      | 0.8482 | 0.8513 | 0.8679 | $\textbf{0.8390}\downarrow$  | 0.86377 | $\textbf{0.8307}\downarrow$ | 0.9195 | $\textbf{0.8235}\downarrow$ |
> |         | 2      | 1.0319 | $\textbf{0.8421}\downarrow$ | 0.9391 | 0.9473 | 0.8235  | 0.8431 | 1.0061 | $\textbf{0.8338}\downarrow$ |
> |         | 3      | 0.8235 | 0.8534 | 0.9287 | $\textbf{0.9102}\downarrow$ | 0.9752  | $\textbf{0.8235}\downarrow$ | 0.8482 | $\textbf{0.8235}\downarrow$ |
> |         | avg.   | $0.8841_{\pm 0.09}$ | $\textbf{0.8425}_{\pm 0.01}\downarrow$ | $0.9171_{\pm 0.03}$ | $\textbf{0.8800}_{\pm 0.05}\downarrow$   | $0.8787_{\pm 0.06}$  | $\textbf{0.8302}_{\pm 0.01}\downarrow$ | $0.9142_{\pm 0.06}$ | $\textbf{0.8260}_{\pm 0.01}\downarrow$  |
>
> [1] Ben-David S, Blitzer J, Crammer K, et al. Analysis of representations for domain adaptation[J]. NIPS 2006.
>
> [2] Yong Lin, Hanze Dong, Hao Wang, Tong Zhang, Bayesian Invariant Risk Minimization, CVPR 2022.
>
> [3] Xiao Zhou, Yong Lin, Weizhong Zhang, Tong Zhang, Sparse Invariant Risk Minimization, ICML 2022.
>
> [4] Towards Principled Disentanglement for Domain Generalization, CVPR 2022.
>
> [5] Wu J, Chen J, Wu J, et al. Understanding contrastive learning via distributionally robust optimization. NIPS 2023.

---

> > ### Author Response · Authors · 2024-11-24
> >
> > **empirical domain divergence study with $\alpha$ isolated level(W10)**
> >
> > We investigated the variations of empirical domain divergence and $\alpha$ under different isolation levels [0.2, 0.4, 0.6, 0.8] from both **theoretical** and **experimental** perspectives.
> >
> > 1. **Theoretical study.**
> > We theoretically derived the fundamental upper bound of domain generalization from a frequency-domain perspective, based on Germain P's PAC-Bayesian theorem[1]. This form of domain generalization upper bound differs from the traditional $H$-divergence [2] measurement, as it measures the divergence between domains using marginal probability distributions. In our paper, **we provide a mathematical proof and demonstrate that reducing frequencies can decrease the gap between domains**. Through numerical analysis, with the number of frequencies $n$ = 100 (corresponding to a maximum time series length $L$=200 usually), we found that the approximate optimal frequency ratio $\alpha$ lies in the range of 20% to 40%. For detailed information on this, please refer to **Appendix A, Section 4, Proposition 5.2 ("Frequency Perspective Risk Bound on Unseen Time Series Domain")**, and **Empirical Domain Divergence Study** section.
> >
> > [1] Germain P, Habrard A, Laviolette F, et al. A new PAC-Bayesian perspective on domain adaptation. ICML 2015.
> >
> > [2] Ben-David S, Blitzer J, Crammer K, et al. A theory of learning from different domains. Machine learning, 2010, 79: 151-175.
> >
> > 2. **Experimental study.**
> > In our theoretical study, an anomalous dataset, UniMiB-SHAR, was identified. This anomaly is primarily due to the volatility metric in its denominator, $\hat{\gamma}<0.01$, being extremely small, which caused the overall calculation of the generalization upper bound to be overestimated. As a result, the empirical estimation of the generalization bound for this dataset failed. **However, from an experimental perspective, we verified that this dataset still aligns with the theoretical conclusions.**
> >
> > | target        | FFT-0.2 | FFT-0.4 | FFT-0.6 | FFT-0.8 | DWT-0.2 | DWT-0.4 | DWT-0.6 | DWT-0.8 |
> > |---------------|---------|---------|---------|---------|---------|---------|---------|---------|
> > | UniMiB-SHAR domain-0 | 57.29   | **61.19**   | 58.59   | 58.59   | 55.22   | 56.77   | **59.89**  | 58.07   |
> > | UniMiB-SHAR domain-1 | **67.75**   | 65.00   | 67.06   | 62.55   | 58.83   | **73.58**   | 66.27   | 66.03  |
> > | UniMiB-SHAR domain-2 | 67.10   | **69.73**   | 65.13   | 68.42   | 75.65   | **75.65**   | 74.01   | 73.68  |
> > | UniMiB-SHAR domain-3 | 35.23   | **42.95**   | 38.25   | 38.92   | 40.26   | **44.29**  | 40.26   | 38.59   |
> >
> > We conducted experiments using $\alpha=[0.2,0.4,0.6,0.8]$ and concluded that once the invariant features reach optimal performance at a lower ratio, further increases do not yield significant gains. When the ratio exceeds 0.8, the performance may even deteriorate compared to lower ratios.
> >
> > **Additionally, we supplemented IRM gain experiments for all datasets and found that the results were consistent with our proposed domain divergence estimates.** The only exception was UniMiB-SHAR, which showed relatively poor gains. You can see the details in **Table 12**.

---

> ### Author Response · Authors · 2024-11-24
>
> **time-stochastic study(W12, W9)**
> The main purpose of the design of time-stochastic is to prevent the loss of information, and another aspect is to serve as auxiliary classification information to prevent the invariant features from over-dependence on the source domain, and increase the space of domain generalisation,, meanwhile, we found that when the time sequential data are missing to interfere with the invariant features, the supplemental time-stochastic information can significantly improve the gain, we We designed three different levels of experiments (faeture, data, domain) to illustrate the role of time-stochastic, detailed information can be found in the appendix **C.3 TIME-STOCHASTIC FEATURE STUDY** of the paper as well as the response of **Reviewer b2Zv time-stochastic study(W4)**.
>
> We have revised **Proposition 5.1** in the paper and combined it with the experimental findings to clarify the roles of the two components, and we have revised the proof section to incorporate the new definitions, which hopefully clears up any confusion you may have.

---

> > ### Author Response · Authors · 2024-11-27
> > **Theory Explaination**
> >
> > Dear Reviewer dy8u,
> > As we have added more theoretical content in this revision, in order to prevent some confusion for you, we have explained it in detail to **Reviewer Fq1k**, and we hope that these explanations will also bring you a better reading experience.

---

> > > ### Author Response · Authors · 2024-11-28
> > > **Benefits from Frequency**
> > >
> > > Dear Reviewer dy8u
> > >
> > > we clearify more details about the benefits of the frequency information, and our theory contributions, which can be seen in the comments **Frequency perspective Benifts** with **Reviewer b2Zv**.

---

> ### Author Response · Authors · 2024-12-02
> **Have we clearly address your concerns?**
>
> Dear Reviewer dy8u,
> We would like to remind you today is the last day for discussion, we really did many experiments to address your concerns. Hope you can have a better understanding on this version, we provide more interesting theory and experiment insight despite the time is limited :)

---

### Author Response · Authors · 2024-11-24

Dear AC and Reviewers,

We would like to sincerely thank the reviewers for their positive feedback and insightful suggestions. For clarity and readability of the paper, the following changes have been made to the manuscript and a new version uploaded. We've marked all the changes in blue.
- We have modified the introduction for background related to domain generalization in the introduction
- We give a more precise definition of temporal distribution shift, pointing out that its essence stems from the changes in the marginal probability distribution caused by the non-stationarity of the time series. We also modified the description of the time-series ood problem in Figure 1.
- We proposed Proposition 5.2 Frequency Perspective Risk Bound on Unseen Time series Domain with proof and Empirical Domain Divergence Study to illustrate the effectiveness of using frequency information for domain generalization. Our proofs do not involve complex assumptions and only need to ensure the basic domain generalization covariate shift assumption, i.e.,  the conditional probability of model feature over a label is unchaged under any shift.
- We add the theoretical background involved in 5.1 Frequency Filters and our motivation for using it
- We complemented this with more diverse dataset performance experiments and contrastive loss ablation studies and fix some data preprocess code before in PAMAP and retest it results.
- We add more study with time-deterministic and time-stochastic features in the Appendix and revise the Proposition 5.1 with time-deterministic and time-stochastic to address time series OOD problem.
- We move the anonymous code link from appendix to abstract and we provide all the scripts and the empirical domain divergence estimation jupyter-notebook for easy implementation.

Thanks again to everyone for their efforts!

The Authors

---

### Meta-Review · Area_Chair_D7a3 · 2024-12-22

**Metareview:**

This paper proposes a method called FEDNet for out-of-distribution (OOD) generalization in time series classification. FEDNet leverages frequency decomposition to separate time series data into time-deterministic and time-stochastic components, aiming to improve the model’s ability to handle domain and temporal shifts. However, multiple reviewers raised concerns about the paper's motivation. The connection between FEDNet’s theoretical foundations and its practical implementation is insufficiently justified. Moreover, reviewers noted that the proposed method shares significant similarities to existing frequency-enhanced approaches, limiting its novelty. Additionally, all reviewers emphasized the need for substantial improvements in the paper’s writing quality. Given these limitations, I recommend rejecting this submission.

**Additional Comments On Reviewer Discussion:**

During the rebuttal, 4 out 5 reviewers responded to the authors’ replies. Reviewer b2Zv was not satisfied with the rebuttal as the motivation of approaching the problem from the frequency perspective was still not very convincing. Reviewer 8bmq acknowledged the rebuttal and did not change the score (it is already a positive score). Reviewer YpR1 did not change the score due to concerns from the experimental perspective. Reviewer Fq1k decided not to change the score due to the novelty issue and the confusion about the mathematical formulation. Reviewer dy8u did not respond to the authors’ replies, however, I would think his/her concern about the unclear motivation was not well addressed by the authors.

---

### Decision · Program_Chairs · 2025-01-22

Reject